# RETHINKING THE EXPRESSIVENESS OF GNNS: A COMPUTATIONAL MODEL PERSPECTIVE

## ABSTRACT

Graph Neural Networks (GNNs) are extensively employed in graph machine learning, with considerable research focusing on their expressiveness. Current studies often assess GNN expressiveness by comparing them to the Weisfeiler-Lehman (WL) tests or classical graph algorithms. However, we identify three key issues in existing analyses: (1) some studies use preprocessing to enhance expressiveness but overlook its computational costs; (2) some claim the limited power of the identical-feature WL test while enhancing expressiveness using distinct features, thus creating a mismatch; and (3) some characterize message-passing GNNs (MPGNNs) with the CONGEST model but make unrealistic assumptions about computational resources, allowing NP-Complete problems to be solved in $O(m)$ depth. We contend that a well-defined computational model is urgently needed to serve as the foundation for discussions on GNN expressiveness. To address these issues, we introduce the Resource-Limited CONGEST (RL-CONGEST) model, incorporating optional preprocessing and postprocessing to form a framework for analyzing GNN expressiveness from an algorithmic alignment perspective. Our framework sheds light on computational aspects, including the computational hardness of hash functions in the WL test and the role of virtual nodes in reducing network capacity. Additionally, we suggest that high-order GNNs correspond to first-order model-checking problems, offering new insights into their expressiveness.

## 1 INTRODUCTION

Graph Neural Networks (GNNs) have attracted widespread attention in the graph machine learning community due to their impressive performance in areas such as recommendation systems, drug discovery, and combinatorial optimization. One key area of research has focused on characterizing the expressive power of existing GNNs and developing new models with enhanced expressive power. Existing work in this area typically aligns GNNs with various algorithms. One line of research focuses on connecting GNNs to the Weisfeiler-Lehman (WL) graph isomorphism test and its variants. For instance, Xu et al. (2019) pioneered the exploration of the relationship between message-passing GNNs (MPGNNs) and the WL test. Several studies (Morris et al., 2019; Maron et al., 2019; Feng et al., 2023) have proposed high-order GNNs inspired by the $k$-WL test and the $k$-Folklore WL (FWL) test, showing that these models exhibit stronger power compared to standard MPGNNs. Additionally, works such as (Alsentzer et al., 2020; Cotta et al., 2021; Papp et al., 2021; Feng et al., 2022; Frasca et al., 2022; Zhou et al., 2023a) introduced subgraph GNNs, where subgraphs are obtained through sampling or partitioning, followed by message-passing on these subgraphs. Furthermore, Zhou et al. (2023b); Zhang et al. (2024) analyzed the counting capabilities of different GNN types. Other studies focus on aligning GNNs with traditional graph algorithms. For instance, Zhang et al. (2023) designed a GD-WL framework, which incorporates precomputed distance information as additional features in message-passing, enabling the detection of graph biconnectivity. Additionally, Loukas (2020) attempted to align MPGNNs with the CONGEST model in distributed computing. They used existing lower bounds on the communication complexity of graph algorithms in the CONGEST model to derive lower bounds on the width and depth of MPGNNs when simulating these algorithms.

We carefully revisit these works and identify **inconsistent or unreasonable results** among them:

- **Underestimated Preprocessing Time Complexity.** Some existing works employ preprocessing techniques, such as substructure recognition or distance computation, to enhance their models and show that the proposed models can perform algorithmic tasks beyond the capabilities of the standard WL test. However, we observe that some of these works underestimate the computational cost of preprocessing, sometimes resulting in a preprocessing time complexity that exceeds that of the algorithmic task intended to show the model's expressiveness, which may lead to undesirable results, such as allowing directly precomputing answers to the algorithmic task as features.

- **Mismatch Between Identical-Feature WL Test and Distinct Features.** Existing works comparing GNNs with WL tests typically derive negative results regarding the WL test, such as its inability to distinguish certain toy example graphs. Some of these works then propose models that incorporate additional features to enhance expressiveness. We identify that their analysis of the WL test is based on the anonymous (identical-feature) setting, whereas the solution employs distinct features. Therefore, directly comparing the identical-feature WL test with models that violate this setting by incorporating distinct features creates a mismatch.

- **CONGEST Model Addresses Mismatch but Retains Unrealistic Assumptions.** A proposal suggests resolving the mismatch between the identical-feature WL test and distinct features by using the CONGEST model as a computational framework to characterize MPGNNs. However, we argue that directly adopting the CONGEST model as a computational model for GNNs can lead to unrealistic outcomes, such as enabling GNNs to solve many NP-Complete problems in $O(m)$ rounds, due to its implicit assumption of unlimited computational resources.

We conclude that these inconsistent or unreasonable results stem from the ad-hoc settings in these works, arising from the **lack of a well-defined computational model to characterize GNNs** and analyze their expressive power. Motivated by this, we propose the **Resource-Limited CONGEST (RL-CONGEST) model**, a simple and elegant computational framework for characterizing GNNs. This model extends the standard CONGEST model by introducing constraints on computational resources and incorporating optional preprocessing and postprocessing phases. It addresses key issues by accounting for the complexity of preprocessing and postprocessing, explicitly allowing the use of distinct features as node IDs, and imposing limitations on nodes' computational resources. Additionally, we present several **novel theoretical results using this model**:

- **WL Test Requires Large Networks to Compute.** We find that previous works have underestimated the complexity of the HASH function in the WL problem. In the RL-CONGEST model, if no preprocessing – such as graph modification or the use of additional features – is permitted, we prove that the HASH function in the WL test typically requires the network capacity (depth multiplied by width) to be linear in relation to the graph's size for computation.

- **Virtual Nodes Reduce Network Size for WL Test.** We present evidence suggesting that virtual nodes can enhance the performance of MPGNNs. Specifically, we prove that introducing a virtual node can reduce the network capacity required to compute one iteration of the WL problem.

- **Aligning High-Order GNNs with Model Checking is Natural.** Additionally, we leverage insights from descriptive and fine-grained complexity theories to argue that aligning high-order GNNs with the model checking problem is more natural.

The content of this paper is organized as follows: Section 2 introduces the notations used throughout the paper and provides a brief overview of the relevant background knowledge. In Section 3, we identify three key issues in existing works. In Section 4, we propose the RL-CONGEST model as a canonical computational framework for analyzing GNNs and present our theoretical findings. In Section 5, we present several open problems that may serve as directions for future work.

## 2 PRELIMINARIES

In this section, we first define the notations used throughout the paper. We then provide an overview of the relevant background knowledge, including GNNs, various variants of the WL tests, distributed computing models, and basic concepts in logic.

### 2.1 Notations

We use curly braces $\{\cdot\}$ to denote a set and double curly braces $\{\!\{\cdot\}\!\}$ for a multi-set where elements can appear multiple times. $[n]$ is shorthand for the set $\{0, 1, \ldots, n-1\}$. Boldface lowercase letters, such as $\boldsymbol{a}$ and $\boldsymbol{b}$, represent vectors, while boldface uppercase letters, such as $\boldsymbol{A}$ and $\boldsymbol{B}$, represent matrices. For two vectors of the same length $k$, we define the Hamming distance between them, denoted as $d_H(\boldsymbol{x}, \boldsymbol{y})$, as the number of differing coordinates. We denote a graph by $G = (V, E)$, where $V$ is the vertex set and $E \subseteq V \times V$ is the edge set. Unless otherwise specified, any graph $G$ mentioned in this paper is undirected, meaning that for any two nodes $i, j \in V$, $(i, j) \in E$ if and only if $(j, i) \in E$. Given a node $u$ in a graph $G$, the neighborhood of $u$, denoted by $N(u)$, is defined as $N(u) = \{v : (v, u) \in E\}$. We use $n \coloneqq |V(G)|$ to denote the number of nodes and $m \coloneqq |E(G)|$ to denote the number of edges when $G$ can be inferred from context. We use $D$ to denote the diameter of a graph, which is the length of the longest shortest path.

### 2.2 Graph Neural Networks and Weisfeiler-Lehman Tests

Graph Neural Networks (GNNs) are neural network models defined on graphs. The most prominent and widely used framework for implementing GNNs, as found in libraries like PyTorch-Geometric (Fey & Lenssen, 2019) and DGL (Wang et al., 2019), is the message-passing GNN (MPGNN) framework proposed by Gilmer et al. (2017). The MPGNNs can be formulated as:

$$\boldsymbol{h}_u^{(\ell+1)} = \textsc{Upd}^{(\ell)}\left(\boldsymbol{h}_u^{(\ell)}, \left\{\!\!\left\{\textsc{Msg}^{(\ell)}\left(\boldsymbol{h}_u^{(\ell)}, \boldsymbol{h}_v^{(\ell)}, \boldsymbol{e}_{(v,u)}\right) : v \in N(u)\right\}\!\!\right\}\right), \forall u \in V, \tag{1}$$

where $\boldsymbol{h}_u^{(\ell)}$ is the feature of node $u$ in the $\ell$-th layer, $\boldsymbol{e}_{(v,u)}$ is the edge feature on $(v, u)$, $\textsc{Upd}^{(\ell)}$ is the updating function in the $\ell$-th layer, and $\textsc{Msg}^{(\ell)}$ is the message function in the $\ell$-th layer, which maps the features of a pair of adjacent nodes and the edge feature to another vector called a message.

Xu et al. (2019) claim that the expressive power of MPGNNs is bounded by the Weisfeiler-Lehman (WL) test, which was proposed by Weisfeiler and Lehman in (Weisfeiler & Leman, 1968) as a graph isomorphism test. Initially, each node is assigned a natural number, called a color, from $[n]$ (usually, all nodes are assigned 0). The iteration formula of the WL test is as follows:

$$C^{(\ell+1)}(u) = \textsc{Hash}^{(\ell)}\left(C^{(\ell)}(u), \left\{\!\!\left\{C^{(\ell)}(v) : v \in N(u)\right\}\!\!\right\}\right), \forall u \in V, \tag{2}$$

where $C^{(\ell)}(u)$ is the color of node $u$ in the $\ell$-th iteration, and $\textsc{Hash}^{(\ell)}$ is a perfect hashing function mapping a multi-set of colors to a new color. It can be observed that the iteration formula of the WL test can be regarded as a special case of MPGNNs, where the message function outputs only the features of the neighboring nodes, and the updating function is a hashing function.

There are several variants of the standard WL test, and we will introduce some of them that will appear in our discussions later. A generalization is the higher-order WL tests, such as $k$-WL or $k$-FWL, which are defined on $k$-tuples of nodes in $G$. The updating formula for $k$-WL is described in (Huang & Villar, 2021) as:

$$C^{(\ell+1)}(\boldsymbol{u}) = \textsc{Hash}^{(\ell)}\left(C^{(\ell)}(\boldsymbol{u}), \{\!\{C^{(\ell)}(\boldsymbol{v}) : \boldsymbol{v} \in N_1(\boldsymbol{u})\}\!\}, \ldots, \{\!\{C^{(\ell)}(\boldsymbol{v}) : \boldsymbol{v} \in N_k(\boldsymbol{u})\}\!\}\right), \forall \boldsymbol{u} \in V^k, \tag{3}$$

where $\boldsymbol{u} = (\boldsymbol{u}_1, \ldots, \boldsymbol{u}_k) \in V^k$ is a $k$-tuple of nodes, and the $i$-th neighborhood of $u$ is defined as $N_i(\boldsymbol{u}) = \{(\boldsymbol{u}_1, \ldots, \boldsymbol{u}_{i-1}, v, \boldsymbol{u}_{i+1}, \ldots, \boldsymbol{u}_k) : v \in V\}$, consisting of all $k$-tuples in which the $i$-th coordinate is substituted with each node $v$. Meanwhile, the updating formula for $k$-FWL is described in (Huang & Villar, 2021) as:

$$C^{(\ell+1)}(\boldsymbol{u}) = \textsc{Hash}^{(\ell)}\left(C^{(\ell)}(\boldsymbol{u}), \{\!\{\left(C^{(\ell)}(\boldsymbol{u}_{[1]\leftarrow w}), \ldots, C^{(\ell)}(\boldsymbol{u}_{[k]\leftarrow w})\right) : w \in V\}\!\}\right), \forall \boldsymbol{u} \in V^k, \tag{4}$$

where $\boldsymbol{u}_{[i]\leftarrow w} = (\boldsymbol{u}_1, \ldots, \boldsymbol{u}_{i-1}, w, \boldsymbol{u}_{i+1}, \ldots, \boldsymbol{u}_k)$ is the $k$-tuple of nodes where the $i$-th coordinate in $\boldsymbol{u}$ is substituted with node $w$.

Another variant is the GD-WL framework proposed by Zhang et al. (2023), which is defined as:

$$C^{(\ell+1)}(u) = \textsc{Hash}^{(\ell)}\left(\left\{\!\!\left\{\left(d_G(u, v), C^{(\ell)}(v)\right) : v \in V\right\}\!\!\right\}\right), \forall u \in V, \tag{5}$$

where $d_G(u, v)$ is a distance, such as Shortest Path Distance (SPD) or Resistance Distance (RD).

High-order GNNs relate to high-order WL tests in the same way that MPGNNs relate to the standard WL test. In other words, if we replace the $\textsc{Hash}$ function in the updating formula of a variant of the WL test with another updating function $\textsc{Upd}$, we obtain a corresponding GNN model. Therefore, we sometimes use the terms WL tests and their corresponding GNN models interchangeably.

### 2.3 Distributed Computing Models

Distributed computing involves multiple processors collaborating to compute a common result. A distributed computing model is an abstract framework used to characterize this process. LOCAL and CONGEST proposed by (Linial, 1987; 1992; Peleg, 2000) are two classic distributed computing models based on synchronous message-passing between processors. In this paper, we follow the model definitions from (Ghaffari, 2022). These models are based on an $n$-node graph $G = (V = [n], E)$, where $G$ is assumed to be simple and undirected unless stated otherwise. Each node in the network hosts a processor. Initially, each processor knows the total number of nodes $n$, its unique identifier in $[n]$, and its initial features. In each round, a node computes based on its knowledge and sends messages to its neighbors, which may differ for each. By the end of the round, it receives all messages from its neighbors. In this model, each node must determine its own portion of the output. This process is described in (Loukas, 2020) as:

$$s_u^{(\ell+1)} = \textsc{Upd}_u^{(\ell)}\left(s_u^{(\ell)}, \left\{\!\!\left\{\textsc{Msg}_{v\to u}^{(\ell)}\left(s_v^{(\ell)}, v, u\right) : v \in N(u)\right\}\!\!\right\}\right), \forall u \in V, \quad (6)$$

where $s_u^{(\ell)}$ is the internal state (which may not be a vector) of the processor at node $u$. The primary difference between the LOCAL and CONGEST models is that, in each communication round, the LOCAL model permits nodes to exchange messages of unbounded length, while the CONGEST model restricts messages to a bounded length, typically $O(\log n)$.

### 2.4 Basic Concepts in First-Order Logic

First-Order Logic (FOL) is a formal system widely used in mathematics and various fields of computer science. An formula in FOL is composed of variable symbols such as $x$, $y$, $z$, and so on; punctuation symbols like parentheses and commas; relation symbols or predicates such as $P$, $Q$, $R$, and so forth; logical connectives including $\vee$, $\wedge$, $\neg$, $\rightarrow$, and $\leftrightarrow$; and logical quantifiers, specifically the universal quantifier $\forall$ and the existential quantifier $\exists$. A sentence is a special case of a formula where all variables are quantified; in other words, there are no free variables. We also introduce an extension to standard FOL called First-Order Logic with Counting (FOLC), which incorporates additional counting quantifiers. Specifically, for any natural number $i \in \mathbb{N}$, we define the counting quantifiers $\exists^{\geq i}$, $\exists^{\leq i}$, and $\exists^{=i}x$. The expression $\exists^{\geq i}x\varphi(x)$ ($\exists^{\leq i}x\varphi(x)$, $\exists^{=i}x\varphi(x)$) means that there exist at least (or at most, exactly, respectively) $i$ elements that satisfy the property $\varphi$. We use $\mathcal{L}^k$ and $\mathcal{C}^k$ to denote the sets of FOL and FOLC sentences, respectively, that use no more than $k$ variables.

## 3 Issues Due to Absence of Well-Defined Computational Model

Many studies have analyzed the expressiveness of GNNs, but they lack a well-defined computational model as a foundation, often relying on ad-hoc methods that lead to unreasonable outcomes.

### 3.1 Underestimated Preprocessing Time Complexity

Many GNNs fit into a "preprocessing-then-message-passing" framework. Given an input graph $G$ with features $\boldsymbol{X}$, they first perform preprocessing to build a new graph $G'$ with updated features $\boldsymbol{X}'$, followed by message passing on $G'$. For example, high-order GNNs based on $k$-WL and $k$-FWL tests construct graphs $G' = (V^k, E')$ on $k$-tuples of nodes, where $E' = \{(\boldsymbol{u}, \boldsymbol{v}) \in (V^k)^2 : d_H(\boldsymbol{u}, \boldsymbol{v}) = 1\}$. Subgraph GNNs and GNNs with additional precomputed features naturally fit this framework. These models typically target algorithmic tasks beyond the capabilities of the standard WL test; however, in some cases, the time complexity of the preprocessing phase exceeds that of the algorithmic task used to show the model's superior expressiveness, which is unreasonable from a complexity alignment perspective.

**Finding Pattern Graphs is Computationally Expensive.** One example comes from subgraph GNNs, which identify pattern subgraphs $H$ in the input graph $G$. We note that without restrictions on $H$, it implies overly powerful preprocessing capabilities. For instance, Thiede et al. (2021); Bouritsas et al. (2023); Wollschläger et al. (2024) proposed variants of GNNs and WL tests that utilize hand-crafted features by recognizing subgraphs. They argued that certain subgraph schemes could make models more powerful than $k$-WL for any $k$. However, these models achieve full expressiveness only when no constraints are imposed on $H$, implicitly assuming the existence of oracles

capable of counting isomorphic subgraphs. Such preprocessing requirements are overly strong from a theoretical perspective. The counting version of subgraph isomorphism is a #P-Complete problem, as its decision version is NP-Complete (Karp, 1972) and by the definition of #P-completeness. By Toda's Theorem (Toda, 1991) $\mathsf{PH} \subseteq \mathsf{P}^{\#\mathsf{P}}$, we conclude that a polynomial number of queries to a subgraph isomorphism counting oracle can solve any problem in the polynomial hierarchy PH, rendering such requirements implausible.

**Computing Distances Exceeds Biconnectivity Detection Time.** Another example can be found in distance-based GNNs. Zhang et al. (2023) introduced the GD-WL framework and proved that the WL test cannot recognize the biconnectivity of a graph, while the proposed framework can. However, this framework encounters the issue where the computation time for all-pair distances exceeds that of biconnectivity detection. Specifically, the biconnectivity of a graph can be determined in linear time, $O(m)$, using Tarjan's algorithm (Tarjan, 1972); whereas the worst-case time complexities for exactly computing all-pairs shortest path distances (SPDs) and resistance distances (RDs) are both $\tilde{O}(\min\{nm, n^{\omega}\})$[1]. Additionally, we point out through the following theorem that the "existence of more efficient approximate algorithms" claim remains debatable due to the lack of sensitivity analysis on the expressiveness with respect to error in the precomputed RDs.

**Theorem 1.** *For any integer $n \geq 3$, there exist two graphs with $n$ nodes, $P_n$ and $C_n$, such that for any two adjacent nodes in $P_n$, the resistance distance is $1$, while in $C_n$, it is $1 - 1/n$. Furthermore, $P_n$ is neither vertex- nor edge-biconnected, whereas $C_n$ is both vertex- and edge-biconnected.*

We defer the proofs of all theorems to the appendices to save space. Therefore, if the RD-WL framework requires the error to be less than $\Theta(1/n)$ to fully show the expressiveness, to the best of our knowledge, no existing approximation algorithms can compute all-pairs RDs within this error threshold in $o(\min\{nm, n^{\omega}\})$ time. Allowing higher preprocessing costs to solve a lower-complexity problem can lead to undesirable outcomes, such as using direct answers as features. The following theorem show this by demonstrating that precomputed RDs can be used to directly identify whether an edge is a cut edge, rendering subsequent message-passing phase unnecessary.

**Theorem 2.** *Given any edge $(u, v) \in E$ in an undirected, unweighted graph $G$, $(u, v)$ is a cut edge if and only if the resistance distance $R(u, v) = 1$.*

### 3.2 Mismatch Between Identical-Feature WL Test and Distinct Features

We observe that many existing works (Bouritsas et al., 2023; Zhang et al., 2023; Wollschläger et al., 2024) which compare GNNs' expressiveness to the WL test handling the anonymous (identical-feature) setting either in an ad-hoc manner. These works equate the expressiveness of MPGNNs with the WL test, which fails to distinguish toy examples. To address this, they propose adding features to enhance expressiveness. However, this approach may conflict with the identical-feature setting, and their models' expressiveness are not compared to distinct-feature MPGNNs (equivalent to CONGEST, as discussed in Section 3.3). Moreover, equating MPGNNs with the WL test overlooks the fact that real-world graphs often have features, making the claim that MPGNNs are weak based solely on the WL test questionable.

**WL Test is Weak in the Identical-Feature Setting.** As described by Huang & Villar (2021), the WL test operates under an identical-feature setting, where all nodes are initially assigned the same color and the process relies solely on node colors, making it unable to distinguish between nodes with identical colors during its iterations. This limited expressive power is often exemplified by its inability to distinguish certain graph pairs, such as two disjoint triangle graphs ($C_3 \cup C_3$) and a six-node cycle ($C_6$). As shown in Figure 1, the "type" of each node, which consists of its color and the multiset of its neighbors' colors, is the same (○, {{○, ○}}) for all nodes in both graphs. After applying the Hash function, all node colors remain identical, so the WL test cannot distinguish between the graphs due to the identical multisets.

[1]The notation $\tilde{O}(\cdot)$ hides polylogarithmic factors. We note that authors were not aware of more efficient algorithms; however, this does not affect our conclusion. For details on the complexity of computing all-pairs distances, please refer to Appendix D and E.

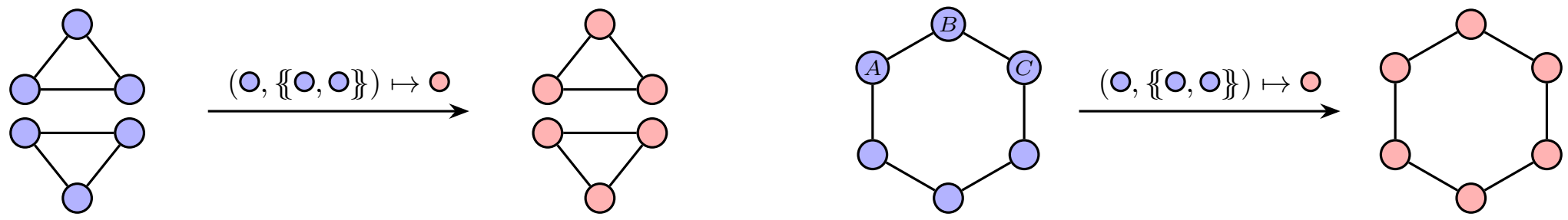

Figure 1: A running example of the anonymous WL test on $C_3 \cup C_3$ and $C_6$.

**Additional Features Empower Models by Breaking Anonymity?** Motivated by this limitation, many works such as (Zhang et al., 2023; Wollschläger et al., 2024) have proposed introducing features to enhance the expressiveness of GNNs. We speculate that these features may enhance the expressiveness by breaking the identical-feature setting. For example, in the $C_6$ graph in Figure 1, if nodes are assigned the same initial color, nodes $B$ and $C$ are indistinguishable from $A$'s perspective. However, if distances are included as features, $B$ and $C$ will not be equivalent. The distinct-feature setting is commonly used in existing models. For example, GCN (Kipf & Welling, 2017) and other models applied to real-world datasets use node features that are distinct with high probability. The LINKX model proposed by Lim et al. (2021) employs $\mathrm{MLP}(\boldsymbol{A})$, which can be reformulated as $\mathrm{MLP}'(\sigma(\boldsymbol{AIW}))$ and uses the identity matrix as features. In the GD-WL framework, the distance matrix serves as distinct features, as only the diagonal elements are zero and each row is unique. Additionally, works on GNN expressiveness (Loukas, 2020; Sato et al., 2021) adopt random features, which are also unique w.h.p.. As a result, previous works that equate GNNs with identical-feature WL tests to claim GNNs' weak expressiveness, while claiming to enhance expressiveness by distinct features, actually create an mismatch. We also include a discussion on the relationship between distinct features and permutation invariance or equivariance in Appendix A.

### 3.3 CONGEST Addresses Mismatch but Retains Unrealistic Assumptions

We observe that one existing work (Loukas, 2020) attempts to align MPGNNs with the CONGEST model, which unintentionally resolves the inconsistency in the identical-feature setting. However, directly using the CONGEST model as a computational model introduces a problem: nodes are assumed to have unlimited computational resources, which leads to impractical outcomes.

**Breaking Anonymity Empowers Models!** In the previous section, we noted that many variants enhance the expressive power by adding precomputed features, which may implicitly rely on node IDs to distinguish nodes. This raises a natural question: can we directly improve the expressive power by explicitly incorporating node IDs into the framework of the WL test?

When comparing the equation of MPGNNs with the WL test, it becomes clear that MPGNNs can be viewed as the WL test without the constraints of identical node features and the HASH function. Meanwhile, MPGNNs are similar to the CONGEST model if we compare Equation 1 and 6. Thus, MPGNNs and CONGEST models are expected to have stronger expressive power than the WL tests once these limitations are removed. Loukas (2020) provides evidence for this through the following theorem by aligning MPGNNs with LOCAL and CONGEST models:

**Theorem 3** ((Loukas, 2020))**.** *MPGNN can compute any computable function over connected graphs if the conditions are jointly met: (1) each node is uniquely identified; (2) the message and update functions are Turing-complete for every $\ell$; and (3) the depth and width are sufficiently large.*

Moreover, Pritchard & Thurimella (2011) proposed a CONGEST algorithm that solves the edge-biconnectivity problem in $O(D)$ rounds, challenging the claim that MPGNNs are weak.

**Direct Use of CONGEST Is Inappropriate.** In the aforementioned paper (Loukas, 2020), the author proposes using the CONGEST model from distributed computing to characterize MPGNNs, as it permits distinct features and supports more complex update functions compared to hash functions, making it a closer representation of the real-world implementation of MPGNNs than the WL test. However, we argue that directly using the CONGEST model as a computational model for MPGNN is not entirely appropriate, as unlimited computational resources assumption can lead to unrealistic and surprising results, as stated in the following theorem:

**Theorem 4.** *If we allow a single node to have unbounded computational power to solve any computable problem, then every* NP-Complete *decision problems on undirected unweighted connected graphs can be solved by the CONGEST model in $O(m)$ rounds.*

This unreasonable outcome shows that directly using the CONGEST model as the computational model for MPGNNs is inappropriate due to the unlimited computational resources for nodes.

## 4 Proposed Computational Model and Our Results

In the previous section, we discussed the inconsistent or unreasonable results in existing studies on the expressiveness of GNNs. We argue that these problems primarily arise from the lack of a well-defined computational model for GNNs, leading researchers to propose various ad-hoc solutions, some of which are inconsistent or unreasonable. In this section, we introduce a computational model to characterize GNNs, introduce how it addresses the aforementioned issues, and prove several interesting conclusions using this model.

We propose the Resource-Limited CONGEST (RL-CONGEST) model, an extension of the CONGEST model with constrained computational resources at each node, to serve as a computational framework for characterizing GNNs.

**Definition 1** (RL-CONGEST Model and Computation Process)**.** *Given a model width $w \in \mathbb{N}$ and a complexity class $\mathsf{C}$ (e.g., $\mathsf{TC}^0$[2], $\mathsf{P}$), the RL-CONGEST model with width $w$ and computational resource $\mathsf{C}$ is defined as a CONGEST model where message sizes are limited to $w \lceil \log |V(G)| \rceil$ bits, and nodes can solve any problems in $\mathsf{C}$.*

*For an attributed graph $G = (V, E, \boldsymbol{X}, \boldsymbol{E})$, where $\boldsymbol{X}$ and $\boldsymbol{E}$ represent node and edge features, the computation of a GNN using the RL-CONGEST model involves the following phases:*

1. ***(Optional) Preprocessing:*** *Operations such as building a hypergraph (for higher-order GNNs), extracting subgraphs (for subgraph GNNs), or computing additional features (for distance-based GNNs) occur in this phase, resulting in a new attributed graph $G' = (V', E', \boldsymbol{X}', \boldsymbol{E}')$. The time complexity of this step must be explicitly provided.*
2. ***Message-Passing with Limited Computational Resources:*** *Each node $u \in V'$ starts with its node features $\boldsymbol{X}'_u$ and the edge features of its incident edges $\left\{\left(v, \boldsymbol{E}'_{(u,v)}\right) : v \in N_{G'}(u)\right\}$. The message-passing proceeds as in the standard CONGEST model, but with each node allowed to update its internal state using computations in $\mathsf{C}$. The total number of communication rounds corresponds to the GNN model's depth $d$.*
3. ***(Optional) Postprocessing:*** *Additional computations, such as a* READOUT *operation, can be performed after message-passing. The time complexity must be explicitly stated.*

In Figure 2, we present a diagram illustrating the three phases of the computation process for a GNN using the RL-CONGEST model.

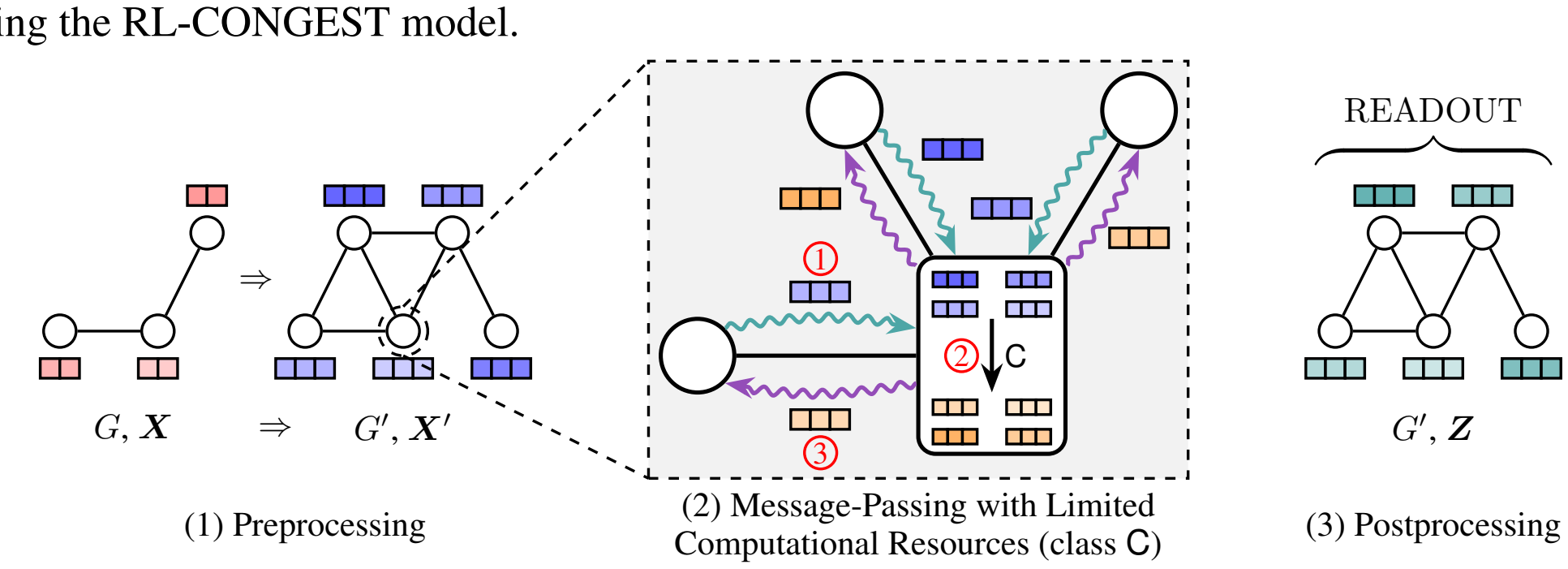

Figure 2: The three phases of computation in GNNs using the RL-CONGEST model.

We have essentially proposed a framework based on the RL-CONGEST model to characterize GNNs, which effectively addresses the three key issues identified earlier. Specifically:

- The RL-CONGEST model does not directly circumvent the "Underestimated Preprocessing Time Complexity" issue. Our framework requires that future works report the preprocessing complex-

[2]This is a complexity class of circuits. For further details on circuits, please refer to Appendix C.

ity. If it exceeds that of the algorithmic task, researchers should reassess whether preprocessing implicitly solves the task, as such results would not reflect the true expressiveness of the model.

- We address the "Mismatch Between WL Test and Features" issue by permitting nodes to know their own unique IDs. It is recommended that future works analyze the tasks GNNs can solve under this setting, rather than confining themselves to alignment with WL tests.
- We mitigate the "CONGEST Retains Unrealistic Assumptions" issue by allowing flexible configurations for the computational resources available to nodes. For instance, assuming each node operates as a Turing machine yields the standard CONGEST model. Alternatively, each node can be restricted to solving problems within $\mathsf{P}$, the same class of problems that LLM-based agents enhanced by Chain-of-Thought (CoT) reasoning can address Li et al. (2024); Merrill & Sabharwal (2024); or be modeled as a $\mathsf{TC}^0$ circuit, similar to MLPs (Shawe-Taylor et al., 1992; Beiu & Taylor, 1996), as observed in real-world GNN models. These different settings may lead to various interesting and independent results, and could even be extended to graph agents.

By using the RL-CONGEST model with preprocessing and postprocessing as our analysis framework, we can establish several interesting results and offer guidelines for future exploration of the expressiveness of high-order GNNs. These will be elaborated in the following subsections.

### 4.1 WL Test Requires Large Networks to Compute

In previous work aligning MPGNNs with the WL test to study their expressive power, researchers aligned the update function of MPGNNs directly with the HASH function in the WL test. Aamand et al. (2022) noted the challenges in constructing the HASH function for the WL test but did not establish a lower bound on the trade-off between network depth and width. Within the RL-CONGEST framework, we rigorously prove the relationship between the depth and width required for an MPGNN to simulate one iteration of color-refinement in the WL test. This enables us to prove that the HASH function in the WL test is computationally hard, as shown by the following theorem.

**Theorem 5.** *If an MPGNN can simulate **one** iteration of the WL test without preprocessing, either deterministically or randomly with zero error, regardless of the computational power available to each node, the model's width $w$ and depth $d$ must satisfy $d = \Omega\left(D + \dfrac{m}{w \log n}\right)$, given that $w = o\left(\dfrac{n}{\log n}\right)$.*

We defer the formal definition of the problem concerning one iteration of the WL test, along with the proof of the above theorem, to Appendix I. Notably, in our proof, we employed techniques from communication complexity without making any assumptions about the complexity class required for computational resources in the RL-CONGEST model. Therefore, the result also holds for the general CONGEST model, indicating that our findings – showing that WL-like HASH functions are hard to compute – are of independent interest to the field of distributed computing.

Furthermore, we design a deterministic RL-CONGEST algorithm with a round complexity that nearly matches the lower bound, indicating that the algorithm is near-optimal.

**Theorem 6.** *There exists a deterministic RL-CONGEST algorithm that can simulate **one** iteration of the WL test without preprocessing, with width $w$ and depth $d$ satisfying $d = O\left(D + \dfrac{m}{w}\right)$. Additionally, it is sufficient to set the nodes' computational resource class to $\mathsf{C} = \mathsf{DTIME}(n^2 \log n)$.*

### 4.2 Virtual Nodes Reduce Network Size for WL Test

Several works have attempted to enhance the performance of GNNs by introducing a virtual node that connects to all or some nodes in the original graph (Gilmer et al., 2017; Hwang et al., 2022). Subsequent studies have analyzed the impact of this node. For instance, Barceló et al. (2020) show that virtual nodes can bring GNNs closer to a $\mathcal{C}^2$ classifier, while Rosenbluth et al. (2024) compare MPGNNs with virtual nodes and graph transformers. To the best of our knowledge, no prior work

has explored how a virtual node helps reduce the network's capacity when simulating one iteration of the WL test. We address this in the following theorem:

**Theorem 7.** *There exists a deterministic RL-CONGEST algorithm that can simulate* ***one*** *iteration of the WL test by adding a virtual node, which connects to other nodes, as preprocessing. The algorithm operates with width $w$ and depth $d$ satisfying $dw = O(\Delta)$, where $\Delta$ is the maximum degree of the graph before the addition of the virtual node. Additionally, it is sufficient to set the nodes' computational resource class to $\mathsf{C} = \mathsf{DTIME}(n^2 \log n)$.*

Some studies suggest that virtual nodes do not enhance expressive power (Zhu et al., 2023), which contrasts with empirical evidence showing improvements in model performance. We find this is because they often equate "expressive power" with the ability to compute specific functions, akin to computability. Through our analysis, we introduce a computational model that provides a more refined view of expressive power by examining problem complexity and focusing on resource usage. This approach shows that virtual nodes can reduce the network size required to simulate the WL test, deepening our understanding of their impact.

### 4.3 Aligning High-Order GNNs with Model Checking is Natural

In this section, we show from both fine-grained and descriptive complexity perspectives that it is more natural to align higher-order GNNs with the $\mathcal{C}^k$ model checking problem. We will begin by introducing the model checking problem and the related model equivalence problem.

The Model Checking (MC) problem asks whether, given a model $A$ and a logic sentence $\varphi$, the sentence $\varphi$ holds in $A$ (i.e., $A \models \varphi$). In this paper, we focus on cases where the model is a graph $G$, and $\varphi$ uses only the edge predicate $E(x, y)$ and the equality predicate $=(x, y)$[3]. The $\mathcal{L}^k$ MC problem is highly expressive, capturing many key problems. For instance, deciding whether $G \models \varphi_\triangle$, where $\varphi_\triangle := \exists x \exists y \exists z (E(x, y) \wedge E(y, z) \wedge E(z, x))$, determines whether $G$ contains a triangle subgraph. Due to its expressiveness, the $\mathcal{L}^k$ MC problem has been widely studied. It applies to database queries like SQL (Gao et al., 2017), formal verification (Godefroid, 1997), and is central to fine-grained complexity in $\mathsf{P}$ (Puatracscu & Williams, 2010; Williams, 2014; Gao et al., 2017). In Appendix L, we provide evidence from theoretical computer science to support the classification of the PNF $\mathcal{L}^k$ model checking problem [4] in the $\tilde{\Theta}\left(\min\{n^k, m^{k-1}\}\right)$ complexity class.

Another related problem is the Model Equivalence (ME) problem. Given two models $A$ and $B$, and a class of logic sentences, the task is to determine whether for any sentence $\varphi$ in that class, $A \models \varphi$ if and only if $B \models \varphi$. In other words, the goal is to check whether that logic cannot distinguish between the two models. Two important results that connect descriptive complexity and WL tests were proven by Cai et al. (1989) and Grohe (2017), showing that, for any $k \geq 3$, the expressiveness of $(k-1)$-FWL and $k$-WL is equivalent to $\mathcal{C}^k$ ME problem. Another result by Grohe (1998) shows that the expressiveness of both the standard WL test and the 2-WL test is equivalent to $\mathcal{C}^2$ ME problem. This means that the output colors from WL tests provide only a "type" of the graph, and we cannot directly interpret it for specific tasks such as determining whether a graph contains a triangle or is biconnected. The most we can infer is that if two graphs produce the same color multisets, then either both contain a triangle (or are biconnected, respectively), or neither does.

Therefore, we argue that from a computational model perspective, it is more meaningful to discuss the expressiveness of GNNs in terms of solving problems, such as model checking, rather than limiting the discussion to model equivalence, which only determines whether graph pairs are indistinguishable. A natural approach is to align higher-order GNNs, inspired by $k$-WL or $(k-1)$-WL tests, with the $\mathcal{C}^k$ MC problem. We support this claim by proving the following weaker theorem:

**Theorem 8** (Informal)**.** *If constructing the $k$-WL graph and additional features as preprocessing is allowed, the RL-CONGEST model can solve the PNF $\mathcal{C}^k$ model checking problem in $O(k^2)$ rounds. Additionally, the computational resources required by each node are $\mathsf{C} = \mathsf{DTIME}(k^2 n)$.*

[3] We use $x = y$ and $x \neq y$ as abbreviations for $=(x, y)$ and $\neg =(x, y)$.

[4] A PNF sentence is of the form $(Q_1 x_1)(Q_2 x_2) \cdots (Q_k x_k) \phi(x_1, \cdots, x_k)$, where $Q_i$ are quantifiers and $\phi(x_1, \cdots, x_k)$ is a quantifier-free formula. Since variables can be reused, as in programming languages, a non-PNF sentence in $\mathcal{L}^k$ may not always be convertible to an equivalent PNF sentence that remains in $\mathcal{L}^k$.

Therefore, using the RL-CONGEST model as the computational framework, GNNs can go beyond WL tests, which only yield hard-to-interpret graph classification types rather than addressing specific problems like model checking.

As a supplement, we list several variants of the WL test, their corresponding ME problems, and the relationships between their expressive power on the left side of Figure 3. On the right side, we present the corresponding $\mathcal{C}^k$ MC problems, PNF $\mathcal{C}^k$ MC problems, and PNF $\mathcal{L}^k$ MC problems. On the far-right, the time complexity of the PNF $\mathcal{L}^k$ MC problems is displayed. It can be observed that the expressiveness and time complexity of the WL test variants, ME problems, and MC problems form a hierarchical structure. Our Theorem 8 positions $k$-WL graph-based higher-order GNNs within the category of PNF $\mathcal{C}^k$ MC problems. This is a weak result because the time complexity of constructing a $k$-WL graph, $O(kn^{k+1})$[5], already exceeds the time complexity of solving PNF $\mathcal{C}^k$ MC problems using non-distributed algorithms. We conjecture that higher-order GNNs, with $k$-WL graph construction as preprocessing, may have the potential to solve PNF $\mathcal{C}^{k+1}$ or general $\mathcal{C}^k$ MC problems. We present this as an open problem in Section 5.

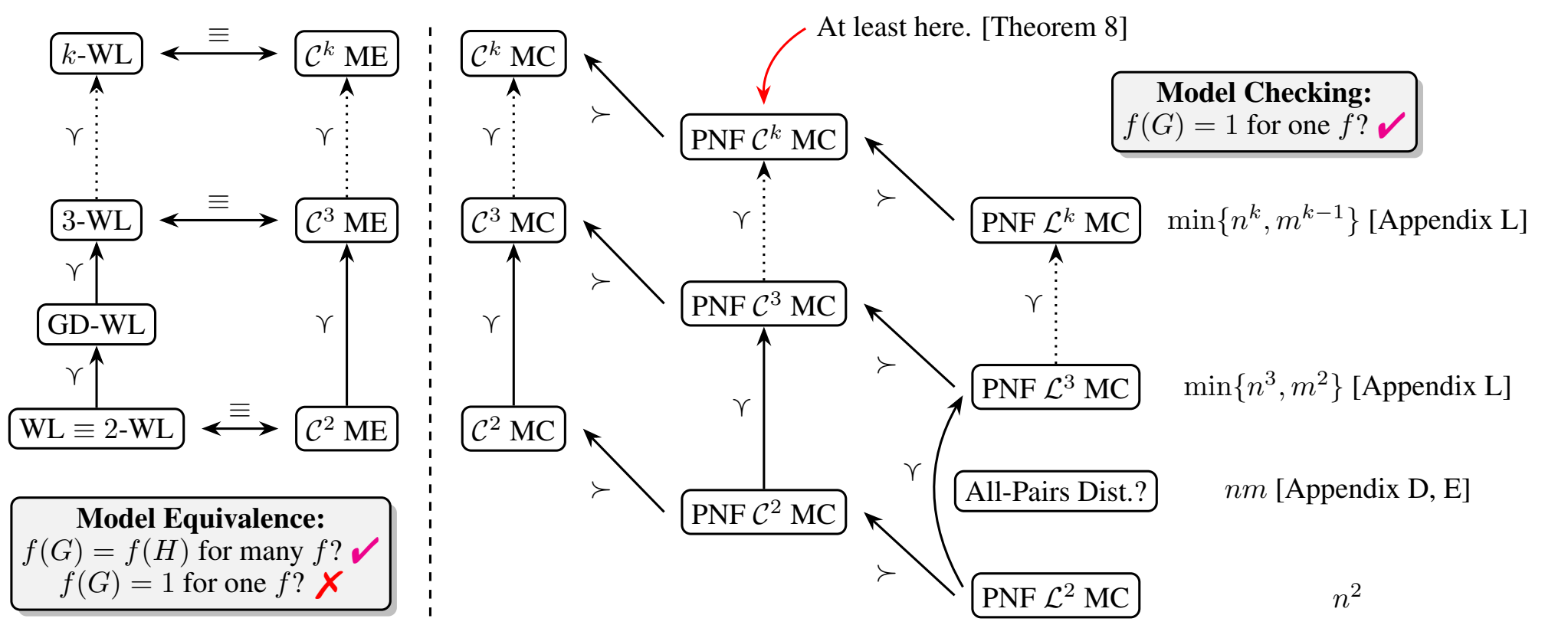

Figure 3: WL tests, model equivalence problems, model checking problems, their relationships, and the time complexity of PNF $\mathcal{L}^k$ model checking problems. The notation $A \prec B$ means $A$ is less powerful than $B$, while $A \succ B$ indicates that $A$ is more powerful than $B$. $A \equiv B$ signifies that $A$ and $B$ have the same expressive power.

## 5 Some Open Problems

Although we present some interesting results with our RL-CONGEST model and analysis framework, many open problems still remain, which are valuable for further research. We outline a few of them: (1) Can we establish a non-trivial trade-off between computational resources and round complexity in the RL-CONGEST model? (2) Is there a corresponding model equivalence problem, or other logic-related problems, for the GD-WL framework? (3) Do higher-order GNNs have the capability to solve the PNF $\mathcal{C}^{k+1}$ model checking problem or general $\mathcal{C}^k$ model checking problems?

## 6 Conclusions

In this paper, we identify three common issues in existing analyses of GNNs' expressive power, stemming from the absence of a well-defined computational model. To address this, we introduce the RL-CONGEST model, which includes optional preprocessing and postprocessing phases, as a standard framework for analyzing GNNs. Our framework addresses these issues and produces several noteworthy results, including the hardness of the WL problem, which may be of independent interest to the field of distributed computing. Additionally, we outline some open problems for potential future research.

[5] The time complexity of treating $k$-tuples as new nodes is $O(n^k)$, with each node connected to $k(n-1)$ neighbors, resulting in a total time complexity of $O(kn^{k+1})$.

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

# Appendices

# A A BRIEF DISCUSSION ON THE RELATIONSHIP BETWEEN DISTINCT FEATURES AND PERMUTATION INVARIANCE OR EQUIVARIANCE

Permutation invariance and equivariance are two vital properties to discuss whether a GNN model possesses them. They are defined as follows:

**Definition 2** (Permutation Invariant)**.** *Let $f$ be a graph neural network model on some graph $G$ with $n$ nodes, and let $\boldsymbol{P} \in \{0,1\}^{n\times n}$ be a permutation matrix. If for any input feature matrix $\boldsymbol{X} \in \mathbb{R}^{n\times F}$,*

$$f(\boldsymbol{PX}) = f(\boldsymbol{X}),$$

*then $f$ is said to be permutation invariant.*

**Definition 3** (Permutation Equivariant)**.** *Let $f$ be a graph neural network model on some graph $G$ with $n$ nodes, and let $\boldsymbol{P} \in \{0,1\}^{n\times n}$ be a permutation matrix. If for any input feature matrix $\boldsymbol{X} \in \mathbb{R}^{n\times F}$,*

$$f(\boldsymbol{PX}) = \boldsymbol{P}f(\boldsymbol{X}),$$

*then $f$ is said to be permutation equivariant.*

In other words, a model is permutation invariant (equivariant) if permuting the input features yields identical (or correspondingly permuted) outputs, respectively.

It should be noted that permutation invariance or equivariance are properties determined by the GNN architecture $f$, or more specifically, by the update functions in each layer, rather than by the features $\boldsymbol{X}$ used as input.

# B A BRIEF INTRODUCTION TO COMMUNICATION COMPLEXITY

To prove lower bounds on the rounds of CONGEST algorithms, a key tool is the communication complexity which was first introduced by Yao (1979).

Two-party communication complexity involves two participants, Alice and Bob, who collaborate to compute a function $f : X \times Y \to Z$, where $X$ and $Y$ are their input domains, respectively. They agree on a strategy beforehand but are separated before receiving their inputs $(x, y) \in X \times Y$. They then exchange messages to compute $f(x, y)$, with the goal of minimizing the total number of bits exchanged.

In deterministic communication, the strategy is fixed, and the minimum number of bits required to compute $f$ in this setting is known as the deterministic communication complexity, denoted by $D(f)$. Similarly, in randomized communication, where Alice and Bob can use random bits and a two-sided error of $\epsilon$ is allowed, the minimum number of bits required is the randomized communication complexity. If the randomness is private, it is denoted by $R_\epsilon^{\mathsf{prv}}(f)$, and if it is public, it is denoted by $R_\epsilon^{\mathsf{pub}}(f)$.

The Equality (EQ) problem between two $n$-bit strings, denoted by $\mathsf{EQ}_n : \{0,1\}^n \times \{0,1\}^n \to \{0,1\}$, is defined as

$$\mathsf{EQ}_n(\boldsymbol{x}, \boldsymbol{y}) = \begin{cases} 1, & \boldsymbol{x} = \boldsymbol{y}, \\ 0, & \text{otherwise.} \end{cases}$$

It is arguably the most well-known problem in two-party communication complexity which has been extensively studied. We summarize its communication complexity under different settings in Table 1 below.

# C A BRIEF INTRODUCTION TO BOOLEAN CIRCUITS

Boolean circuits are computational models used to represent Boolean function computations. A Boolean circuit with input size $n$ can be described as a Directed Acyclic Graph (DAG), with $n$ source nodes as inputs and one sink node as the output. The nodes represent gates, including NOT ($\neg$, with one input), AND ($\wedge$, with two or more inputs), OR ($\vee$, with two or more inputs), and threshold gates ($\mathrm{Th}_{\boldsymbol{a},\theta}$), which output 1 if and only if $\boldsymbol{a}^\top \boldsymbol{x} \geq \theta$, where $\boldsymbol{a}$ and $\theta$ are independent of the input $\boldsymbol{x}$.

Table 1: The communication complexity of the $\mathsf{EQ}_n$ function under different settings.

| Function | Deterministic | Randomized | | | |
|---|---|---|---|---|---|
| | | Private Coin | | Public Coin | |
| | $D(\cdot)$ | $R_0^{\mathsf{prv}}(\cdot)$ | $R_{1/3}^{\mathsf{prv}}(\cdot)$ | $R_0^{\mathsf{pub}}(\cdot)$ | $R_{1/3}^{\mathsf{pub}}(\cdot)$ |
| $\mathsf{EQ}_n$ | $\Theta(n)^\dagger$ | $\Theta(n)^\dagger$ | $\Theta(\log n)^\dagger$ | $\Theta(n)^*$ | $\Theta(1)^\dagger$ |

† The proofs can be found in (Kushilevitz & Nisan, 1997).

* Since $R_0^{\mathsf{prv}}(f) = O(R_0^{\mathsf{pub}}(f) + \log n)$, as per Exercise 3.15 in (Kushilevitz & Nisan, 1997).

A circuit family $(C_n)_{n\geq 1}$ is a sequence of circuits with input size growing from 1 to infinity. A family is L-uniform if a Turing machine can construct $C_n$ in $O(\log n)$ space, given $n$ in unary. Circuit families are assumed uniform unless otherwise stated. There are three major circuit complexity classes:

- $\mathsf{AC}^k$ consists of problems solvable by circuits with $\neg$, $\wedge$, and $\vee$ gates, polynomial size, depth $O(\log^k n)$, and unbounded fan-in.
- $\mathsf{NC}^k$ includes problems solvable by circuits with $\neg$, $\wedge$, and $\vee$ gates, polynomial size, depth $O(\log^k n)$, and fan-in 2.
- $\mathsf{TC}^k$ comprises problems solvable by circuits with polynomial size, depth $O(\log^k n)$, unbounded fan-in gates.

## D Time Complexity of All-Pairs Shortest Paths in Unweighted Undirected Graphs

The shortest path problem is one of the fundamental problems in graph theory. The All-Pairs Shortest Path (APSP) problem seeks to determine the shortest path distance between all pairs of nodes in a given graph $G$. To the best of our knowledge, the fastest algorithm for APSP on unweighted and undirected graphs can be formally stated as follows:

**Lemma 1** (Folklore; (Seidel, 1995))**.** *The computation of APSP for an unweighted, undirected graph with $n$ nodes and $m$ edges can be achieved with a time complexity of $\tilde{O}\left(\min\left(nm, n^\omega\right)\right)$, where $\omega < 2.372$ is the matrix multiplication exponent.*

## E A Brief Introduction to Resistance Distance

In this section, we introduce the concept of Resistance Distance (RD), covering its definition and the time complexity of approximately computing All-Pairs Resistance Distances (APRD). We begin with the definition of resistance distance:

**Definition 4** (Resistance Distance)**.** *Given an undirected graph $G$ and a pair of nodes $s$ and $t$, the resistance distance between $s$ and $t$, denoted by $R(s,t)$, is defined as:*

$$R(s,t) = (\boldsymbol{e}_s - \boldsymbol{e}_t)^\top \boldsymbol{L}^\dagger (\boldsymbol{e}_s - \boldsymbol{e}_t) = \boldsymbol{L}^\dagger_{ss} - \boldsymbol{L}^\dagger_{st} - \boldsymbol{L}^\dagger_{ts} + \boldsymbol{L}^\dagger_{tt}, \tag{7}$$

*where $\boldsymbol{e}_s$ is a one-hot vector with a 1 in the $s$-th position, and $\boldsymbol{L}^\dagger$ is the Moore-Penrose pseudo-inverse of the graph Laplacian matrix $\boldsymbol{L} := \boldsymbol{D} - \boldsymbol{A}$, satisfying $\boldsymbol{L}\boldsymbol{L}^\dagger = \boldsymbol{\Pi}$ and $\operatorname{span}(\boldsymbol{L}^\dagger) = \operatorname{span}(\boldsymbol{L}) = \{\boldsymbol{v} \in \mathbb{R}^n : \boldsymbol{v}^\top \mathbf{1} = 0\}$. Here, $\boldsymbol{\Pi} = \boldsymbol{I}_n - \frac{1}{n}\mathbf{1}\mathbf{1}^\top$ is the projection matrix onto $\operatorname{span}(\boldsymbol{L})$.*

As shown by Klein & Randić (1993), $R(s,t)$ is a valid distance metric on graphs. Additionally, we present the following lemma, which connects resistance distance to spanning trees:

**Lemma 2** ((Lovász, 1993; Hayashi et al., 2016))**.** *Given an edge $(s,t)$ in an unweighted undirected graph $G$, we have*

$$R(s,t) = \Pr_{T\sim\mu_G}\left(\mathbb{I}[(s,t)\in E(T)]\right),$$

*where $T$ is a spanning tree sampled from the uniform distribution of spanning trees of $G$, denoted by $\mu_G$, and $\mathbb{I}[\cdot]$ is the indicator function.*

Next, we define the approximate computation of APRD:

**Definition 5** (Approximate Computation of APRD). *Given an undirected, unweighted graph $G = (V, E)$, an error threshold $\epsilon > 0$, and a failure probability $0 \leq p_f \leq 1$, compute a matrix $\boldsymbol{R} \in \mathbb{R}^{n \times n}$ such that for any node pair $u, v$,*

$$\Pr\left(|\boldsymbol{R}_{uv} - R(u,v)| > \epsilon R(u,v)\right) \leq p_f.$$

To the best of our knowledge, the fastest algorithm for approximating APRD can be formally stated as follows:

**Lemma 3** ((Dwaraknath et al., 2023)). *The approximate computation of APRD for a graph with $n$ nodes and $m$ edges can be achieved with a time complexity of*

$$\tilde{O}\left(\min\left(nm, n^{\omega}, \frac{m}{\epsilon}\kappa(\boldsymbol{D}^{-1/2}\boldsymbol{L}\boldsymbol{D}^{-1/2}) + n^2\right)\right),$$

*where $\omega < 2.372$ is the matrix multiplication exponent, and $\kappa$ denotes the condition number of the matrix.*

Note that the $\tilde{O}(nm)$ time complexity is achieved using near-linear time Laplacian solvers, as proposed in a series of works (Spielman & Teng, 2004; Koutis et al., 2010; 2011; Cohen et al., 2014; Jambulapati & Sidford, 2021), while the $\tilde{O}(n^{\omega})$ complexity comes from fast matrix multiplication techniques. However, under the $\Theta\left(\frac{1}{n}\right)$ error requirement, as discussed in Section 3, the time complexity degenerates to $\tilde{O}\left(\min\left(nm, n^{\omega}\right)\right)$, which matches the time complexity of APSP.

## F Proof of Theorem 1

**Theorem 1.** *For any integer $n \geq 3$, there exist two graphs with $n$ nodes, $P_n$ and $C_n$, such that for any two adjacent nodes in $P_n$, the resistance distance is $1$, while in $C_n$, it is $1 - 1/n$. Furthermore, $P_n$ is neither vertex- nor edge-biconnected, whereas $C_n$ is both vertex- and edge-biconnected.*

*Proof.* Let $P_n$ be the path graph with $n$ nodes, and $C_n$ be the cycle graph with $n$ nodes.

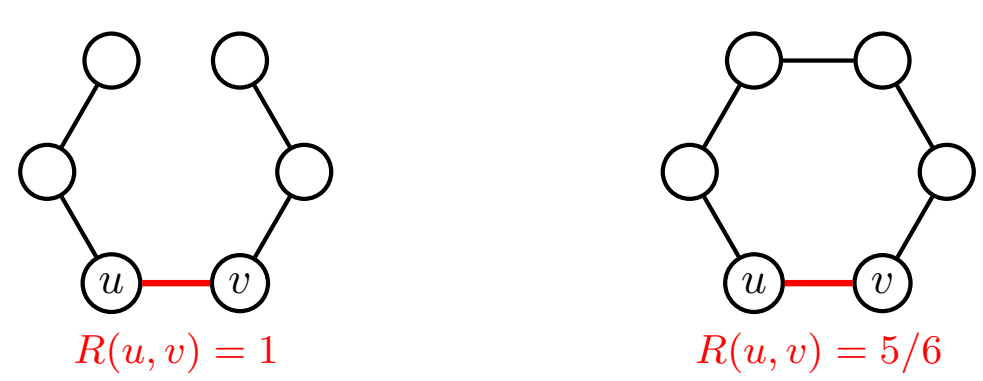

Figure 4: $P_6$ and $C_6$.

We observe that $P_n$ has exactly one spanning tree, which is the graph itself. In contrast, $C_n$ has $n$ spanning trees, each formed by removing a single edge from $E(C_n)$. Therefore, for each edge $(u, v) \in E(P_n)$, we have

$$\Pr_{T \sim \mu_{P_n}}\left(\mathbb{I}[(u,v) \in E(T)]\right) = 1,$$

and for each edge $(u, v) \in E(C_n)$,

$$\Pr_{T \sim \mu_{C_n}}\left(\mathbb{I}[(u,v) \in E(T)]\right) = \frac{n-1}{n} = 1 - \frac{1}{n}.$$

Thus, by applying Lemma 2, we arrive at the desired conclusion: for $(u, v) \in E(P_n)$, $R(u, v) = 1$, while for $(u, v) \in E(C_n)$, $R(u, v) = 1 - \frac{1}{n}$. The biconnectivity of these two graph types is evident. □

## G Proof of Theorem 2

**Theorem 2.** *Given any edge $(u, v) \in E$ in an undirected, unweighted graph $G$, $(u, v)$ is a cut edge if and only if the resistance distance $R(u, v) = 1$.*

*Proof.* It is straightforward to show that an edge $(u, v)$ is a cut edge if and only if it is included in every spanning tree of $G$. Therefore, by Lemma 2, we have

$$R(u, v) = \Pr_{T \sim \mu_G} (\mathbb{I}[(u, v) \in E(T)]) = 1.$$

□

## H Proof of Theorem 4

Before proving Theorem 4, we present some basic facts about the CONGEST model. First, we show that a spanning tree rooted at a node $u$ can be constructed using the Flood algorithm.

**Lemma 4** ((Peleg, 2000), Flood Algorithm)**.** *There exists a CONGEST algorithm in which a designated node $u \in V$ can construct a spanning tree $T$ rooted at $u$ with depth $\mathrm{depth}(T) = \max_v d_G(u, v)$ in $\max_v d_G(u, v) = O(D)$ rounds, where $D$ is the diameter of the graph.*

The idea behind the Flood algorithm is straightforward: Initially, the source node $u$ sends a special token to all its neighbors. Each node, upon receiving the token for the first time, stores it and forwards it to its neighbors. If a node receives the token again, it discards it and does nothing.

Additionally, we include the following lemmas, which describe the ability to broadcast and collect messages to and from a designated node.

**Lemma 5** ((Peleg, 2000), Downcast Algorithm)**.** *There exists a CONGEST algorithm in which, given $M$ messages (of $\Theta(\log n)$ bit) stored at a designated node $u \in V$, and a spanning tree $T$ rooted at $u$, the messages can be broadcast to other nodes in $O(\mathrm{depth}(T) + M)$ rounds.*

**Lemma 6** ((Peleg, 2000), Upcast Algorithm)**.** *There exists a CONGEST algorithm in which, given $M$ messages stored at different nodes and a spanning tree $T$ rooted at $u$, the messages can be collected at node $u$ in $O(\mathrm{depth}(T) + M)$ rounds.*

It is important to note that the conclusions for the Downcast and Upcast algorithms above are derived under the standard CONGEST model, where each edge can transmit only $O(1)$ messages of size $\Theta(\log n)$ bits per communication round. If we relax this restriction to allow the transmission of $w$ messages of size $\Theta(\log n)$ bits per round, the round complexities of the two algorithms reduce to $O\left(\mathrm{depth}(T) + \frac{M}{w}\right)$ by grouping messages together.

With these tools in hand, we are now ready to prove the theorem.

**Theorem 4.** *If we allow a single node to have unbounded computational power to solve any computable problem, then every* NP-Complete *decision problems on undirected unweighted graphs can be solved by the CONGEST model in $O(m)$ rounds.*

*Proof.* Given an NP-Complete problem on an undirected, unweighted graph, such as deciding whether there is a $k$-clique in a graph $G$, we proceed as follows. Since each node is assigned a unique ID in $[n]$, we designate node 0 as the leader without loss of generality.

First, in one round, each node $u$ collects the IDs of its neighbors and forms $d(u)$ messages of the form $(\mathrm{ID}(u), \mathrm{ID}(v))$ for each $v \in N(u)$. Next, we invoke the Flood algorithm to construct a spanning tree rooted at node 0 in $O(D)$ rounds. Afterward, we apply the Upcast algorithm to gather a total of $\sum_{u \in V} d(u) = \Theta(m)$ messages in $O(D + m)$ rounds. At this point, node 0 has complete knowledge of the graph's topology. Since node 0 can solve any computable problem, it can solve the NP-Complete problem locally in one round. Finally, node 0 uses the Downcast algorithm to broadcast the result to all other nodes in $O(D)$ rounds.

Thus, the total number of communication rounds is

$$1 + O(D) + O(D + m) + O(D) = O(m).$$

□

# I THE WEISFEILER-LEHMAN PROBLEM AND ITS HARDNESS (THEOREM 5)

In this section, we formally define the meaning of one iteration of the Weisfeiler-Lehman (WL) test and prove its computational hardness. To facilitate our presentation, we first define the WL relation.

**Definition 6** (Weisfeiler-Lehman Relation). *Given an unweighted, undirected graph $G$, the Weisfeiler-Lehman relation on $G$, denoted by* $\mathrm{WL}(G)$*, is the set of color vector pairs $(\boldsymbol{x}, \boldsymbol{y}) \in [n]^n \times [n]^{n6}$ satisfying:*

$$\forall u \in V, v \in V, \boldsymbol{y}_u = \boldsymbol{y}_v \Leftrightarrow \boldsymbol{x}_u = \boldsymbol{x}_v \wedge \{\!\{\boldsymbol{x}_z : z \in N(u)\}\!\} = \{\!\{\boldsymbol{x}_z : z \in N(v)\}\!\}.$$

The formal definition of one iteration of the WL test is captured by the deterministic, zero-error randomized, and bounded-error randomized WL problems, as described below.

**Definition 7** (Weisfeiler-Lehman Problem). *Given an unweighted, undirected graph $G$ with $n$ nodes, where each node $v$ is initially assigned a color $\boldsymbol{x}_v \in [n]$, the goal is to assign a new color $\boldsymbol{y}_v \in [n]$ for each node such that:*

- ***Deterministic:*** $(\boldsymbol{x}, \boldsymbol{y}) \in \mathrm{WL}(G)$*, or*
- ***Randomized, Zero-Error:*** $\Pr\left((\boldsymbol{x}, \boldsymbol{y}) \in \mathrm{WL}(G)\right) = 1$*, or*
- ***Randomized, Bounded-Error:*** $\Pr\left((\boldsymbol{x}, \boldsymbol{y}) \in \mathrm{WL}(G)\right) \geq 1 - \epsilon$.

We now formally state and prove theorems regarding the hardness of one iteration of the WL test.

**Theorem 5.** *If an RL-CONGEST model can simulate **one** iteration of the WL test without preprocessing, either deterministically or randomly with zero error, regardless of the computational power available to each node, the model's width $w$ and depth $d$ must satisfy $d = \Omega\left(D + \frac{m}{w \log n}\right)$, given that $w = o\left(\frac{n}{\log n}\right)$.*

The proof of the theorem relies on tools from communication complexity, so we recommend that readers refer to Appendix B for a basic understanding of these concepts.

*Proof.* We will prove that for any positive integer $n$ and any positive integer $m$ such that $m \in [n, n^2]$, there exists a hard-case graph with $\Theta(n)$ nodes and $\Theta(m)$ edges. Given $n$ and $m$, we first construct an incomplete "basic" graph $G_{(n,m)}$ with $\Theta(n)$ nodes and $\Theta(n)$ edges, partitioned between Alice (A) and Bob (B), as follows:

- Alice and Bob each have nodes $x^{(A)}$ and $x^{(B)}$, connected by an edge;
- They also hold nodes $w_i^{(A)}$ and $w_i^{(B)}$ ($i = 1, 2, \cdots, \left\lceil \frac{m}{n} \right\rceil$), connected to $x^{(A)}$ and $x^{(B)}$ respectively;
- Additionally, they possess nodes $u_i^{(A)}$, $u_i^{(B)}$, $v_i^{(A)}$, and $v_i^{(B)}$ ($i = 1, 2, \cdots, n$);
- Add edges $\left\{\left(u_n^{(A)}, v_n^{(A)}\right)\right\} \cup \left\{\left(u_i^{(A)}, u_{i+1}^{(A)}\right) : i \in \{1, 2, \cdots, n-1\}\right\} \cup \left\{\left(v_i^{(A)}, v_{i+1}^{(A)}\right) : i \in \{1, 2, \cdots, n-1\}\right\}$ to form a path with nodes $u_i^{(A)}$, and $v_i^{(A)}$. Repeat similarly for nodes $u_i^{(B)}$, and $v_i^{(B)}$, to form another path.

Then, we assign each node's color $\boldsymbol{x}_u$ as follows:

- $\boldsymbol{x}_{x^{(A)}} = \boldsymbol{x}_{x^{(B)}} = 0$;

[6] Our theorems also hold for any bounded color spaces, such as $[p(n)]$ for any polynomial $p$. However, since $n$ colors are always sufficient, we can simply set the color space to $[n]$.

- For each $i \in \{1, 2, \cdots, n\}$, $\boldsymbol{x}_{u_i^{(A)}} = \boldsymbol{x}_{u_i^{(B)}} = i$;
- For each $i \in \{1, 2, \cdots, n\}$, $\boldsymbol{x}_{v_i^{(A)}} = \boldsymbol{x}_{v_i^{(B)}} = n + i$;
- For each $i \in \left\{1, 2, \cdots, \left\lceil \frac{m}{n} \right\rceil\right\}$, $\boldsymbol{x}_{w_i^{(A)}} = \boldsymbol{x}_{w_i^{(B)}} = 2n + i$;

The constructed basic graph is illustrated in Figure 5.

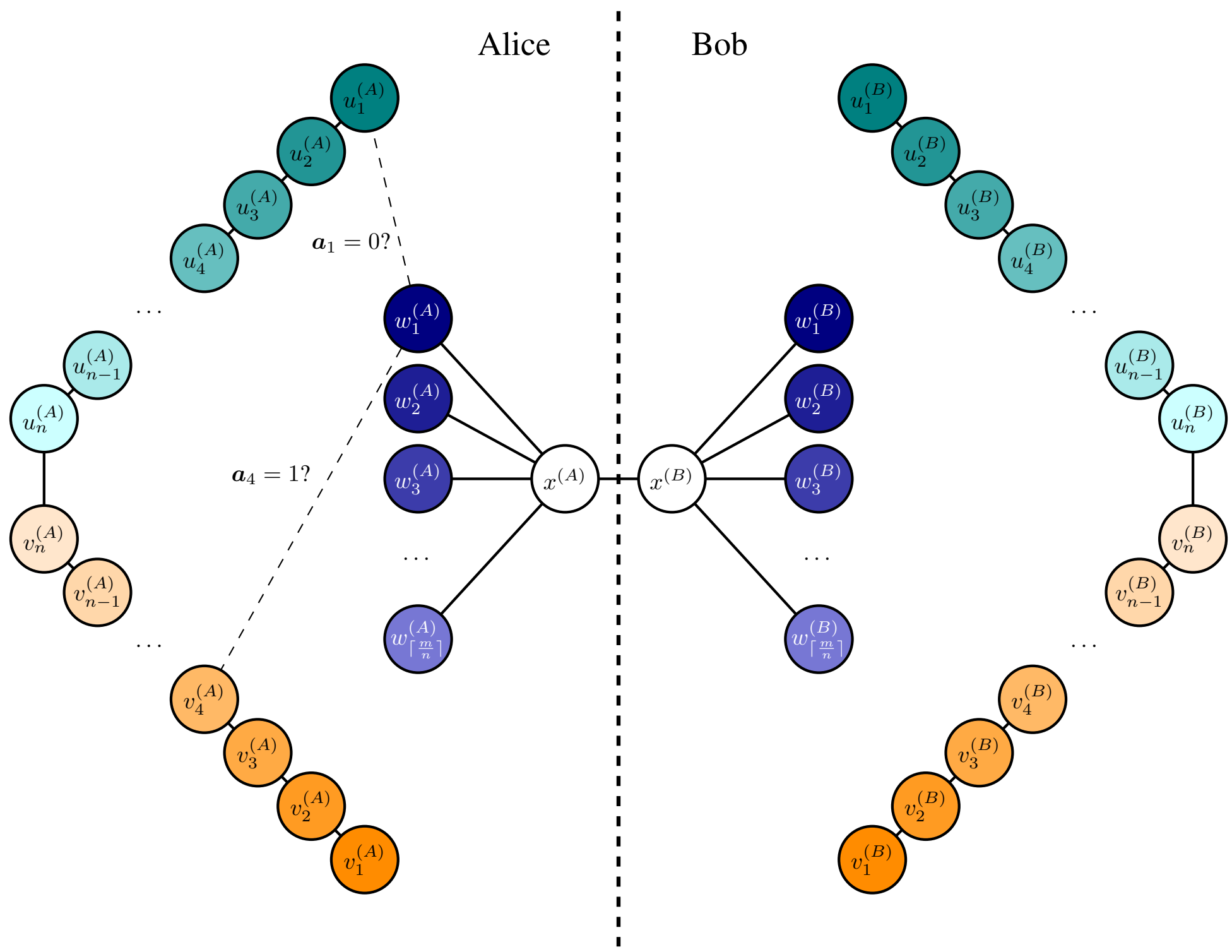

Figure 5: The constructed basic graph $G_{(n,m)}$. Nodes are colored according to $\boldsymbol{x}$.

Alice and Bob also fix a bijection $c$ in advance between the set of index pairs $\left\{1, 2, \cdots, \left\lceil \frac{m}{n} \right\rceil\right\} \times \{1, 2, \cdots, n\}$ and the set $\left\{1, 2, \cdots, n \cdot \left\lceil \frac{m}{n} \right\rceil\right\}$[7]. For example, define $c((i, j)) = (i-1)n + j$ and $c^{-1}(i) = \left(\left\lceil \frac{i}{n} \right\rceil, (i-1) \bmod n + 1\right)$.

Given an instance $(\boldsymbol{a}, \boldsymbol{b}) \in \{0,1\}^m \times \{0,1\}^m$ of $\mathsf{EQ}_m$, Alice receives $\boldsymbol{a} = (\boldsymbol{a}_1, \boldsymbol{a}_2, \cdots, \boldsymbol{a}_m)$ and Bob receives $\boldsymbol{b} = (\boldsymbol{b}_1, \boldsymbol{b}_2, \cdots, \boldsymbol{b}_m)$, they complete $G_{(n,m)}$ to $G_{(n,m);(\boldsymbol{a},\boldsymbol{b})}$ with $\Theta(n)$ nodes and $\Theta(m)$ edges as follows:

- For each $k \in \{1, 2, \cdots, m\}$, let $(i, j) = c^{-1}(k)$. If $\boldsymbol{a}_k = 0$, Alice adds edge $(w_i^{(A)}, u_j^{(A)})$; otherwise, Alice adds edge $(w_i^{(A)}, v_j^{(A)})$;
- For each $k \in \{1, 2, \cdots, m\}$, let $(i, j) = c^{-1}(k)$. If $\boldsymbol{b}_k = 0$, Bob adds edge $(w_i^{(B)}, u_j^{(B)})$; otherwise, Bob adds edge $(w_i^{(B)}, v_j^{(B)})$;

[7] Since $\frac{m}{n} \le \left\lceil \frac{m}{n} \right\rceil < \frac{m}{n} + 1$, we have $m \le n \left\lceil \frac{m}{n} \right\rceil < m + n$.

We claim that given $G_{(n,m);(\boldsymbol{a},\boldsymbol{b})}$, for any new color vector $\boldsymbol{y}$ such that $(\boldsymbol{x},\boldsymbol{y}) \in \mathrm{WL}(G_{(n,m);(\boldsymbol{a},\boldsymbol{b})})$, we have:

$$\boldsymbol{a} = \boldsymbol{b} \Longleftrightarrow \forall i \in \left\{1, 2, \cdots, \left\lceil \frac{m}{n} \right\rceil \right\}, \boldsymbol{y}_{w_i^{(A)}} = \boldsymbol{y}_{w_i^{(B)}}.$$

- "$\Rightarrow$" is straightforward. Since the construction is symmetric, for any $i \in \left\{1, \cdots, \left\lceil \frac{m}{n} \right\rceil \right\}$ we have $\left\{\!\!\left\{ \boldsymbol{x}_k : k \in N\left(w_i^{(A)}\right) \right\}\!\!\right\} = \left\{\!\!\left\{ \boldsymbol{x}_k : k \in N\left(w_i^{(B)}\right) \right\}\!\!\right\}$ and thus
$$\left(\boldsymbol{x}_{w_i^{(A)}}, \left\{\!\!\left\{ \boldsymbol{x}_k : k \in N\left(w_i^{(A)}\right) \right\}\!\!\right\}\right) = \left(\boldsymbol{x}_{w_i^{(B)}}, \left\{\!\!\left\{ \boldsymbol{x}_k : k \in N\left(w_i^{(B)}\right) \right\}\!\!\right\}\right).$$
Therefore, $\forall i \in \left\{1, \cdots, \left\lceil \frac{m}{n} \right\rceil \right\}$, $\boldsymbol{y}_{w_i^{(A)}} = \boldsymbol{y}_{w_i^{(B)}}$.
- To prove the other direction, we show its contrapositive: $\boldsymbol{a} \neq \boldsymbol{b} \Rightarrow \exists i, \boldsymbol{y}_{w_i^{(A)}} \neq \boldsymbol{y}_{w_i^{(B)}}$. Since $\boldsymbol{a} \neq \boldsymbol{b}$, there exists $1 \leq k \leq m$ such that $\boldsymbol{a}_k \neq \boldsymbol{b}_k$. Without loss of generality, assume $\boldsymbol{a}_k = 0$ and $\boldsymbol{b}_k = 1$, and let $(i,j) = c^{-1}(k)$. According to our construction, for Alice, $j \in \left\{\!\!\left\{ \boldsymbol{x}_v : v \in N\left(w_i^{(A)}\right) \right\}\!\!\right\}$ and $n + j \notin \left\{\!\!\left\{ \boldsymbol{x}_v : v \in N\left(w_i^{(A)}\right) \right\}\!\!\right\}$, while for Bob, $j \notin \left\{\!\!\left\{ \boldsymbol{x}_v : v \in N\left(w_i^{(B)}\right) \right\}\!\!\right\}$ and $n + j \in \left\{\!\!\left\{ \boldsymbol{x}_v : v \in N\left(w_i^{(B)}\right) \right\}\!\!\right\}$. Therefore, $\boldsymbol{y}_{w_i^{(A)}} \neq \boldsymbol{y}_{w_i^{(B)}}$.

Now, assume that $\mathrm{WL}(G_{(n,m);(\boldsymbol{a},\boldsymbol{b})})$ can be solved by an RL-CONGEST model on $G_{(n,m);(\boldsymbol{a},\boldsymbol{b})}$ deterministically, or randomly with zero error, in $d$ rounds. We can construct an algorithm that solves $\mathsf{EQ}_m$ in no more than $d + 1 + \left\lceil \frac{m}{n} \right\rceil$ rounds. This can be done by first using $d$ rounds to compute $\boldsymbol{y}$, then using 1 round for $x^{(A)}$ and $x^{(B)}$ to collect the colors of their neighbors, and finally using $\left\lceil \frac{m}{n} \right\rceil$ rounds to compare the colors between $w_i^{(A)}$ and $w_i^{(B)}$.

According to the results in communication complexity, we have $D(\mathsf{EQ}_m) = R_0(\mathsf{EQ}_m) = \Theta(m)$, which implies that the total number of communicated bits satisfies

$$\left(d + 1 + \left\lceil \frac{m}{n} \right\rceil\right) w \log |V(G_{(n,m);(\boldsymbol{a},\boldsymbol{b})})| = \Omega(m).$$

This is equivalent to

$$d + \Theta\left(\frac{m}{n}\right) = \Omega\left(\frac{m}{w \log n}\right).$$

Therefore, when $w = o\left(\frac{n}{\log n}\right)$, we have $d = \Omega\left(\frac{m}{w \log n}\right)$.

The $\Omega(D)$ component is straightforward. By adding a path of length $\Theta(D)$ between $x^{(A)}$ and $x^{(B)}$, any message exchange between the two parties will require $\Omega(D)$ rounds. □

## J Proof of Theorem 6

**Theorem 6.** *There exists a deterministic RL-CONGEST algorithm that can simulate **one** iteration of the WL test without preprocessing, with width $w$ and depth $d$ satisfying $d = O\left(D + \frac{m}{w}\right)$. Additionally, it is sufficient to set the nodes' computational resource class to $\mathsf{C} = \mathsf{DTIME}(n^2 \log n)$.*

*Proof.* We present the framework of our algorithm and analyze the round complexity for each step:

1. Each node $u$ sends a message $(u, \boldsymbol{x}_u)$ to its neighbors and receives messages from them to form the set $S_u = \{(u, v, \boldsymbol{x}_v) : v \in N(u) \cup u\}$. This process takes $O(1)$ rounds.
2. Node 0 initiates the Flood algorithm to construct a BFS spanning tree rooted at node 0. This process takes $O(D)$ rounds.

3. The UPCAST algorithm is used to collect all sets $S_u$ at the root node 0 along the spanning tree. This process takes $O\left(D + \frac{m}{w}\right)$ rounds, as there are $\sum_{u \in V} O(d_u) = O(m)$ messages to gather, and each edge can transmit $w$ messages per round.

4. Node 0 merges all $S_u$ to form the set $K = \{(u, (\boldsymbol{x}_u, \{\!\{\boldsymbol{x}_v : v \in N(u)\}\!\})) : u \in V\}$. This step can be completed in one round and requires $O(m)$ time[8], since there are $O(m)$ tuples in $\bigcup_u S_u$.

5. Node 0 sorts $K$ by $(\boldsymbol{x}_u, \{\!\{\boldsymbol{x}_v : v \in N(u)\}\!\})$ to create the ordered set $K' = \{((\boldsymbol{x}_u, \{\!\{\boldsymbol{x}_v : v \in N(u)\}\!\}), u) : u \in V\}$. It then determines new colors for each node by the rank of $(\boldsymbol{x}_u, \{\!\{\boldsymbol{x}_v : v \in N(u)\}\!\})$. The computed color mapping is represented as $\{(u, \boldsymbol{y}_u) : u \in V\}$. This process can be done in one round and requires $O(n \log n)$ comparisons, with each comparison taking $O(\Delta)$ time, resulting in $O(n\Delta \log n)$ time.

6. The DOWNCAST algorithm is used to send the results back to each node along the spanning tree. This process takes $O\left(D + \frac{n}{w}\right)$ rounds, as there are $O(n)$ messages to transmit.

Thus, the WL problem can be computed by the RL-CONGEST model in

$$d = O\left(1 + D + D + \frac{m}{w} + D + \frac{n}{w}\right) = O\left(D + \frac{m}{w}\right)$$

rounds. The computational resource class of each node is determined by steps 4 and 5. □

# K PROOF OF THEOREM 7

**Theorem 7.** *There exists a deterministic RL-CONGEST algorithm that can simulate* ***one*** *iteration of the WL test by adding a virtual node, which connects to other nodes, as preprocessing. The algorithm operates with width $w$ and depth $d$ satisfying $dw = O(\Delta)$, where $\Delta$ is the maximum degree of the graph before the addition of the virtual node. Additionally, it is sufficient to set the nodes' computational resource class to $\mathsf{C} = \mathsf{DTIME}(n^2 \log n)$.*

*Proof.* Without loss of generality, we denote the added virtual node as node $n$, which is known to all nodes in the original graph. We then outline the framework of our algorithm and analyze the round complexity for each step:

1. Each node $u$, except the virtual node, sends a message $(u, \boldsymbol{x}_u)$ to its neighbors and receives messages from them, forming the set $S_u = \{(u, v, \boldsymbol{x}_v) : v \in N(u) \cup u\}$. This process takes $O(1)$ rounds.

2. The virtual node $n$ sends a token to each node to notify them of the edge along which the virtual node can be reached. This process takes 1 round.

3. Each node $u$, except the virtual node, sends its set $S_u$ to the virtual node $n$ along the edge connecting to it. This process takes $O\left(\frac{\Delta}{w}\right)$ rounds, as each node has at most $\Theta(\Delta)$ messages to send, excluding the virtual node.

4. The virtual node $n$ merges all sets $S_u$ to form the set $K = \{(u, (\boldsymbol{x}_u, \{\!\{\boldsymbol{x}_v : v \in N(u)\}\!\})) : u \in V\}$. This step can be completed in 1 round and requires $O(m)$ time[9], as there are $O(m)$ tuples in $\bigcup_u S_u$.

5. The virtual node $n$ sorts $K$ by $(\boldsymbol{x}_u, \{\!\{\boldsymbol{x}_v : v \in N(u)\}\!\})$ to create the ordered set $K' = \{((\boldsymbol{x}_u, \{\!\{\boldsymbol{x}_v : v \in N(u)\}\!\}), u) : u \in V\}$. It then assigns new colors to each node based on the rank of $(\boldsymbol{x}_u, \{\!\{\boldsymbol{x}_v : v \in N(u)\}\!\})$. The color mapping is represented as $\{(u, \boldsymbol{y}_u) : u \in V\}$. This process can be completed in 1 round and requires $O(n \log n)$ comparisons, with each comparison taking $O(\Delta)$ time, resulting in $O(n\Delta \log n)$ total time.

[8]Assuming a WordRAM machine where each word consists of $\Theta(\log n)$ bits.
[9]Assuming a WordRAM model where each word consists of $\Theta(\log n)$ bits.

6. The virtual node $n$ sends the new colors $\boldsymbol{y}_u$ to their corresponding nodes $u$ along the connecting edges. This process takes 1 round, as each edge transmits only one message.

Thus, the WL problem can be computed in the RL-CONGEST model with the addition of a virtual node in

$$d = O\left(1 + 1 + \frac{\Delta}{w} + 1\right) = O\left(\frac{\Delta}{w}\right)$$

rounds, or equivalently, $dw = O(\Delta)$. The computational resource class of each node is determined by steps 4 and 5. □

# L Complexity of the PNF Model Checking Problem

As mentioned in Section 4.3, the $\mathcal{L}^k$ model checking problem is crucial in various areas of computer science. We will introduce some key results that characterize the complexity of its special case, the PNF $\mathcal{L}^k$ model checking problem.

On one hand, the first two results establish the upper bound of the time complexity for the PNF $\mathcal{L}^k$ model checking problem **on graphs**, where the predicates are limited to $E(x, y)$ and $=(x, y)$.

**Theorem 9** ((Eisenbrand & Grandoni, 2004; Williams, 2014))**.** *For $k \geq 3$, every $k$-quantifier (PNF) first-order sentence on $n$-node graphs can be decided in $\tilde{O}(n^{k-3+\omega})$ time.*

**Theorem 10** ((Eisenbrand & Grandoni, 2004; Williams, 2014))**.** *For $k \geq 9$, every $k$-quantifier (PNF) first-order sentence on $n$-node graphs can be decided in $n^{k-1+o(1)}$ time.*

On the other hand, the following result establishes a lower bound for the PNF $\mathcal{L}^k$ model checking problem on graphs, conditioned on a well-known hypothesis in theoretical computer science:

**Theorem 11** ((Puatracscu & Williams, 2010; Williams, 2014))**.** *For $k \geq 4$, if the model checking problem for $k$-quantifier (PNF) first-order sentences over graphs can be solved in $O(n^{k-1-\epsilon})$ time for some $\epsilon > 0$, then the Strong Exponential Time Hypothesis (SETH) is false.*

Note that the algorithms in Theorem 9 and 10 leverage fast matrix multiplication techniques to improve the brute-force algorithm by a factor of $n$, in contrast to the combinatorial operations typically employed in GNNs. Motivated by this, many works have attempted to find a faster combinatorial algorithm that does not rely on fast matrix multiplication and can outperform the brute-force approach by a polynomial factor of $n^\epsilon$ for some $\epsilon > 0$, but to date, all such attempts have failed. This has led to the formulation of the combinatorial $k$-clique conjecture, which is stated as follows:

**Conjecture 1** (Combinatorial $k$-Clique Conjecture, (Abboud et al., 2017; Bringmann et al., 2017; Abboud et al., 2024))**.** *For any $k \geq 3$ and any $\epsilon > 0$, no combinatorial algorithm can determine whether a graph contains a $k$-clique in $O(n^{k-\epsilon})$ time.*

Since the $k$-clique problem is a special case of the PNF $\mathcal{L}^k$ model checking problem, it follows that no combinatorial algorithm can solve the PNF $\mathcal{L}^k$ model checking problem in $O(n^{k-\epsilon})$ time for $k \geq 3$. Therefore, when focusing exclusively on combinatorial algorithms, it is reasonable to classify the PNF $\mathcal{L}^k$ model checking problem in the $\tilde{\Theta}(n^k)$ complexity class.

For more general cases, such as PNF $\mathcal{L}^k$ model checking not limited to graphs, there are also results concerning both upper and lower bounds. First, Gao et al. (2017) designed an algorithm to solve the PNF $\mathcal{L}^k$ model checking problem in $m^{k-1-o(1)}$ time, where $m$ is the size of the input structure, i.e., the number of all tuples in the relations, which is equivalent to the number of edges when the model is a graph.

**Theorem 12** ((Gao et al., 2017))**.** *There exists an algorithm that solves the (PNF) $\mathcal{L}^k$ model checking problem in time $m^{k-1}/2^{\Theta\left(\sqrt{\log m}\right)}$.*

They also prove the near-optimality of their algorithm under SETH:

**Theorem 13** ((Gao et al., 2017))**.** *Assuming SETH, no algorithm can solve the (PNF) $\mathcal{L}^k$ model checking problem in $O(m^{k-1-\epsilon})$ time for any $\epsilon > 0$.*

Therefore, it is reasonable to classify the PNF $\mathcal{L}^k$ model checking problem in the $\tilde{\Theta}\left(m^{k-1}\right)$ complexity class. By combining the two results, it is reasonable to classify the PNF $\mathcal{L}^k$ model checking problem in the $\tilde{\Theta}\left(\min\left\{n^k, m^{k-1}\right\}\right)$ complexity class.

# M PROOF OF THEOREM 8

**Theorem 8.** *If constructing the $k$-WL graph and additional features in $O\left(k \cdot \text{poly}|\phi| \cdot n^{k+1}\right)$ time is allowed, where $|\phi|$ represents the length of the quantifier-free formula, the RL-CONGEST model can solve the PNF $\mathcal{C}^k$ model checking problem in $O(k^2)$ rounds. Additionally, the computational resources required by each node are $\mathsf{C} = \mathsf{DTIME}(k^2 n)$.*

*Proof.* We prove this theorem by directly constructing a solution under the RL-CONGEST framework to solve the PNF $\mathcal{C}^k$ model checking problem. This involves constructing the $k$-WL graph and computing features as part of the preprocessing, followed by a message-passing process.

Given an input graph $G$ and a PNF $\mathcal{C}^k$ sentence $\varphi := (Q_1 u_1)(Q_2 u_2)\cdots(Q_k u_k)\phi(u_1, u_2, \cdots, u_k)$, where $Q_i$ are quantifiers from $\{\forall, \exists, \exists^{\geq i}, \exists^{\leq i}, \exists^{=i}\}$, and $\phi(u_1, \cdots, u_k)$ is a quantifier-free formula[10] with predicates limited to $E(\cdot,\cdot)$ and $=(\cdot,\cdot)$, the overall framework using the RL-CONGEST model to decide whether $G \models \varphi$ proceeds as follows:

1. **Preprocessing:**
   - **Construct the $k$-WL graph $G'$:** Enumerate all $k$-tuples $\boldsymbol{v} = (v_1, \cdots, v_k) \in V^k$ to form the new node set $V'$ for the $k$-WL graph. Each node has a unique ID $(\text{ID}(v_1), \cdots, \text{ID}(v_k))$ in $[n]^k$. For simplicity, we use $v_i$ and $\text{ID}(v_i)$ interchangeably. Then, connect nodes with Hamming distance one, that is, $E' = \{(\boldsymbol{u}, \boldsymbol{v}) \in V' \times V' : d_H(\boldsymbol{u}, \boldsymbol{v}) = 1\}$.
   - **Construct a circuit for $\phi(u_1, \cdots, u_k)$:** We define a bijection $f$ between $\{1, 2, \cdots, k\} \times \{1, 2, \cdots, k\}$ and $\{1, 2, \cdots, k^2\}$ such that $f(i,j) = (i-1)k + j$. Next, we introduce $2k^2$ auxiliary variables corresponding to $E(u_i, u_j)$ and $=(u_i, u_j)$:
     - For each pair $(i,j) \in \{1, \cdots, k\} \times \{1, \cdots, k\}$, let variable $x_{f(i,j)}$ represent $E(u_i, u_j)$.
     - For each pair $(i,j) \in \{1, \cdots, k\} \times \{1, \cdots, k\}$, let variable $x_{k^2+f(i,j)}$ represent $=(u_i, u_j)$.

     A Boolean circuit $C_\phi(x_1, \cdots, x_{k^2}, \cdots, x_{2k^2})$ is built, taking $2k^2$ inputs. This is done by replacing atomic formulas $E(u_i, u_j)$ and $=(u_i, u_j)$ in $\phi$ with $x_{f(i,j)}$ and $x_{k^2+f(i,j)}$, respectively, and adding logic gates to form $C_\phi$.
   - **Compute features $\phi(\boldsymbol{v})$:** For each node $\boldsymbol{v} = (v_1, \cdots, v_k)$, representing an assignment of the variables $(u_1, \cdots, u_k)$, we compute a feature vector $\boldsymbol{x}(\boldsymbol{v}) = (x_1(\boldsymbol{v}), \cdots, x_{2k^2}(\boldsymbol{v}))$, where:
     - For each pair $(i,j)$, $x_{f(i,j)}(\boldsymbol{v}) = \mathbb{I}[E(v_i, v_j)] \in \{0,1\}$, indicating whether $v_i$ and $v_j$ are connected.
     - Similarly, $x_{k^2+f(i,j)}(\boldsymbol{v}) = \mathbb{I}[=(v_i, v_j)] \in \{0,1\}$, indicating whether $v_i$ and $v_j$ are the same node in $G$.

     By applying the $C_\phi$ circuit with input $\boldsymbol{x}(\boldsymbol{v})$ to each vertex $\boldsymbol{v}$, we obtain a one-bit feature for each node. It can be easily verified that $C_\phi(\boldsymbol{x}(\boldsymbol{v})) = \phi(\boldsymbol{v})$ for any node $\boldsymbol{v} \in V'$.

   We illustrate this preprocessing process with an example. Let $G$ be the graph in Figure 6, $k = 3$, and $\phi(u_1, u_2, u_3) := E(u_1, u_2) \wedge E(u_2, u_3) \wedge E(u_3, u_1)$. The auxiliary variables $x_i$ and their corresponding atomic formulas are listed in Table 2. The circuit $C_\phi$ is given by $C_\phi(x_1, \cdots, x_{18}) = x_2 \wedge x_6 \wedge x_7$. For node $\boldsymbol{v} = (0,1,3) \in V'$, the values of each auxiliary variable on $\boldsymbol{v}$ are shown in the third column of Table 2. Applying the circuit to $\boldsymbol{x}((0,1,3))$, we obtain an output of 1.

   We then analyze the time complexity of the preprocessing phase. For the construction of the $k$-WL graph, since there are $\Theta(n^k)$ nodes and $\Theta\left(kn^{k+1}\right)$ edges, the construction takes $\Theta\left(kn^{k+1}\right)$

[10] We assume that the logical connectives are limited to $\wedge$, $\vee$, and $\neg$. If other connectives are present, they can be transformed into an equivalent formula using only these connectives in $\text{poly}|\phi|$ time, where $|\phi|$ is the length of the formula.

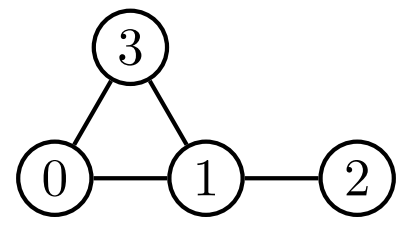

Figure 6: An example graph illustrating the preprocessing process.

Table 2: Auxiliary variables, their corresponding atomic formulas, and their values on node $(0, 1, 3)$.

| Auxiliary Variables | Corresponding Atomic Formulas | Values on Node $(0, 1, 3)$ |
|---|---|---|
| $x_1$ | $E(u_1, u_1)$ | $= [E(0, 0)] = 0$ |
| $x_2$ | $E(u_1, u_2)$ | $= [E(0, 1)] = 1$ |
| $x_3$ | $E(u_1, u_3)$ | $= [E(0, 3)] = 1$ |
| $x_4$ | $E(u_2, u_1)$ | $= [E(1, 0)] = 1$ |
| $x_5$ | $E(u_2, u_2)$ | $= [E(1, 1)] = 0$ |
| $x_6$ | $E(u_2, u_3)$ | $= [E(1, 3)] = 1$ |
| $x_7$ | $E(u_3, u_1)$ | $= [E(3, 0)] = 1$ |
| $x_8$ | $E(u_3, u_2)$ | $= [E(3, 1)] = 1$ |
| $x_9$ | $E(u_3, u_3)$ | $= [E(3, 3)] = 0$ |
| $x_{10}$ | $=(u_1, u_1)$ | $= [=(0, 0)] = 1$ |
| $x_{11}$ | $=(u_1, u_2)$ | $= [=(0, 1)] = 0$ |
| $x_{12}$ | $=(u_1, u_3)$ | $= [=(0, 3)] = 0$ |
| $x_{13}$ | $=(u_2, u_1)$ | $= [=(1, 0)] = 0$ |
| $x_{14}$ | $=(u_2, u_2)$ | $= [=(1, 1)] = 1$ |
| $x_{15}$ | $=(u_2, u_3)$ | $= [=(1, 3)] = 0$ |
| $x_{16}$ | $=(u_3, u_1)$ | $= [=(3, 0)] = 0$ |
| $x_{17}$ | $=(u_3, u_2)$ | $= [=(3, 1)] = 0$ |
| $x_{18}$ | $=(u_3, u_3)$ | $= [=(3, 3)] = 1$ |

time. The circuit construction involves simple operations, such as substituting variables, which takes $O(\text{poly}|\phi|)$ time. For the feature computation, we compute $\Theta(k^2)$ features for each node $\boldsymbol{v} \in V^k$ and pass it through the circuit $C_\phi$ to obtain $\phi(\boldsymbol{v})$. This process takes $O\left(\text{poly}|\phi| \cdot n^k\right)$ time. Thus, the total time complexity for the preprocessing phase is $O\left(\text{poly}|\phi| \cdot n^k + kn^{k+1}\right) = O\left(k \cdot \text{poly}|\phi| \cdot n^{k+1}\right)$.

2. **Message-Passing with Limited Computational Resources:** We present an $O(k^2)$-round message-passing algorithm on $G'$, where each node $\boldsymbol{v} \in V'$ has a feature $\phi(\boldsymbol{v})$ obtained during the preprocessing phase. The key idea is to treat the $k$-WL graph as an implicit $n$-ary prefix tree and perform dynamic programming to eliminate quantifiers over this tree. The algorithm is described as follows:

   (a) **Initial Setup:**
   - Each node $\boldsymbol{v} \in V'$ has a one-bit feature $\phi(\boldsymbol{v})$ from the preprocessing phase.
   - For each node $\boldsymbol{v} \in V'$:
     - Sends its ID along with its feature to its neighbors[11].
     - Receives messages from neighbors.

   (b) **Main Iteration ($\ell = k$ to $1$):**
   - We define the active nodes in the $\ell$-th iteration as those whose IDs take the form $(v_1, \cdots, v_{\ell-1}, 0, 0, \cdots)$, meaning the last $k - \ell + 1$ coordinates are zeros.
   - For each active node $(v_1, \cdots, v_{\ell-1}, 0, \cdots)$:
     - Collects all received features from nodes whose IDs match the pattern $(v_1, \cdots, v_\ell, 0, \cdots)$, where $v_\ell \in V$. This forms a multiset of $n$ bits.
     - Updates its feature based on the quantifier $Q_\ell$:
       * If $Q_\ell$ is $\forall$, set the node's feature to 1 if all $n$ bits in the multiset are 1; otherwise, set it to 0.

[11] This message passing takes $O(k)$ communication rounds.

    * If $Q_\ell$ is $\exists$, set the feature to $1$ if at least one $1$ exists in the multiset; otherwise, set it to $0$.
    * Similarly, handle other quantifiers ($\exists^{\geq i}$, $\exists^{\leq i}$, etc.).
  - Sends its updated feature and ID to its neighbors.
  - Receives messages from neighbors.
- Non-active nodes change their feature to $\perp$ and remain silent for the rest of the process.

(c) **Final Result:** The final result bit is found at the node $(0, 0, \cdots, 0) \in V'$. To propagate the result to all nodes in $V'$, we use an additional $k$ rounds.

Since there are $k$ iterations, and each iteration requires $O(k)$ communication rounds to send messages, the total number of communication rounds is $k \cdot O(k) + k = O(k^2)$. In each iteration, each node examines $O(kn)$ messages, each containing $O(k)$ words, resulting in $O(k^2 n)$ time complexity. Therefore, the computational complexity class for this RL-CONGEST algorithm is $\mathsf{C} = \mathsf{DTIME}(k^2 n)$.

□