# OpenReview forum: "Rethinking the Expressiveness of GNNs: A Computational Model Perspective"
_ICLR.cc/2025/Conference — Submitted to ICLR 2025_

### Official Review · Reviewer_b62D · 2024-10-26

**Soundness:** 3
**Presentation:** 4
**Contribution:** 2
**Rating:** 6
**Confidence:** 4

**Summary:**

In this paper, the authors first explain the limitations and unrealistic assumptions of several current approaches in analyzing the expressive power of GNNs, including underestimated preprocessing time, anonymous WL tests with non-anonymous features, and unrealistic assumptions in the CONGEST model. Next, the authors propose the RL-CONGEST model to address these issues. Several results are derived: (1) GNNs require substantial width and depth to simulate the WL test; (2) virtual nodes can help reduce computation costs, although they do not improve theoretical expressive power; (3) the RL-CONGEST model can solve the PNF model-checking problem with
$k$-WL graph transformation in $O(k^2)$ rounds.

**Strengths:**

1. The paper is well-structured and nicely presented.
2. The stated limitations of existing approaches make sense to me, and the examples are intuitive.
3. The new results derived by the RL-CONGEST model are interesting.

**Weaknesses:**

My main concern is about the practical implication of the proposed model beyond what the author presented.
1. One question is how we can use the RL-CONGEST model to effectively estimate and compare the representational power of different GNN variants or even predict their performance in real-world applications.
2. The authors claim that the proposed framework can be used for analyses involving non-anonymous node features. I wonder how this framework can be leveraged to truly evaluate differences between various added features, such as SPD or resistance distance. In my view, although the broken symmetry introduced by these additional features is undoubtedly a source of improved expressivity, different features have varying degrees of power; some can help count more complex graph structures than others.

**Questions:**

See above.

---

> ### Author Response · Authors · 2024-11-15
> **Initial Response to Reviewer b62D (1/2)**
>
> Dear Reviewer b62D,
>
> Thank you for taking the time to review our paper. We would like to address your concerns as follows:
>
> **W1**:
>
> Our primary goal is to reveal limitations in current analyses of GNNs' expressive power and to introduce a new analytical approach that addresses these issues, rather than to develop a specific GNN model with enhanced performance or expressiveness. Specifically, as demonstrated in Theorems 5-8, we leverage the RL-CONGEST framework to provide a more reasonable evaluation of GNNs' expressive power on simulating one iteration of the WL test. Additionally, in Section 5, we also propose open questions that may be investigated within the RL-CONGEST framework. It is important to note that our RL-CONGEST framework is designed to assess a model's expressive power in executing algorithmic tasks or achieving "algorithmic alignment", rather than to predict its quantitative performance on learning tasks such as node classification.
>
> **W2**:
>
> Yes, the second half of your question, "different features have varying degrees of power; some can help count more complex graph structures than others", precisely reflects what we aim to convey. Under the non-anonymous setting, CONGEST and MPGNNs can exhibit greater expressiveness than the anonymous WL test. Our main argument in Section 3.2 is that while existing works claim their models' expressiveness advantage by proving they can perform tasks beyond the WL test's scope, this approach is questionable. Equating anonymous WL with MPGNNs, as previous works have done, is not entirely reasonable, and consequently, concluding that MPGNNs are weak because the WL test is weak is also debatable. In fact, MPGNNs can perform certain algorithms (such as solving edge biconnectivity in $O(D)$ rounds within the CONGEST model [Pritchard, 2006]).
>
> Our logical flow is as follows:
> 1. Numerous studies claim that the vanilla WL test has limited expressive power --- a claim that we affirm, as discussed in Figure 2. However, the appropriateness of using the anonymous WL test to characterize MPGNNs is debatable, given that real-world graphs often contain rich features. Additionally, [Loukas, 2020] demonstrated that with unique IDs (and other assumptions), MPGNNs can perform a wide range of algorithmic tasks.
> 2. To address the "limited" expressiveness of MPGNNs (stemming from the limitation of the vanilla WL test), some works incorporate additional features (e.g., [Loukas, 2020]) to enhance the expressiveness of their proposed models. Nonetheless, as outlined in (1), the suitability of the anonymous WL test as a characterization for MPGNNs is questionable. Consequently, the practice in some studies of demonstrating the advantage of their model's expressiveness by proving it can perform algorithmic tasks beyond the WL test's capabilities may not be entirely valid. A more reasonable approach would be to compare these models with MPGNNs under a non-anonymous setting (as suggested in [Loukas, 2020]). Further, evidence from [Loukas, 2020; Suomela, 2013; den Berg et al., 2018; You et al., 2021; Abbound et al., 2021; Sato et al., 2021] suggests that the non-anonymous setting can enhance model expressiveness, highlighting a mismatch in works that argue for a "weak MPGNN" yet use additional features that break the anonymous setting in the WL test to improve expressiveness.
> 3. As you mentioned, "different features have varying degrees of power; some can help count more complex graph structures than others", our RL-CONGEST analysis framework can be applied in studies proposing new GNN variants that use additional features and claim the ability to perform certain algorithmic tasks, with the only requirement being a **clear specification of the preprocessing time** complexity of the features.
>
> Additionally, points (1) and (2) highlight the need to reconsider the validity of comparing a proposed model's expressiveness directly with the vanilla WL test. We hope this discussion encourages the community to more accurately assess existing results on GNNs' expressiveness.
>
> Thank you again for reviewing our paper, and we are looking forward to any further discussions with you.

---

> > ### Author Response · Authors · 2024-11-15
> > **Initial Response to Reviewer b62D (2/2)**
> >
> > **References**:
> >
> > [Pritchard, 2006] David Pritchard. An Optimal Distributed Edge-Biconnectivity Algorithm. arXiv 2006.
> >
> > [Loukas, 2020] Andreas Loukas. What Graph Neural Networks Cannot Learn: Depth vs Width. ICLR 2020.
> >
> > [Zhang et al., 2023] Bohang Zhang, Shengjie Luo, Liwei Wang, and Di He. Rethinking the Expressive Power of GNNs via Graph Biconnectivity. ICLR 2023.
> >
> > [Xu et al., 2019] Keyulu Xu*, Weihua Hu*, Jure Leskovec, Stefanie Jegelka. How Powerful are Graph Neural Networks? ICLR 2019.
> >
> > [Suomela, 2013] Jukka Suomela. Survey of Local Algorithms. ACM Computing Surveys (CSUR), 45(2):24, 2013.
> >
> > [den Berg et al., 2018] Rianne van den Berg, Thomas N Kipf, and Max Welling. Graph Convolutional Matrix Completion. KDD 2018.
> >
> > [You et al., 2021] Jiaxuan You, Jonathan M Gomes-Selman, Rex Ying, and Jure Leskovec. Identity-aware Graph Neural Networks. AAAI 2021.
> >
> > [Abbound et al., 2021] Ralph Abboud, Ismail Ilkan Ceylan, Martin Grohe, and Thomas Lukasiewicz. The Surprising Power of Graph Neural Networks with Random Node Initialization. IJCAI 2021.
> >
> > [Sato et al., 2021] Ryoma Sato, Makoto Yamada, and Hisashi Kashima. Random Features Strengthen Graph Neural Networks. SDM 2021.

---

> ### Comment · Reviewer_b62D · 2024-11-19
>
> Thank the authors for the detailed response to my concerns. I have some follow-up questions.
>
> **RL-CONGEST framework is designed to assess a model's expressive power in executing algorithmic tasks or achieving "algorithmic alignment"**
>
> I am a little confused about this. First, I don't find such a statement in the paper. I assumed that the authors were talking about the general expressivity of GNNs and their downstream performance.
>
> I quite agree with the authors for the arguments they made in the paper: (1) hidden precomputation time; (2) constrained analysis on anonymous WL test. However, I agree with them mainly from a more general perspective. If we confine the discussion to algorithmic tasks, things become different. First, most papers discuss the expressiveness of GNNs in the general setting: to improve the performance of GNNs on downstream tasks. However, downstream tasks not only contain algorithmic tasks. For example, GD-WL [1] forms the story from the bi-connectivity problem, which can be solved with less complexity than computing resistance distance. However, the GD-WL can be used to approximate many other graph properties or count important substructures [2, 3], which is crucial for downstream tasks and may not be done with an algorithm of less complexity.
>
> Back to my original question (W1), I think the authors did a great job of formulating all these conclusions in the paper and I do find many conclusions interesting and original. However, when I start to think about these conclusions from a broader perspective, for example, whether these conclusions can help me gain more insight into evaluating or comparing existing GNNs, or can help me design new expressive GNNs, these conclusions seem limited. That is said, the GNNs are eventually designed for solving real-world problems like node classification or graph classification, which is much more boring than algorithmic tasks.
>
> **anonymous WL vs MPNN + ID**
>
> I think the authors are totally right in the statement that: by equipping MPNN with non-anonymous node features, MPNN can solve many algorithmic tasks that previous literatures claim not. However, I think the discrepancy here is still the scope of the discussion. Most existing works use anonymous WL tests as a tool because they want to make sure the result expressive GNNs are still permutation invariant and equivariant. Without this assumption, the result GNNs cannot have good performance on **real-world tasks**. It's true that given a unique ID, MPNN can solve many algorithmic tasks, but it cannot transfer to real-world tasks. For example, [4] injects random features into MPNN to improve expressiveness but many follow-up experiments actually show it achieves bad performance in real-world tasks.
>
> Back to my original question, what I really want to ask is that: additional features can improve the expressive power of MPNN by (1) breaking symmetry and leveraging message passing to learn on that and enhance performance; (2) directly adding additional knowledge about graph structures. It the proposed framework quantitively or qualitatively analyze the portion of these two parts given node features? Or, whether the proposed model can be used to analyze the effect of node features in real-world datasets for the expressiveness of MPNN?
>
>
> I still hold a positive perspective on the paper. But the above concerns somehow prevents me from further increasing my score.
>
> ### Reference
> [1] Zhang, Bohang, et al, Rethinking the Expressive Power of gnns via Graph Biconnectivity, ICLR23.
>
> [2] Zhang, Bohang, et al, A Complete Expressiveness Hierarchy for Subgraph GNNs via Subgraph Weiisfeiler-Lehman Tests ICML23.
>
> [3] Zhang, Bohang, et al, Beyond Weisfeiler-Lehman: A Quantitative Framework for GNN Expressiveness ICLR24.
>
> [4] Sato, Ryoma, et al, Random Features Strengthen Graph Neural Networks, SDM21.

---

> > ### Author Response · Authors · 2024-11-21
> >
> > Dear Reviewer b62D,
> >
> > Thank you for your continued discussion and for maintaining an overall positive perspective on our work. We would like to further clarify our ideas and address your remaining questions.
> >
> > ----------
> >
> > ### **On the "RL-CONGEST framework is designed to ..." part:**
> >
> > In the Introduction, we aimed to convey our idea of analyzing GNN expressiveness from the perspective of performing algorithmic tasks or algorithm alignment by surveying related works that use WL tests or other algorithmic tasks to evaluate GNNs. To make this point clearer, we have added a one-sentence explicit description in the abstract (Line 24, highlighted in blue) in our revised manuscript.
> >
> > ----------
> >
> > ### **On your "General Perspective":**
> >
> > We respectfully disagree with your assertion due to inconsistencies in your position. It confuses us that you cite these papers as examples of what you wish our results to achieve, yet their expressiveness analysis is also limited to algorithm alignment, with downstream task results being purely empirical.
> >
> > In the works you cited [1-3], the authors first theoretically analyze model expressiveness by evaluating whether the models can perform WL tests, biconnectivity decision, or subgraph counting (all algorithmic tasks, e.g., WL tests correspond to graph isomorphism tests, while the other two are direct algorithmic tasks). They then empirically evaluate the models' performance on downstream tasks. Notably, their theoretical analysis of expressiveness is also limited to the algorithmic tasks the models can perform, without guaranteeing real-world performance. Our focus aligns with this theoretical aspect of expressiveness. Besides, there are many well-known purely theoretical papers on the expressiveness of $k$-WL tests from an algorithmic alignment perspective, such as [Cai et al., 1989; Grohe, 1998; Grohe, 2017].
> >
> > You agree that analyzing expressiveness by examining the algorithmic tasks models can perform is valid, as demonstrated by your citation of [1-3]. This is precisely what we have done—evaluating models' expressiveness through the algorithmic tasks they can perform.
> >
> > ----------
> >
> > ### **On "whether these conclusions can help me ... design new expressive GNNs":**
> >
> > Loukas has already shown that MPGNNs can compute any computable problem if nodes are provided sufficient computational resources. Therefore, designing more expressive GNNs should prioritize enhancing the expressiveness of the update function rather than pursuing higher levels in the WL hierarchy. For instance, researchers might explore replacing MLPs with LLM agents empowered with CoTs, which are claimed to have the expressiveness of the $\mathsf{P}$ class, rather than relying on feature precomputation.
> >
> > ----------
> >
> > ### **On your concern about breaking equivariance or invariance:**
> >
> > Reviewer YAM3 raised a similar concern, which we respectfully disagree with. Providing unique identifiers does not inherently break equivariance or invariance. Our RL-CONGEST framework allows nodes to know their IDs but does not enforce their use as features, ensuring flexibility. Consider the following points:
> > 1. The RL-CONGEST model only requires nodes to have unique identifiers to ensure they are distinguishable. Researchers are free to analyze equivariance or invariance by permuting node IDs during experiments.
> > 2. In practical implementations (e.g., PyG), nodes are typically assigned IDs to manage their features. This setting does not conflict with equivariance or invariance, as models can freely decide whether to use these IDs as input features. RL-CONGEST explicitly states that nodes can be uniquely identified, which aligns with practical implementations and does not impose stricter conditions.
> >
> > As an example, consider Zhang et al.'s GD-WL test. Under a non-anonymous setting, RL-CONGEST can solve edge-biconnectivity if nodes have unique IDs. However, this result only assumes nodes are distinguishable and does not require a specific "canonical" ID assignment. If one ID assignment solves the problem, any permuted ID assignment would also work, preserving permutation-invariance.
> >
> > ----------
> >
> > We hope these clarifications address your concerns and provide further insight into the rationale behind our framework. Thank you again for your thoughtful engagement.
> >
> > ----------
> >
> > **References:**
> >
> > [Cai et al., 1989] Jin-yi Cai, Martin Furer, and Neil Immerman. An optimal lower bound on the number of variables for graph identification. FOCS 1989.
> >
> > [Grohe, 1998] Martin Grohe. Finite variable logics in descriptive complexity theory. Bull. Symb. Log., 4(4):345–398, 1998.
> >
> > [Grohe, 2017] Martin Grohe. Descriptive Complexity, Canonisation, and Definable Graph Structure Theory, volume 47 of Lecture Notes in Logic. Cambridge University Press, 2017.

---

> > > ### Comment · Reviewer_b62D · 2024-11-21
> > >
> > > **To make this point clearer, we have added a one-sentence explicit description in the abstract (Line 24, highlighted in blue) in our revised manuscript.**
> > >
> > > Thanks for that.
> > >
> > > **General Perspective**
> > >
> > > I try to make my point more correctly. I am wrong in assuming the analysis in the mentioned paper is not an algorithmic task. But what I want to bring up is that these models like GD-WL are not only able to solve a particular algorithmic task, but they have been proven to be much more than that. The authors state that the "RL-CONGEST framework is designed to assess a model's expressive power in executing algorithmic tasks or achieving "algorithmic alignment"". I am wondering, how the RL-CONGEST framework is able to access that one model can achieve algorithmic alignment on all tasks a model can perform, in order to decide whether a particular model is not algorithmic alignment. I believe only if a model requires more complexity than all tasks it can perform, we are safe to state that the model is not algorithmic alignment.
> > >
> > > **Loukas has already shown that MPGNNs can compute any computable problem if nodes are provided sufficient computational resources.**
> > >
> > > I didn't go deep into this reference, but I believe to make it true, you still need to break the permutation invariance or assign a unique ID for GNN. However, the ultimate goal for GNNs or all other deep learning models is that: we want to use them to solve some real-world problems, by only training the model on the training set and hoping it can generalize to unseen samples. However, by breaking the permutation invariance or equivariance, the model is just not able to generalize well, compared to models that preserve the permutation [1].
> > >
> > > **Therefore, designing more expressive GNNs should prioritize enhancing the expressiveness of the update function rather than pursuing higher levels in the WL hierarchy.**
> > >
> > > Still, my opinion is that all theoretical models or results should finally be able to have implications in a practical way. Enhancing the expressiveness of the update function is indeed important, as shown in the paper from the theoretical view. However, empirical experiments still show that by continues improve the expressive power, (from MPNN [2] to subgraph GNN [3-4], finally to even more expressive GNNs [5]), we witness better and better results on the ZINC dataset. However, the comparison in [2] shows that by simply varying the architecture of MPNN, the difference is marginal.
> > >
> > >
> > > **Our RL-CONGEST framework allows nodes to know their IDs but does not enforce their use as features, ensuring flexibility; Our RL-CONGEST framework allows nodes to know their IDs but does not enforce their use as features, ensuring flexibility**
> > >
> > > The point here is that whether we use unique ID or not in GNNs can have a significant impact on the downstream performance, which may imply the discrepancy between the theoretical model and real-world scenarios. Basically, a GNN that is trained on a specific ID assignment algorithm will not work if, in the test set, we use a different ID assignment algorithm or even if the graph distribution (like graph size) changes. Of course, we can permute the ID during the training. However, the model must see all $O(n!)$ different permutations to have appropriate generalization ability.
> > >
> > >
> > > **In practical implementations (e.g., PyG), nodes are typically assigned IDs to manage their features**
> > >
> > > The ID used in the code implementation is not the ID I am talking about. PyG uses node ID just to implement the MPNN algorithm. However, the permutation of the ID used in PyG will not result in a difference in the final computation result. However, premutate the ID in the input feature.
> > >
> > > Or I can ask it in another way, given an RL-CONGEST model, how do you train an MPNN to solve connectivity problems using a train graph set and predict an unseen graph set with maybe a different graph distribution?
> > >
> > > [1] Elesedy, Bryn, et al. “Provably Strict Generalisation Benefit for Equivariant Models”, ICML21.
> > >
> > > [2] Dwivedi, Vijay, et al. "Benchmarking Graph Neural Networks", ArXiv.
> > >
> > > [3] Zhang, Muhan, et al. "Nested Graph Nerual Networks", NeurIPS21.
> > >
> > > [4] Zhao, Lingxiao, et al. "From Stars to Subgraphs: Uplifting Any GNN with Local Structure Awareness", ICLR22
> > >
> > > [5] Feng, Jiarui, et al. "Extending the Design Space of Graph Neural Networks by Rethinking Folklore Weisfeiler-Lehman", NeurIPS23.

---

> > > > ### Author Response · Authors · 2024-11-24
> > > >
> > > > Dear Reviewer b62D,
> > > >
> > > > As a supplement, we have updated our PDF, replacing "anonymous" with "identical-feature" and "non-anonymous" with "distinct-feature" or "unique-feature" to make these concepts clearer and more accessible to readers. Additionally, we have included a discussion on the four common unique-feature settings in Section 3.2, highlighted in magenta.
> > > >
> > > > Thank you.

---

> ### Author Response · Authors · 2024-11-22
>
> Dear Reviewer b62D,
>
> Thank you for your kind reply. We believe there are two key points where we may not yet have reached a consensus, and we would like to further clarify our perspective.
>
> ----------
>
> ### **On "node IDs" and "non-anonymity":**
>
> These terms refer to unique features that allow nodes to be distinguishable (e.g., $[n] = \\{0, 1, \cdots, n - 1\\}$ would also suffice). The **distinct-feature setting** is commonly applied in **almost all existing models**, as listed below:
> 1. LINKX [Lim et al., 2021]:
>
> LINKX uses:
> - $\mathbf{H}^{(\mathbf{A})} = \mathrm{MLP}_{\mathbf{A}}(\mathbf{A})$
> - $\mathbf{H}^{(\mathbf{X})} = \mathrm{MLP}_{\mathbf{X}}(\mathbf{X})$
> - $\mathbf{Y} = \mathrm{MLP}(\sigma(\mathbf{W}[\mathbf{H}^{(\mathbf{A})}; \mathbf{H}^{(\mathbf{X})}] + \mathbf{H}^{(\mathbf{A})} + \mathbf{H}^{(\mathbf{X})}))$
>
> The $\mathbf{H}^{(\mathbf{A})}$ term can be reformulated as $\mathrm{MLP}'(\sigma(\mathbf{A} \cdot \mathbf{I} \cdot \mathbf{W}))$, which uses the identity matrix (unique node features).
>
> 2. GCN, GAT, etc., on real-world datasets:
>
> These models often use real-world features, which are unique and distinguishable with high probability. Actually, models that are applicable to real-world datasets fall into this category.
>
> 3. GNN expressiveness works (e.g., [Loukas, 2020; Sato et al., 2021]):
>
> These works use random features, which are unique with high probability. For example, assigning each node a feature randomly chosen from $[n^4]$ would result in distinct features with high probability.
>
> 4. GD-WL framework:
>
> In the GD-WL framework by Zhang et al., resistance distances $R(s, t)$ are used as features. Since $R(s, t) = 0$ iff $s = t$, each row of the resistance distance matrix is unique, creating distinguishable node features.
>
> Actually, according to our theory, they are all capable of solving the biconnectivity problem using the unique features.
>
> ----------
>
> ### **On whether unique features break equivariance or invariance:**
>
> Unique features do not break permutation equivariance or invariance. Instead, it is the properties of the update functions and pooling layers that determine whether a GNN model is equivariant or invariant. For example, in LINKX, when performing node classification, $\mathrm{MLP}(\mathbf{A})$ ensures permutation equivariance. To achieve permutation invariance for graph classification, we only need to add a permutation-invariant pooling layer after this step.
>
> Similarly, consider Dijkstra's single-source shortest path algorithm. Unique IDs are used solely to determine whether the shortest path to a node has been found. The resulting shortest path distance vector is always permutation equivariant. This demonstrates that it is not the presence of unique IDs but rather the design of the update function that determines whether a GNN model is permutation equivariant or invariant.
>
> ----------
>
> We hope these clarifications address your concerns and further show the flexibility of our framework. Thank you again for your engagement and constructive feedback.
>
>
> **Reference:**
>
> [Lim et al., 2021]. Large Scale Learning on Non-Homophilous Graphs: New Benchmarks and Strong Simple Methods. NeurIPS 2021.
> [Sato et al., 2021] Ryoma Sato, Makoto Yamada, and Hisashi Kashima. Random Features Strengthen Graph Neural Networks. SDM 2021.

---

> ### Comment · Reviewer_b62D · 2024-11-25
>
> The LINKX is only applicable to transductive settings, where we only have a single graph and do not require the model to generalize. At this time, by assigning each nodes a unique ID, the single MLP can achieve a "universal approximation" on any functions within this particular graph. If this is the unique ID you are referring, I think the statement becomes somehow meaningless. All GNN's expressiveness is analyzed with an assumption of the inductive setting, that is we want to train a GNN on some train graphs set and generalize it to unseen graphs with different sizes and graph distribution. And that's why the permutation invariant and equivariant are important.
>
> I have a similar feeling to reviewer YAM3 that authors always try not to answer my concerns directly and ignore some of my questions. So I try to make my question even more direct:
>
> **Could the authors give me a concrete example of how to use RL-CONGEST model and distinct-feature to train an MPNN model that can solve the bi-connectivity problem with some training graphs and labels? How can the model generalize to unseen graph samples with different graph structures and size?**
>
> I believe if your example is reasonable, most of my concerns can be solved.

---

> > ### Author Response · Authors · 2024-11-27
> >
> > Dear Reviewer b62D,
> >
> > Thank you for your discussion. We now have a clear understanding of your main concern: a concrete construction of an RL-CONGEST model capable of solving the biconnectivity problem.
> >
> > ----------
> >
> > ### **A Concrete RL-CONGEST Example for Edge-Biconnectivity**
> >
> > First, we would like to clarify that, as with most expressiveness results, our claims focus on existing results and impossibility results. Whether a real-world model **can be trained** to solve specific algorithmic tasks are out of the scope of our paper, and may depend on the flexibility and strength of the update functions.
> >
> > Since the RL-CONGEST model uses distributed algorithms to characterize the message-passing process in GNNs, each distributed algorithm corresponds to an RL-CONGEST model. Here, we provide a sketch of constructing such a concrete RL-CONGEST model for edge-biconnectivity. The algorithm is designed by Pritchard [Pritchard, 2006], and we encourage reviewers to refer to Pritchard's slides (http://ints.io/daveagp/research/2006/ac-bicon.pdf) for visual aids and proofs of correctness.
> >
> > ----------
> >
> > **Steps:**
> > 1. Build a spanning tree $T$ with the FLOOD algorithm rooted at node $0$ (Since nodes have unique features and are distinguishable, we can "rename" them as $\\{0, 1, \cdots, n-1\\}$ for the description):
> > 2. Compute the number of descendants on $T$:
> > 	- Step 2.1: The root node $0$ sends a message to its children: "Compute the number of descendants". This message propagates down the tree.
> > 	- Step 2.2: Leaf nodes determine their size as $1$ (here we define each node is its own descendant) and report this value to their parent. Internal nodes wait for responses from all their children, sum the values, add $1$ for themselves, and report the total to their parent.
> > 3. Preorder (i.e., the label of a vertex is smaller than the label of each of its children) the nodes:
> > 	- Step 3.1: The root assigns itself label $1$.
> > 	- Step 3.2: When node $v$ assigns itself label $x$, it determines labels for its children $c_1, c_2, \cdots$ in some arbitrary order. For child $c_i$, the label is computed as: $\ell_i = x + 1 + \sum_{j < I}\\#\text{desc}(c_j)$.
> > 4. Marking cycles (from this step, we refer to nodes by its preorder label.):
> > 	- Step 4.1: For a given non-tree edge $(u, v)$, a message $M[u, v]$ is sent along the edge in both directions: "If you are an ancestor of both $u$ and $v$, ignore this message. Otherwise, pass the message to your parent and mark the edge connecting you to your parent". A node $w$ checks the ancestry condition by verifying if $\\{u, v\\} \subseteq \\{w, w + 1, \ldots, w + \\#\text{desc}(w) - 1\\}$.
> > 	- Step 4.2: Each node tracks the cumulative $\min u_i$ and $\max v_i$ of all $M[u_i, v_i]$ messages received.
> > 	- Step 4.3: Even if $v$ determines that its edge to its parent should not be marked, it sends a token message to its parent.
> > 	- Step 4.4: Once $v$ has received all non-to-parent edge messages, it sends a message to its parent.
> >
> > After completing phases 1–4, the non-marked edges are bridges.
> >
> > This also shows that in scenarios where distinct node features are available (as is common), enhancing the expressiveness of update functions would further enhance the expressiveness of GNNs.
> >
> > ----------
> >
> > ### **On LINKX and the Inductive Setting**
> >
> > Actually, LINKX itself can be directly applied to the inductive setting, similar to GCN and GCNII (GCNII's paper include inductive learning experiments). The high-level idea is that the model learns a weight matrix  $\mathbf{W}$ from a graph $\mathbf{A}_1$ and features $\mathbf{X}_1$ (or a training set of graphs and features) and then directly uses this learned representation for inference on new data. Although the authors of LINKX have not explicitly tested it in the inductive setting, two similar models—SA-MLP [Chen et al., 2024] and SymphoNEI [Kim et al., 2024]—both of which utilize $\mathrm{MLP}(\mathbf{A})$, have showed effectiveness in inductive scenarios.
> >
> > ----------
> >
> > We hope these explanations provide clarity and address your concerns. Thank you again for your engagement.
> >
> > ----------
> >
> > **Reference:**
> >
> > [Pritchard, 2006] David Pritchard. An Optimal Distributed Edge-Biconnectivity Algorithm. arXiv 2006.
> >
> > [Chen et al., 2024] Jie Chen, Mingyuan Bai, Shouzhen Chen, Junbin Gao, Junping Zhang, and Jian Pu. SA-MLP: Distilling Graph Knowledge from GNNs into Structure-Aware MLP. TMLR 2024.
> >
> > [Kim et al., 2024] Kyusik Kim, and Bongwon Suh. SymphoNEI: Symphony of Node and Edge Inductive Representations on Large Heterophilic Graphs. DASFAA 2024.

---

> > > ### Comment · Reviewer_b62D · 2024-11-29
> > >
> > > I believe I tried my best to explain my question multiple times, but now I feel authors either don't have enough understanding of the related topic or are deliberately avoiding my central concern through sophistry and answering something that looks reasonable but actually not even close to my question. Therefore, I will stop discussing with the author here and leave my discussion to the AC-reviewer phase. This is my final response to the authors.
> > >
> > > **A Concrete RL-CONGEST Example for Edge-Biconnectivity**
> > >
> > > I know there are algorithms that can solve edge connectivity problems. But my question is there a concrete approach that can train an MPNN model to solve edge-connectivity problems for **unseen graphs with different sizes and distributions** based on the RL-congest framework? I will not expect authors to really train a model or achieve 100% accuracy (you are free to assume that your update function is powerful enough in this conceptual question.). I am just asking if it is possible and how, as the author continues to say the unique ID can improve the expressiveness of MPNN and enable MPNN to solve edge-connectivity problems. Using a statement like **out of the scope of our paper** is a sign of deliberately avoiding a direct answer and indicates the incapability of the proposed model.
> > >
> > > **Inductive learning**
> > >
> > > GCNII still falls under the message-passing category, which is fundamentally different from MLP(A). Therefore, GCNII can do inductive learning but that does not mean LINKX can do it. By using MLP(A), you already assume the ID for each node in A, if you permute the order of A, the result will change and an MLP trained on a graph of $A\in R^{n\times n}$ can not be applied on a graph of $A \in R^{m \times m}$. Therefore, it only works on transductive settings for graph data.
> > >
> > > SA-MLP focuses on the point cloud, where each sample has the same size (or say same number of nodes, and each node actually has its absolute position). SymphoNEI is just wrong in its statement of inductive learning. inductive learning means a model can generalize to **graphs with different size (node number) and distribution (structure)**
> > >
> > > Using GCNII, SA-MLP, and SymphoNEI as examples indicates the author either doesn't understand the meaning of inductive learning or deliberately avoids answering my central concern.

---

> ### Author Response · Authors · 2024-11-30
>
> Dear Reviewer b62D,
>
> Thank you for your response. We would like to further clarify our ideas and address your concerns.
>
> ----------
>
> ### **On Existence and Trainability**
>
> We have clearly stated in our previous response that in our paper, we focus on existence and impossibility results, and **do not address how to use real-world optimizers to train a model** to solve problems like biconnectivity. These are two aspects of independent interest. It is not fair to criticize us that we are "avoiding problems" simply because we state that trainability is beyond the scope of this paper. Furthermore, we are not aware of any GNN expressiveness paper (proving that GNNs can solve specific algorithmic tasks) that has theoretically showed how to train a GNN using SGD or other optimization techniques to solve such tasks. If you are aware of such works, please list them, as we would be eager to learn from their techniques for future improvements.
>
> In our paper, as in the works we cite, we prove theorems of the form: "**for each graph** $G$, there exists an RL-CONGEST model that operates on it and solves the algorithmic task". These existence results are universal and hold for every graph. However, we do not claim to prove how such a model can be practically trained.
>
> ----------
>
> ### **On LINKX in the Inductive Learning Setting**
>
> Simply stating that LINKX is not applicable to the inductive learning setting is a misunderstanding. We can address this by setting a maximum number of nodes for graphs, say $N$, and **padding all adjacency matrices** $\mathbf{A}_i \in \mathbb{R}^{n_i \times n_i}$ to $\mathbf{A}_i' = \begin{bmatrix} \mathbf{A}_i & 0 \\\\ 0 & 0 \end{bmatrix} \in \mathbb{R}^{N \times N}$. This approach mirrors the padding technique commonly used in the NLP domain.
>
> Your concerns are akin to questions in the NLP area like, **"The length of inputs varies, how can they be input into the same Transformer?" or "Your model can only handle sequences of the same length and cannot be applied to the inductive setting"**. The first concern is resolved using padding, and the second has been validated by the success of Transformer-based large language models.
>
> ----------
>
> Overall, we deeply appreciate your effort in engaging in discussions with us.

---

### Official Review · Reviewer_DTJH · 2024-10-30

**Soundness:** 3
**Presentation:** 3
**Contribution:** 2
**Rating:** 5
**Confidence:** 4

**Summary:**

The authors very correctly point out that the current theoretical analysis of GNNs is lacking in a few key ways (e.g. granularity and taking into account computational expense). To remedy that they propose using Resource-Limited CONGEST model, instead of usual CONGEST and relating WL-tests to model-checking problems that can prove a more granular expresivity testing.

**Strengths:**

I agree with the authors that the theoretical expresivity analysis of GNNs is quite lacking. It makes a lot of sense to limit the computational power of the nodes (GNN update functions). As that is more realistic. The idea to use model-checking problems instead of WL to judge the theoretical power of GNNs is novel and I think quite promissing, as it allows for higher granularity.

This work also provides interesting motivation for why virtual nodes help, as they are a very common tool in practice. One of the first works to look at this theoretically to the best of my knwoledge.

It's generally well written and easy to follow.

**Weaknesses:**

Authors stress that "unlimited computational resources of CONGEST" is an issue and chose to just use a more restrictive computation class for the node updates. Ideally I'd like to see this being contrasted with the universal approximation theorem for MLPs. As the update function is usually an MLP it's power I'd say is more defined by approximation quality of whatever computation it needs to perform.

In the section "Additional Features Empower Models by Breaking Anonymity?" authors say that it's not good that some expressive GNNs might be breaking anonymous setting by using additional features. I would say that this is not a good way to look at this. In my opinion that the point of a good chunk of more expressive GNN research is precisely how to add pseudo-indentifiers to a graph with as few negative impacts (bad generalization).

Speaking about negative impacts of node identifiers, in the proposed computation model authors permit "nodes to be aware of their own unique IDs". This doesn't make much sense from ML perspective as generalization will be terrible if a stable ID assignment is not possible, and normally it is not possible to do so on general graphs. So for a paper arguing about making theoretical GNN analysis more realistic I think this is a noteable issue.
Authors do motivate this choice by saying that "real-world graph datasets are rich in node features". I'd argue that this is still very far away from node IDs, e.g. say if features are just a few different atom types in case of many molecular tasks. I'd like to see some data analysis showing the unique identifiability of nodes in multitude of real world datasets to convince me that this is the case.

The work also lacks direct applicability to fixing or ranking GNN architectures. Which would be the main benefit of the newly proposed GNN analysis. To make the paper complete I would like to see analysis/ranking of some few popular GNN architectures and hopefully showing that this translates to some real tasks, for example ones for which the assumptions, such as unique identifiability by node features, more or less hold.

Also, speaking about popular GNN architectures, authors skipped the two first subgraph GNN papers, when discussing subgraph GNNs (https://arxiv.org/abs/2110.00577 https://arxiv.org/abs/2111.06283)

**Questions:**

Distributed computing has various computation models already, besides LOCAL and CONGEST. It would be nice if authors would dig a bit deeper in the distributed computing literature to see what alternatives already exist and if they would be more fitting than CONGEST. It's been a while since I looked at those myself, but for example https://arxiv.org/pdf/1202.1186 investigates a very restricted computational model, that should still be able to simulate a WL test (it was also used in some simpified GNNs https://arxiv.org/pdf/2205.13234). I'm sure that others exist as well.

---

> ### Author Response · Authors · 2024-11-15
> **Initial Response to Reviewer DTJH (1/3)**
>
> Dear Reviewer DTJH,
>
> Thank you for reviewing our paper. We are very grateful for your detailed feedback and appreciate the opportunity to address some misunderstandings that may have arisen in the “Weaknesses” section of your review.
>
> **Regarding Unlimited Computational Resources in the CONGEST Model**:
>
> We respectfully disagree with the comment that "just use a more restrictive computation class for the node updates". Our goal is to introduce flexible constraints on the resources class $\mathsf{C}$ to derive different independent results, as discussed in Lines 381-389. For instance, setting $\mathsf{C} = \mathsf{R}$ (the class of recursive languages decidable by Turing machines) and network width $w = O(1)$ transforms our RL-CONGEST framework into the CONGEST model. By setting $\mathsf{C}$ to a class such as $\mathsf{TC}^0$, which reflects the capabilities of MLPs, the resulting model would resemble "real-world" GNNs with MLPs as update functions. Alternatively, if node update functions used transformer-based LLM agents enhanced by Chain-of-Thought (CoT) reasoning, which are claimed to solve problems in $\mathsf{P}$ exactly [Merrill et al., 2024; Li et al., 2024], we could set $\mathsf{C} = \mathsf{P}$ to derive new theoretical results based on this adjustment. We hope that our framework can inspire future research on graph agents, and have added it in red color in the revised PDF (Lines 384-387). As discussed in Lines 381-389, adjusting $\mathsf{C}$ in different ways may yield diverse outcomes, making our RL-CONGEST framework a "framework scheme" or "framework template".
>
> We respect your statement that "is more defined by approximation quality of whatever computation it needs to perform". We recognize the importance of the Universal Approximation Theorem (UAT) in machine learning and are aware of work addressing the approximation capabilities of GNNs, such as [Azizian et al., 2021; Wang et al., 2022]. However, as indicated by our paper's title, our work aligns with a different research path, focusing on a model's expressiveness through its capability to perform algorithmic tasks. For example, [Loukas, 2020] uses the CONGEST model to analyze MPGNNs' algorithmic abilities, while the Outstanding Paper at ICLR 2023 [Zhang et al., 2023] assesses GNNs' power to determine graph biconnectivity. These two lines of research --- expressiveness for algorithmic tasks versus approximation quality --- are largely orthogonal and develop independently. Additionally, discussions in the literature (e.g., Section 1.1 in [Loukas, 2020], which states that "**Turing completeness is a strictly stronger property** than universal approximation") suggest that Turing completeness is indeed a stronger property than universal approximation. Therefore, we believe that our focus on computability is sufficiently general and without loss of scope.

---

> > ### Author Response · Authors · 2024-11-15
> > **Initial Response to Reviewer DTJH (2/3)**
> >
> > **On Additional Features Enhancing Models by Breaking Anonymity**:
> >
> > Our framework permits nodes to access unique IDs, but this **does not imply that models must use them**. This flexible setting is compatible with various feature types, including pseudo-identifiers or molecular types, as you mentioned. This choice is motivated by our observation that existing works often equate MPGNNs' expressive power with the anonymous WL test, which we find to be a mismatch due to the questionable anonymous setting. In Section 3.2, we aim to point out that previous works' equating anonymous WL with MPGNNs is not entirely reasonable, and thus concluding that MPGNNs are weak because WL test is weak is also debatable. In fact, MPGNNs can perform certain algorithms (such as solving edge biconnectivity in $O(D)$ rounds within the CONGEST model [Pritchard, 2006], Lines 311-313).
> >
> > For clarity, we summarize the logical flow of Section 3.2 as follows:
> > 1. Numerous studies following the seminal work GIN [Xu et al., 2019] claim that the vanilla WL test has limited expressive power --- a claim that is true, as shown in Figure 2. However, the appropriateness of using the **anonymous WL test to characterize MPGNNs is debatable**, given that real-world graphs frequently contain rich features. Additionally, [Loukas, 2020] demonstrated that with unique IDs (and other assumptions), MPGNNs can perform a wide range of algorithmic tasks.
> > 2. To address the "limited" expressiveness of MPGNNs (stemming from the WL test's limitations), some works incorporate additional features (e.g., [Zhang et al., 2023]) to increase their models' expressiveness. Nonetheless, as discussed in (1), the anonymous WL test may not be the appropriate characterization for MPGNNs. Consequently, some studies' approach of demonstrating their model's expressiveness advantage by proving it can perform tasks beyond the WL test's capabilities may not be entirely valid. A more reasonable comparison would use MPGNNs in a non-anonymous setting (as suggested in [Loukas, 2020]). Further, evidence from [Loukas, 2020; Suomela, 2013; den Berg et al., 2018; You et al., 2021; Abboud et al., 2021; Sato et al., 2021] shows that non-anonymous settings can enhance model expressiveness, highlighting a mismatch when studies argue for "weak MPGNNs" yet use features that break the WL test's anonymity to boost expressiveness.
> > 3. Our framework allows nodes to know their unique IDs, though this is **optional**. This flexibility is compatible with the use of features such as "a few different atom types in molecular tasks". Our RL-CONGEST analysis framework can apply to studies proposing new GNN variants that leverage additional features and claim the ability to perform specific algorithmic tasks, with the only requirement being a **clear specification of the preprocessing time complexity for these features**.
> >
> > **On "Lacks Direct Applicability to Fixing or Ranking GNN Architectures"**:
> >
> > Our RL-CONGEST analysis framework has practical applications, as illustrated through results like model checking. For example, we show that $k$-WL GNNs can perform PNF $C^k$ model checking --- a class of significant problems in theoretical computing --- while previous research aligned with WL tests, which are equivalent to the model equivalence problem. These results are discussed in detail in Section 4.3. However, please note that our paper's primary goal is to **highlight issues** in existing studies on GNN expressiveness and to propose a new analytical framework that **avoids these issues**. We do not aim to design a specific GNN model with improved performance or expressiveness or to provide guidance for such future work. Rather, we hope our framework will assist future research by helping to avoid issues discussed in Section 3 and encouraging a **re-evaluation of common assumptions** in GNN expressiveness studies.
> >
> > **On Subgraph GNNs**:
> >
> > Thank you for providing references to additional models. We have incorporated these references into the paper and marked them in red (Lines 43-44).

---

> > > ### Author Response · Authors · 2024-11-15
> > > **Initial Response to Reviewer DTJH (3/3)**
> > >
> > > **Regarding Other Distributed Computing Models**:
> > >
> > > Indeed, we are aware of various distributed computing models, such as the CONGEST-CLIQUE, Coordinator, and Blackboard models. Some of these models can be considered special cases of the CONGEST model. For example, the CONGEST-CLIQUE model can be implemented by adding virtual edges to make the original graph a complete graph; the Coordinator model can be implemented by adding a virtual node connected to all other nodes. However, the LOCAL and CONGEST models are still the most widely mentioned in distributed computing books [Peleg 2000], courses [Hirvonen et al., 2020; Ghaffari, 2022], and conferences, so we chose to focus our discussion on these two. Additionally, some of our ideas are inspired by [Loukas, 2020], which explores the relationship between GNNs and these two models. Our framework generalizes their results, but it is based on the CONGEST model.
> > >
> > >
> > > Thank you again for your detailed feedback. We hope our response clarifies our approach and addresses your concerns. We look forward to any further discussions.
> > >
> > >
> > > **References**:
> > >
> > > [Merrill et al., 2024] William Merrill, and Ashish Sabharwal. The Expressive Power of Transformers with Chain of Thought. ICLR 2024.
> > >
> > > [Li et al., 2024] Zhiyuan Li, Hong Liu, Denny Zhou, and Tengyu Ma. Chain of Thought Empowers Transformers to Solve Inherently Serial Problems. ICLR 2024.
> > >
> > > [Azizian et al., 2021] Waiss Azizian, and Marc Lelarge. Expressive Power of Invariant and Equivariant Graph Neural Networks. ICLR 2021.
> > >
> > > [Wang et al., 2022] Xiyuan Wang, and Muhan Zhang. How Powerful are Spectral Graph Neural Networks. ICML 2022.
> > >
> > > [Loukas, 2020] Andreas Loukas. What Graph Neural Networks Cannot Learn: Depth vs Width. ICLR 2020.
> > >
> > > [Pritchard, 2006] David Pritchard. An Optimal Distributed Edge-Biconnectivity Algorithm. arXiv 2006.
> > >
> > > [Xu et al., 2019] Keyulu Xu*, Weihua Hu*, Jure Leskovec, Stefanie Jegelka. How Powerful are Graph Neural Networks? ICLR 2019.
> > >
> > > [Zhang et al., 2023] Bohang Zhang, Shengjie Luo, Liwei Wang, and Di He. Rethinking the Expressive Power of GNNs via Graph Biconnectivity. ICLR 2023.
> > >
> > > [Suomela, 2013] Jukka Suomela. Survey of Local Algorithms. ACM Computing Surveys (CSUR), 45(2):24, 2013.
> > >
> > > [den Berg et al., 2018] Rianne van den Berg, Thomas N Kipf, and Max Welling. Graph Convolutional Matrix Completion. KDD 2018.
> > >
> > > [You et al., 2021] Jiaxuan You, Jonathan M Gomes-Selman, Rex Ying, and Jure Leskovec. Identity-aware Graph Neural Networks. AAAI 2021.
> > >
> > > [Abbound et al., 2021] Ralph Abboud, Ismail Ilkan Ceylan, Martin Grohe, and Thomas Lukasiewicz. The Surprising Power of Graph Neural Networks with Random Node Initialization. IJCAI 2021.
> > >
> > > [Sato et al., 2021] Ryoma Sato, Makoto Yamada, and Hisashi Kashima. Random Features Strengthen Graph Neural Networks. SDM 2021.
> > >
> > > [Peleg, 2000] Distributed Computing: A Locality-Sensitive Approach. 2000.
> > >
> > > [Hirvonen et al., 2020] Juho Hirvonen and Jukka Suomela. Distributed Algorithms (course). 2020. https://jukkasuomela.fi/da2020/
> > >
> > > [Ghaffari, 2022] Mohsen Ghaffari. Distributed Graph Algorithms (course). 2022. https://people.csail.mit.edu/ghaffari/DA22/Notes/DGA.pdf

---

> ### Author Response · Authors · 2024-11-21
> **Follow-Up on Rebuttal Discussion**
>
> Dear Reviewer DTJH,
>
> As we are now midway through the rebuttal phase, we want to kindly follow up to ensure that our responses have adequately addressed your concerns. Your feedback is highly valued, and we still looking forward to further discussion to clarify or expand on any points as needed. Please feel free to share any additional thoughts or questions you might have.
>
> Thank you once again for your time and effort in reviewing our paper.

---

> ### Comment · Reviewer_DTJH · 2024-11-26
>
> Thank you for your rebuttal.
>
> I have a  few additional followups on your response:
>
> > Our framework permits nodes to access unique IDs, but this does not imply that models must use them.
>
> Can you give an example of how a model without such features (or better yet with say 5 potential features/classes, think atom types, with number of nodes n >> 5) would be analysed in your framework?
>
> > We do not aim to design a specific GNN model with improved performance or expressiveness or to provide guidance for such future work. Rather, we hope our framework will assist future research by helping to avoid issues discussed in Section 3 and encouraging a re-evaluation of common assumptions in GNN expressiveness studies.
>
> I never asked for any state of the art results. What I meant, similar to the comment above, is that it would be helpful if the paper would actually use RL-CONGEST to analyse a few different popular existing GNN architectures. To show how it's done and that RL-CONGEST can provide a meaningful differentiation between different GNN approaches, that was not possible so far.

---

> ### Author Response · Authors · 2024-11-28
>
> Dear Reviewer DTJH,
>
> Thank you for your response. We would like to further clarify our ideas and address your concerns.
>
> ----------
>
> ### **On the Usage of Unique IDs**
>
> First, let us state "unique IDs" more precisely: they refer to nodes being uniquely identifiable, such as through distinct features.
>
> Second, our intended meaning can be formally described as follows: when analyzing GNN expressiveness from an algorithmic alignment perspective using the RL-CONGEST model, the RL-CONGEST model requires unique IDs solely to identify nodes and compare whether two nodes are distinct, as in traditional graph algorithms. The model **does not rely on the concrete values of the IDs**, as these values are typically arbitrary and carry no intrinsic meaning.
>
> In our paper, in line with related works, we primarily focus on the algorithmic tasks that GNNs can perform rather than downstream tasks, so we provide an example of constructing an RL-CONGEST model (or equivalently, a distributed algorithm) to solve the edge-biconnectivity problem. This algorithm is designed by Pritchard [Pritchard, 2006]. For further details, we encourage reviewers to consult Pritchard's slides (http://ints.io/daveagp/research/2006/ac-bicon.pdf), which include visual aids and proofs of correctness.
>
> ----------
>
> **Steps:**
> 1. Build a spanning tree $T$ with the FLOOD algorithm rooted at node $0$ (Since nodes have unique features and are distinguishable, we can "rename" them as $\\{0, 1, \cdots, n-1\\}$ for the description):
> 2. Compute the number of descendants on $T$:
> 	- Step 2.1: The root node $0$ sends a message to its children: "Compute the number of descendants". This message propagates down the tree.
> 	- Step 2.2: Leaf nodes determine their size as $1$ (since each node is its own descendant) and report this value to their parent. Internal nodes wait for responses from all their children, sum the values, add $1$ for themselves, and report the total to their parent.
> 3. Preorder (i.e., the label of a vertex is smaller than the label of each of its children) the nodes:
> 	- Step 3.1: The root assigns itself label $1$.
> 	- Step 3.2: When node $v$ assigns itself label $x$, it determines labels for its children $c_1, c_2, \cdots$ in some arbitrary order. For child $c_i$, the label is computed as: $\ell_i = x + 1 + \sum_{j < i} \\#\text{desc}(c_j)$.
> 4. Marking cycles (from this step, we refer to nodes by its preorder label.):
> 	- Step 4.1: For a given non-tree edge $(u, v)$, a message $M[u, v]$ is sent along the edge in both directions: "If you are an ancestor of both $u$ and $v$, ignore this message. Otherwise, pass the message to your parent and mark the edge connecting you to your parent". A node $w$ checks the ancestry condition by verifying if $\\{u, v\\} \subseteq \\{w, w + 1, \ldots, w + \\#\text{desc}(w) - 1\\}$.
> 	- Step 4.2: Each node tracks the cumulative $\min u_i$ and $\max v_i$ of all $M[u_i, v_i]$ messages received.
> 	- Step 4.3: Even if $v$ determines that its edge to its parent should not be marked, it sends a token message to its parent.
> 	- Step 4.4: Once $v$ has received all non-to-parent edge messages, it sends a message to its parent.
>
> After completing phases 1–4, the non-marked edges are bridges.
>
> ----------
>
> ### **Using RL-CONGEST to Analyze Existing Models**
>
> Theorem 2 and Theorems 6-8 correspond to existing GNN models. Since we analyze the expressiveness of GNNs from an algorithmic alignment perspective, our results focus on the models' ability to solve algorithmic tasks rather than traditional node classification or link prediction tasks. Therefore, at this initial stage of using the RL-CONGEST analysis framework, we do not have results for models designed specifically for downstream tasks, such as GCN and GAT. Instead, our results focus exclusively on models proposed in GNN expressiveness studies. The findings are summarized in the table below.
>
> |Preprocessing Time (Content)|Nodes' Computational Resources Class $\mathsf{C}$|Message-Passing Rounds|Algorithmic Task Solved|Corresponding Model (Reference)|
> |-|-|-|-|-|
> |$0$|$\mathsf{TIME}(n)$|$O(D)$|Edge-Biconnectivity|**MPGNN** ([Pritchard, 2006])|
> |$O(\min(nm, n^{\omega}))$ (All-Pair RDs)|$\mathsf{TIME}(1)$|$0$| Edge-Biconnectivity|**GD-WL** (Thm. 2)|
> |$O(m)$ (Tarjan)| $\mathsf{TIME}(1)$|$0$|Edge-/Vertex-Biconnectivity|**Any MPGNN** (with Computed Answers)|
> |$0$ | $\mathsf{TIME}(n^2\log n)$ | $O(D + m/w)$|One Iteration of WL Test| **MPGNN** (Thm. 6)|
> |$O(n)$ (Virtual Node)|$\mathsf{TIME}(n^2\log n)$|$O(D + \Delta/w)$|One Iteration of WL Test|**MPGNN + Virtual Node** (Thm. 7)|
> |$O(kn^{k+1})$ ($k$-WL Graph & Features) |$\mathsf{TIME}(k^2n)$| $O(k^2)$|PNF $\mathcal{C}^k$ Model Checking|**High-Order GNNs** (Thm. 8)|
>
> ----------
>
> **Reference:**
>
> [Pritchard, 2006] David Pritchard. An Optimal Distributed Edge-Biconnectivity Algorithm. arXiv 2006.

---

> ### Comment · Reviewer_DTJH · 2024-12-03
>
> Thank you for your further clarification and the model comparison.
>
> From your answer about unique IDs, it stands to reason that models must have access to IDs and use them for the theory to hold. As pretty much all algorithms in distributed computing as you say, rely on nodes knowing which node sent which message (same in your example). Which contradicts your previous statement.
>
> With the not-fully unique node example I was asking about, I was also interested again, in how your thery meshes with real world scenarios. In the paper you claim that features make the nodes uniquely identifyiable in most real tasks, but as far as I can see its at best possible not to an abosolute degree (partial idetifiability). So I was wondering if your theory can deal with that. Your answer doesn't really help in this regard.
>
> I also looked at your discussion with the other reviewers who also largely expressed doubts. Thus I will keep my score.

---

> > ### Author Response · Authors · 2024-12-03
> >
> > Dear Reviewer DTJH,
> >
> > Thank you for your reply and engagement in the discussion.
> >
> > Regarding your first concern, please allow us to provide further clarification. First, it seems we share the understanding that the WL test requires identical initial features for all nodes. This requirement imposes a limitation on the form of input features. We just remove this limitation by allowing nodes to access unique IDs. Consequently, depending on the specific task, the model's learned mapping may or may not rely on the concrete values of the IDs, thereby making it more general compared to the WL test.
> >
> > As far as we know, existing works that analyze GNN expressiveness in the context of algorithmic tasks (e.g., biconnectivity or distinguishing certain graph types, as targeted by WL tests) have not provided theoretical guarantees for improving quantitative metrics in practical tasks such as node classification or graph classification. **At most, these works show the ability to distinguish certain graph pairs, which relates to problems of model equivalence or model checking**. Our Theorem 8 shows that RL-CONGEST can also perform such analyses, and we would greatly appreciate it if you could take a closer look.
> >
> > We understand that it is challenging to persuade all reviewers to fully agree with all our claims. However, we believe that our preliminary work provides value to the community by encouraging a reevaluation of the reasonableness of existing approaches.
> >
> > Once again, thank you for your comments and for engaging in this discussion. We hope our explanation addresses some your concerns.

---

### Official Review · Reviewer_YAM3 · 2024-11-01

**Soundness:** 2
**Presentation:** 3
**Contribution:** 3
**Rating:** 3
**Confidence:** 4

**Summary:**

This paper examines the limitations of the theoretical expressiveness of GNNs and introduces a novel computational framework, RL-CONGEST, which factors out pre- and postprocessing and limits the computational power of nodes. The authors further analyze the WL-test within this framework and contribute some theoretical insights. RL-CONGEST, while positioned primarily for GNNs, also offers implications for understanding computational constraints in other computation models.

**Strengths:**

* The paper introduces RL-CONGEST, a new computational model that addresses aspects previously overlooked in the GNN literature, particularly computational constraints at the node level.
* Some shortcomings in prior work are highlighted and critically analyzed, including preprocessing complexities and computational limits.
* RL-CONGEST has potential standalone value beyond GNNs, as it provides a framework to study computational complexity and expressiveness that could benefit other areas.

**Weaknesses:**

* Section 3.1: The authors argue that preprocessing time complexity is often underestimated in the GNN literature, with Wollschläger et al. (2024) as an example. However, this appears to be an isolated case rather than a trend in the field. A more robust case for this claim could be made by referencing additional studies or a systematic analysis that demonstrates the prevalence of overlooked preprocessing complexities. Zhang et al. (2023), which the authors cite and analyze, actually discusses preprocessing time explicitly in the paper, which weakens the generality of this argument. While it is valuable to account for preprocessing, demonstrating that this issue extends across multiple papers would strengthen the point. Further, as most of these papers mainly focus on expressiveness, computational complexity might just not be the main focus.

* Section 3.2: The “mismatch” claim between models with and without features lacks clear evidence. The advantage provided by features in model initialization is well-known, and the WL test is adaptable to both anonymous and pre-colored contexts. More detail and examples of specific instances where this mismatch has led to issues in the literature would clarify and strengthen the claim. The authors tend to write around what the mismatch actually is in this section and should clearly define it.

* Section 3.3: The assertion that CONGEST is “inappropriate” for direct use is somewhat unconvincing, as it can still serve as an upper bound for computational capacity. While RL-CONGEST’s constraint on node computation is a useful contribution, existing models are still relevant for the purpose of their analysis. Furthermore, Theorem 4 should explicitly assume a connected graph and the version stated in the paper is technically wrong. It is also worth noting that in many GNN studies, expressiveness rather than computational complexity is the focus, so adding computational constraints could shift the narrative and purpose of the study. If the authors are proposing RL-CONGEST as a practical standard for GNNs, specific examples and a discussion on which complexity classes should be used for GNNs would help contextualize it within the field.

* Adding computational constraints to CONGEST is an interesting approach, but it becomes very detached from the application in GNNs. For example, the authors do not go into detail on what complexity classes we should allow for GNNs. One could make an argument that as GNNs are usually implemented with fixed size networks that run in constant time, the computational envelope should also be constant to yield the most realistic bounds. RL-CONGEST is interesting on its own, but how the computational constraints should be best put to use should be discussed in paper that claims to investigate the GNNs. The paper would benefit from more guidance on how GNN practitioners should employ RL-CONGEST, along with concrete examples of benefits. A more precise articulation of the expected impact or practical value this framework could offer would also strengthen the contribution.

Overall, the paper makes several claims and only backs up some of them. In the end, it is not clear how the newly proposed model is supposed to be used in future work (should everybody just use their own complexity classes for the local computation, what benefit does this have?) and leaves the question on what impact this work can have. The authors should address this issue and formulate some clear benefits of their framework.

**Questions:**

* Could the authors clarify specific insights from the RL-CONGEST model that would be practically useful for GNN practitioners?
* Do the authors envision RL-CONGEST serving as a new standard or benchmark model for GNN complexity analysis? If so, could they suggest specific complexity classes for GNN applications or examples that showcase RL-CONGEST’s advantages?
* Could you clarify your position on CONGEST's usefulness as an upper bound and discuss whether RL-CONGEST complements rather than replaces existing models?
* Could you add a discussion on appropriate complexity classes for GNN analysis using RL-CONGEST? In that context, can you provide guidelines or a framework for GNN practitioners on how to effectively use RL-CONGEST in their research or applications?

---

> ### Author Response · Authors · 2024-11-15
> **Initial Response to Reviewer YAM3 (1/3)**
>
> Dear Reviewer YAM3,
>
> We are very grateful for your detailed feedback and appreciate the opportunity to address some misunderstandings.
>
> **W1**:
>
> Yes, we use two examples to demonstrate that preprocessing complexity is often underestimated in the literature. Please note that the GD-WL paper [Zhang et al., 2023] was awarded ***Outstanding Paper at ICLR 2023***. We chose this work as it is representative enough: its recognition by the community underscores that **even well-regarded papers can exhibit this "underestimated preprocessing complexity" issue**, making it a persuasive example to support our claim. However, we have also identified other examples, such as [Thiede et al., 2021, Bouritsas et al., 2022], which use hand-crafted features by recognizing subgraphs. The theoretical analysis also suggests that the proposed model achieves full expressiveness only when the subgraph is unrestricted, which is the same as [Wollschlager et al., 2024]. We have added them in Section 3.1 in red (Lines 212-216 in the revised PDF). Listing every example exhaustively is infeasible, so we have selected a recent example from ICML'24 [Wollschlager et al., 2024] and the notable ICLR'23 outstanding paper [Zhang et al., 2023] to substantiate our point in Section 3.1.
>
> Additionally, we **respectfully disagree** with "computational complexity might just not be the main focus".
> + First, for [Zhang et al., 2023] (and similar works), the authors show the model's expressiveness by assessing its capability in performing algorithmic tasks, making time complexity crucial in evaluating their results.
> + Second, while the authors discuss preprocessing time complexity (as noted in Lines 223-230), their GD-WL framework requires $O(\min\\{mn, n^{\omega}\\})$ time to precompute all-pair resistance distances (RDs), though the target algorithmic task --- determining biconnectivity -- only requires $O(m)$ time. Additionally, as stated in our Theorem 2, RDs can **directly imply edge biconnectivity**; thus, the **message-passing phase is actually unnecessary for this task in GD-WL framework** when RDs are precomputed. We argue that **overlooking the comparison between preprocessing time and the task's time complexity** leads to questionable conclusions.
> + Third, we also found that a CONGEST model proposed by [Pritchard, 2006] can solve the **edge biconnectivity problem in $O(D)$ rounds** (we add this in Lines 311-313 in red color). [Loukas, 2020] further suggests that the CONGEST model can handle many algorithms. These findings highlight that with unique IDs, MPGNNs might indeed solve the biconnectivity problem, supporting our view and **challenging studies that rely on WL tests** --- which they deem "weak" --- to define MPGNN expressiveness.
>
> **W2**:
>
> Thank you for your suggestion on clarifying this mismatch. Your review aligns with our discussion in the paper. Our main argument is that while existing works claim the proposed models' expressiveness advantage by proving they can perform tasks beyond the WL test's scope, this approach is questionable. The previous works' equating anonymous WL with MPGNNs is not entirely reasonable, and thus concluding that MPGNNs are weak because WL test is weak is also debatable. In fact, MPGNNs can perform certain algorithms (such as solving edge biconnectivity in $O(D)$ rounds within the CONGEST model [Pritchard, 2006], Lines 311-313). The logical flow of Section 3.2 is as follows:
> 1. The claim that the vanilla WL test has limited expressive power is true, as discussed in Figure 2. However, real-world graphs often contain rich features, and [Loukas, 2020] demonstrated that with unique IDs (and other assumptions), MPGNNs (Loukas used CONGEST to characterize) can perform a wide range of algorithmic tasks. Thus, using the anonymous WL test to characterize MPGNNs is debatable.
> 2. To address MPGNNs' "limited" expressiveness (stemming from the vanilla WL test's limitations, as many works use the WL test to characterize GNNs), some studies, such as [Zhang et al., 2023], incorporate additional features to enhance model expressiveness. Nonetheless, as outlined in (1), using the anonymous WL test as a characterization of MPGNNs is questionable. Consequently, demonstrating a model's expressiveness by proving it can perform tasks beyond the WL test's capabilities may not be entirely valid.
> 3. A more reasonable approach would be to compare these models to MPGNNs in a **non-anonymous setting**, as suggested in [Loukas, 2020]. Furthermore, evidence from [Suomela, 2013; den Berg et al., 2018; You et al., 2021; Abbound et al., 2021; Sato et al., 2021] indicates that the non-anonymous setting can enhance expressiveness, again highlighting the mismatch when works argue for "weak MPGNNs" but use additional features, breaking the WL test's anonymous setting to enhance expressiveness.
>
> We have revised the introduction in Section 3.2 (Lines 253-258) and highlighted the changes in red to clarify our points more effectively.

---

> ### Author Response · Authors · 2024-11-15
> **Initial Response to Reviewer YAM3 (2/3)**
>
> **W3**:
>
> Yes, the CONGEST model can still serve as an upper bound for computational capacity. Our point is that selecting the unrestricted CONGEST model as the computational model for GNNs would yield impractical outcomes, such as $O(m)$-depth GNNs that could theoretically solve $\mathsf{NP}$-$\mathsf{complete}$ problems on connected graphs (we appreciate your feedback on Theorem 4 and have now corrected this condition in red color, Lines 323). In reality, we cannot expect a polynomial-sized neural network to solve $\mathsf{NP}$-$\mathsf{complete}$ problems without error (unless $\mathsf{P} = \mathsf{NP}$). This misalignment results from overly strong assumptions about nodes' computational power. Our intention is to introduce **flexible constraints** on the computational resources class $\mathsf{C}$ to derive independent results, as is discussed in Lines 381-389. For instance, by setting $\mathsf{C}$ to a class reflecting MLPs, such as $\mathsf{TC}^0$, the resulting model would resemble "real-world" GNNs with MLPs as update functions. Alternatively, if nodes' update functions used transformer-based LLM agents enhanced by Chain-of-Thought (CoT) reasoning, which are claimed to solve problems in $\mathsf{P}$ [Merrill et al., 2024; Li et al., 2024], we could set $\mathsf{C} = \mathsf{P}$ and derive new theoretical results based on this adjustment. We hope our framework can inspire future research on graph agents, and have added it in red color in the revised PDF (Lines 384-387). As discussed in Lines 381-389, adjusting $\mathsf{C}$ in different ways may yield diverse outcomes. Thus, our RL-CONGEST framework serves as a **"framework scheme"** or **"framework template"**.
>
> **W4**:
>
> As noted in the first open problem in Section 5, deriving general resource-round tradeoffs for the RL-CONGEST model is challenging, and we leave this problem for future work.
>
> We also believe your statement "GNNs are usually implemented with fixed-size networks that run in constant time" may not be accurate, as a node $v$'s aggregation function takes at least $\Omega(d(v))$ time. Moreover, studies aligning GNNs' expressiveness with the WL test **assume that MLPs can execute $\mathsf{HASH}$ functions** for node recoloring --- an assumption whose practicality is also debatable. Your question supports our idea that WL tests may not be as straightforward as prior studies suggest, which aligns with our findings in **Theorem 5**. Regarding concerns about our framework's practicality, **Theorems 5-8** illustrate our RL-CONGEST model's application in analyzing the **unreasonableness of certain assumptions in previous studies**. We invite you to kindly review these examples.
>
> **Q1**:
>
> Please note that our paper aims to conduct a theoretical analysis that identifies issues in existing studies on the expressive power of GNNs and to propose a new framework that avoids these issues. We do not intend to design a specific GNN model with improved performance or expressiveness, nor to offer guidance for future work directed toward these goals.
>
> **Q2**:
>
> As mentioned in our response to W3, we do not treat our entire framework, including the RL-CONGEST model with preprocessing and postprocessing, as a "benchmark model". Rather, it functions as a "framework scheme" or "framework template". We hope our framework will assist future research on GNN expressiveness by helping to **avoid issues discussed in Section 3** and **encouraging a re-evaluation** of the validity of common assumptions in the field.
>
> **Q3**:
>
> We believe this point is addressed in Lines 381-389, and we reiterate it in our response to W3. Adjusting $\mathsf{C}$ in different ways may lead to varied outcomes. For example, setting $\mathsf{C} = \mathsf{R}$  (recursive languages, which Turing machines can decide) and network width $w = O(1)$ turns our RL-CONGEST model into the CONGEST model. Thus, the RL-CONGEST model can be seen as a generalization of the standard CONGEST model, allowing flexible settings on the computational resource class $\mathsf{C}$.
>
> **Q4**:
>
> As discussed in our response to W3, there is no universally "appropriate" complexity class for all GNN researchers. Researchers focused on current MPGNNs with MLP-based update functions might set $\mathsf{C} = \mathsf{TC}^0$ or $\mathsf{AC}^0$ to derive their theoretical results, while those interested in graph agents could set $\mathsf{C} = \mathsf{P}$. By setting $\mathsf{C} = \mathsf{R}$, our model also connects to CONGEST algorithms, so results proposed by Loukas [Loukas, 2020] are special cases within our RL-CONGEST framework.
>
> Thank you again for your detailed feedback. We hope our response addresses your concerns and questions to some extent, and we look forward to further discussions with you.

---

> > ### Author Response · Authors · 2024-11-15
> > **Initial Response to Reviewer YAM3 (3/3)**
> >
> > **References**:
> >
> > [Zhang et al., 2023] Bohang Zhang, Shengjie Luo, Liwei Wang, and Di He. Rethinking the Expressive Power of GNNs via Graph Biconnectivity. ICLR 2023.
> >
> > [Thiede et al., 2021] Erik H. Thiede, Wenda Zhou, Risi Kondor. Autobahn: Automorphism-based Graph Neural Nets. NeurIPS 2021.
> >
> > [Bouritsas et al., 2022] Giorgos Bouritsas, Fabrizio Frasca, Stefanos Zafeiriou, and Michael M. Bronstein. Improving Graph Neural Network Expressivity via Subgraph Isomorphism Counting. TPAMI, 2022.
> >
> > [Wollschlager et al., 2024] Tom Wollschlager, Niklas Kemper, Leon Hetzel, Johanna Sommer, and Stephan Gunnemann. Expressivity and Generalization: Fragment-biases for Molecular GNNs. ICML 2024.
> >
> > [Pritchard, 2006] David Pritchard. An Optimal Distributed Edge-Biconnectivity Algorithm. arXiv 2006.
> >
> > [Loukas, 2020] Andreas Loukas. What Graph Neural Networks Cannot Learn: Depth vs Width. ICLR 2020.
> >
> > [Suomela, 2013] Jukka Suomela. Survey of Local Algorithms. ACM Computing Surveys (CSUR), 45(2):24, 2013.
> >
> > [den Berg et al., 2018] Rianne van den Berg, Thomas N Kipf, and Max Welling. Graph Convolutional Matrix Completion. KDD 2018.
> >
> > [You et al., 2021] Jiaxuan You, Jonathan M Gomes-Selman, Rex Ying, and Jure Leskovec. Identity-aware Graph Neural Networks. AAAI 2021.
> >
> > [Abbound et al., 2021] Ralph Abboud, Ismail Ilkan Ceylan, Martin Grohe, and Thomas Lukasiewicz. The Surprising Power of Graph Neural Networks with Random Node Initialization. IJCAI 2021.
> >
> > [Sato et al., 2021] Ryoma Sato, Makoto Yamada, and Hisashi Kashima. Random Features Strengthen Graph Neural Networks. SDM 2021.
> >
> > [Merrill et al., 2024] William Merrill, and Ashish Sabharwal. The Expressive Power of Transformers with Chain of Thought. ICLR 2024.
> >
> > [Li et al., 2024] Zhiyuan Li, Hong Liu, Denny Zhou, and Tengyu Ma. Chain of Thought Empowers Transformers to Solve Inherently Serial Problems. ICLR 2024.

---

> > > ### Comment · Reviewer_YAM3 · 2024-11-16
> > >
> > > Thank you for your detailed response. However, I believe some claims still lack sufficient support.
> > >
> > > Regarding W2: It is indeed standard practice to use the anonymous Weisfeiler-Lehman (WL) test for analyzing graphs without node features, subsequently demonstrating that additional features can enhance a GNN's expressiveness by providing nodes with pseudo-identifiers. In this context, comparisons between unlabeled graphs (or the anonymous WL algorithm) and their labeled counterparts are entirely natural to illustrate how additional features improve expressiveness. While other graph features might contribute similarly, these are often more challenging to analyze, and they do not diminish the impact of the features under investigation.
> > >
> > > One of your main claims is that RL-CONGEST addresses issues identified in the literature, yet the explanation of how or why this is achieved remains unclear. On one hand, you propose your work as a framework template; on the other, you suggest that its specific application is left for future exploration. This raises a key question: If one were to design a new GNN architecture to overcome the issues highlighted in your paper, aside from reporting processing times (which is indeed essential for many reasons), how else would RL-CONGEST and your work be beneficial?
> > >
> > > For instance, considering your critique of Zhang et al.'s paper as a case study: How would you expect them to apply your framework to avoid the identified weaknesses? They already provide processing times, and your criticism relates to the edge biconnectivity task, where an analysis is required that goes beyond simply aligning to another computational model. Your point about underestimating preprocessing times seems to be based on the observation that the preprocessing has comparable or greater computational complexity than the problem being addressed. However, the objective of these GNN architectures is not necessarily to solve problems in the most efficient manner possible but rather to demonstrate that specific node features enable the GNN to make the right predictions for certain tasks. It is generally understood that GNNs will not match the efficiency of the best classical algorithms, and the works you reference do not make such claims.

---

> ### Author Response · Authors · 2024-11-17
> **Follow-Up Discussion with Reviewer YAM3 (1/2)**
>
> Dear Reviewer YAM3,
>
> Thank you for engaging in further discussion with us. We will begin by addressing your concerns regarding our case study, followed by a discussion of your other concerns.
>
> ### **Regarding Zhang et al.'s Paper as a Case Study**
>
> While the **total runtime** of GNNs for solving an algorithmic task does not necessarily need to outperform classical algorithms, this is not the focus of our argument. Instead, we emphasize that **researchers need to be careful** when the **preprocessing time** for features or graphs exceeds the algorithmic task's time complexity. If it happens, the **theoretical results may become questionable, as the precomputed features may directly solve the task**, rendering the GNNs less relevant. In such cases, attributing the results to GNN expressiveness might not be entirely appropriate.
>
> Take Zhang et al.'s paper as an example. In our Theorem 2, we demonstrate that $R(u, v) = 1$ is equivalent to the edge $(u, v)$ being a cut edge, and hence $G$ is not edge-biconnected. For the edge-biconnectivity task, the precomputed features (RDs) **directly provide the solution**, which reduces the role of message-passing to unnecessary redundancy. As such, this result highlights the **expressiveness of the precomputed features rather than that of the GNN** itself.
>
> To illustrate further, consider a binary classification task $\mathcal{X} \to Y = \\{0, 1\\}$, where the preprocessing $\mathcal{X} \to Z = \\{-1, 1\\}$ generates a binary feature $Z_v \in \\{-1, 1\\}$ for each sample $v$, such that $Z_v = -1$ iff $Y_v = 1$, and $Z_v = 1$ iff $Y_v = 0$. A simple model, such as an MLP or linear SVM, could easily solve the task using $Z_v$ as features. ***Does this suggest the model itself is expressive?*** We believe the answer is "No". ***It only shows "feature expressiveness" or "preprocessing expressiveness" rather than "model expressiveness".*** It is the preprocessing step, not the model, that contributes the expressiveness. Similarly, in Zhang et al.'s work, **the RDs alone suffice to solve edge-biconnectivity (Theorem 2)**, and message-passing adds no significant value. Moreover, the preprocessing time is substantially higher than the direct complexity of solving the problem algorithmically. (A more reasonable approach for this problem can be found in Lines 312-313, where, **with unique IDs, the CONGEST model solves the problem in $O(D)$ rounds without costly preprocessing such as computing RDs**.)
>
> We believe that disregarding preprocessing time relative to the algorithmic task's time complexity can lead to problematic interpretations. For another instance, consider using GNNs to solve $\mathsf{NP}$-$\mathsf{Complete}$ problems such as $\mathsf{MIN}$-$\mathsf{VERTEX}$-$\mathsf{COVER}$ or $\mathsf{HAMILTON}$-$\mathsf{CYCLE}$. Without constraints on preprocessing, the answers could be computed as binary features (e.g., whether each node is in the vertex cover or each edge is in the cycle) using classical algorithms. This would enable **a trivial one-layer GNN (or even a single neuron) to "solve" NP-complete problems** with these precomputed features, creating a very absurd conclusion of the model's expressiveness.
>
> We hope this clarification helps to clarify why we emphasize the importance of researchers exercising **caution when the preprocessing time (not total running time) exceeds the algorithmic task’s time complexity**.
>
> ----------
>
> We then address your other concerns as follows:
>
> **1. About Specific Applications**
>
> We have already demonstrated some initial results using our RL-CONGEST framework, as shown in Theorems 5-8. We kindly invite you to review these results. Please note that our exploration of this framework is in its early stages, and there is significant potential for future work, as outlined in Section 5.
>
> **2. How Our RL-CONGEST Framework Addresses the "Underestimated Preprocessing Time Complexity" Issue**
>
> The main point is the **importance of carefully analyzing the relationship between preprocessing time complexity and the time complexity of the chosen algorithmic task** to correctly evaluate a model's expressiveness. If the preprocessing time is less than or comparable to the algorithmic task's time, there are no inherent issues. However, if the preprocessing time significantly exceeds the algorithmic task's time, it becomes crucial to **analyze whether the resulting features or graphs can directly imply the solution to the algorithmic task**, as in the discussions above. If this is the case, the subsequent GNN model and message-passing steps are unnecessary. In such scenarios, it is more appropriate to attribute the success to "feature expressiveness" or feature engineering rather than GNN expressiveness, as the message-passing component becomes redundant.

---

> > ### Author Response · Authors · 2024-11-17
> > **Follow-Up Discussion with Reviewer YAM3 (2/2)**
> >
> > **3. How Our RL-CONGEST Framework Addresses the "Mismatch Between WL Test and Features" Issue**
> >
> > It is essential to remember that our framework is designed to analyze the expressive power of GNNs in **solving algorithmic tasks**, including the WL test and biconnectivity. The RL-CONGEST framework permits nodes to access unique IDs and utilize various features (e.g., distances), with **the key requirement being that the time complexity of computing these additional features must be explicitly stated**. When proposing a new model and demonstrating its expressiveness, authors should (or other researchers analyzing the model can) compare the preprocessing time complexity with the algorithmic task's complexity and ensure that the "feature expressiveness" issue is avoided.
> >
> > ----------
> > We hope this response addresses your concerns and provides clarity regarding the issues you raised. Thank you for your continued engagement in this discussion!

---

> > > ### Comment · Reviewer_YAM3 · 2024-11-18
> > >
> > > Thank you for your detailed response. I will be more direct with my questions, as I feel that the core issues are still not being addressed:
> > >
> > > Problem 1: While I agree that reporting processing times and being mindful of them is important, I do not see why RL-CONGEST is necessary for this analysis. You have yet to clearly explain how RL-CONGEST would be beneficial in future work on GNNs or how it could have been used by Zhang et al. to avoid the problems you mentioned. Furthermore, its applicability is limited by the fact that it assumes node IDs, which most models do not.
> > >
> > > Problem 2: The “Mismatch Between WL Test and Features” issue is still not fully clarified. Your recent reply largely repeated points from your previous answers, which do not directly address this specific concern. See also the comment in my previous comment "regarding W2". The paper still lacks clarity in this regard.

---

> > > > ### Author Response · Authors · 2024-11-19
> > > >
> > > > Dear Reviewer YAM3,
> > > >
> > > > Thank you for your additional comments and for summarizing your questions in a more direct way. We will address your concerns concisely as follows:
> > > >
> > > > ----------
> > > >
> > > > ### **For Problem 1**:
> > > >
> > > > **Regarding your concern about the RL-CONGEST model's benefits for future work**:
> > > >
> > > > The RL-CONGEST framework partly addresses this issue by requiring the explicit reporting of both preprocessing time and the algorithmic task's time complexity. In applications such as Zhang et al.'s work or future studies, if these two complexity bounds are reported and it is observed that the preprocessing time exceeds the algorithmic task's time, researchers are reminded to immediately re-evaluate whether the additional features directly solve the algorithmic task. While our RL-CONGEST model cannot entirely prevent such issues — just as no computational model (e.g., the RAM model) can stop someone from spending significantly more time to solve a simpler problem — it does provide a structured framework to warn researchers of this potential issue. By adhering to RL-CONGEST's analytical approach, researchers are prompted to consider this problem critically.
> > > >
> > > > ***We have revised Lines 239-242 and Lines 376-379 in blue color in our updated PDF to make this point clearer***.
> > > >
> > > >
> > > > **Regarding "its applicability is limited by the fact that it assumes node IDs, which most models do not"**:
> > > >
> > > > We respectfully disagree with this for the following reasons:
> > > > 1. Allowing nodes to access IDs actually relaxes the constraints and generalizes the WL tests, as nodes in the RL-CONGEST model have the flexibility to decide whether or not to use node IDs as input features.
> > > > 2. Models that incorporate additional features inherently assume the availability of node IDs, and we will elaborate on this point further in our response to your Problem 2.
> > > >
> > > > ----------
> > > >
> > > > ### **For Problem 2**:
> > > >
> > > > **Also on benifits for future work**:
> > > >
> > > > In brief, although the anonymous WL test has almost become a standard benchmark for works on GNN expressiveness, our suggestion for future works is that aligning GNN expressiveness with the anonymous WL test is not an appropriate approach and it is more reasonable to analyze GNN expressiveness by showing the algorithms they can perform under non-anonymous settings.
> > > >
> > > >
> > > > **Regarding your comment "comparisons between unlabeled graphs and their labeled counterparts are entirely natural to illustrate how additional features improve expressiveness" in "regarding W2"**:
> > > >
> > > > We disagree for the following reasons:
> > > > 1. Computing additional features (e.g., resistance distances) inherently requires treating nodes in a non-anonymous and distinguishable manner, thereby violating the anonymous setting. For instance, in Zhang et al.'s work, if matrix inversion is used, nodes must first be assigned unique IDs for the computation.
> > > > 2. Additional features computed "externally" cannot be considered as enhancing the GNN model's expressiveness. These features are derived from models outside the GNN itself. Drawing concepts from theoretical computer science, if a GNN can compute the required features internally with node IDs, it can be considered expressive enough to solve the task. Otherwise, the GNN is merely solving the task with the aid of a feature oracle (e.g., a resistance distance oracle in Zhang et al.'s work), which shifts the expressiveness to the oracle rather than the GNN.
> > > >
> > > > Since computing features implicitly relies on node IDs, our claim is: why not explicitly allow nodes to know their IDs? For application example, prior work has shown that with unique node IDs, CONGEST (and also our RL-CONGEST framework since it's a generalization) can solve the edge-biconnectivity problem in $O(D)$ rounds. This is a more reasonable expressiveness result achieved by removing the anonymity constraint and allowing nodes to access their IDs or other features, as proposed by our RL-CONGEST framework.
> > > >
> > > > ***We have revised Lines 284-290 and Lines 381-383 in blue color in our updated PDF to make this point clearer.***
> > > >
> > > > ----------
> > > >
> > > > Thank you again for your feedback. We hope this response clarifies our points further and addresses your concerns.

---

> > > > > ### Comment · Reviewer_YAM3 · 2024-11-19
> > > > >
> > > > > Thank you for your response. I believe I now understand part of the misunderstanding. As you have reiterated multiple times in your last reply and now explicitly state in the paper, you assume that the “precomputation of additional features (e.g., through matrix inversion to compute RDs) requires nodes to be assigned IDs”. This assumption underpins your argument for always providing node IDs. However, this conclusion is fundamentally flawed. Popular features like subgraph counts (e.g., triangle counts for nodes) do not require fixed node IDs and can be computed in a permutation-equivariant manner. Any arbitrary ordering of the graph suffices for the computation of these features. Indeed, GNNs rarely use node IDs, even when such features are employed, because node IDs inherently break permutation-equivariance, a core design principle of GNNs that facilitates generalization. Consequently, incorporating node IDs into your proposed computational framework compromises its relevance for the majority of GNN applications.
> > > > >
> > > > > Without assuming node IDs, we can also still conclude that externally computed features enhance the GNN's expressiveness while maintaining permutation-equivariance. These features, therefore, should be analyzed under the framework of a (non-anonymous) WL test, as has been done in many prior works.
> > > > >
> > > > > Regarding your question:
> > > > > > Since computing features implicitly relies on node IDs, our claim is: why not explicitly allow nodes to know their IDs?
> > > > >
> > > > > While features can be designed to maintain permutation-equivariance, this is not guaranteed for a GNN if it relies on an arbitrary ordering of node IDs. Moreover, computing canonical IDs to address this issue is computationally infeasible. Again, in practice, node IDs are rarely used.
> > > > >
> > > > > For Problem 1 and the necessity of RL-CONGEST: I believe we agree that, beyond accurately reporting processing times (as, for example, Zhang et al. already does) and considering whether tasks can be addressed directly from precomputed features, RL-CONGEST is not essential. Specifically, RL-CONGEST does not provide insight into how predictions can be directly derived from features. Asserting that researchers should adopt RL-CONGEST merely because they report processing times or analyze features feels like an overreach.
> > > > >
> > > > > Please correct me if I'm wrong.

---

> > > > > > ### Author Response · Authors · 2024-11-21
> > > > > >
> > > > > > Dear Reviewer YAM3,
> > > > > >
> > > > > > Thank you for your reply. We appreciate the opportunity to further clarify our claims.
> > > > > >
> > > > > > ----------
> > > > > >
> > > > > > ### **For your first concern:**
> > > > > >
> > > > > > **In two sentences:** Providing unique identifiers does not necessarily break equivariance or invariance. Our RL-CONGEST framework allows nodes to **know their IDs** but does **not enforce their use as features**, thereby offering flexibility.
> > > > > >
> > > > > > We disagree with the assertion that "node IDs inherently break permutation-equivariance", as this is a misunderstanding for the following reasons:
> > > > > > 1. The RL-CONGEST model only requires nodes to have unique identifiers to ensure they are uniquely distinguishable. There are no constraints preventing researchers from analyzing equivariance or invariance by permuting node IDs and further analyze.
> > > > > > 2. In practical implementations (e.g., PyG), nodes also have been **assigned IDs to manage their features**. However, this setting **does not conflict with equivariance or invariance** since models can freely choose whether or not to use unique IDs as input features. Our RL-CONGEST just clearly states that nodes can have be uniquely identified, which is not stricter than practical implementation.
> > > > > >
> > > > > > Thus, our framework represents a relaxation of the WL tests rather than a contradiction.
> > > > > >
> > > > > > Again, consider Zhang et al.'s GD-WL test. Under a non-anonymous setting, RL-CONGEST can solve the edge-biconnectivity if nodes have unique IDs. However, this result only assumes that nodes are distinguishable, and **no specific "canonical" ID** assignment is required. If one ID assignment solves the problem, **any permuted ID assignment would also work**, preserving the flexibility inherent in permutation-invariance.
> > > > > >
> > > > > > ----------
> > > > > >
> > > > > > ### **For your second concern:**
> > > > > >
> > > > > > **In one sentence**: While a computational model alone cannot entirely prevent certain issues, our analysis framework (comprising preprocessing, message-passing within the RL-CONGEST model, and postprocessing) functions **as a whole** to mitigate these concerns.
> > > > > >
> > > > > > It is true that one RL-CONGEST computational model component alone cannot entirely avoid these issues. However, the integrated framework, when applied comprehensively, helps highlight and address these problems. This does not imply the RL-CONGEST model alone is meaningless; rather, the framework as a whole must be considered in its entirety and **cannot be devided into isolated components**.
> > > > > >
> > > > > > ----------
> > > > > >
> > > > > > Thank you again for your thoughtful engagement.

---

> > > > > > > ### Comment · Reviewer_YAM3 · 2024-11-23
> > > > > > >
> > > > > > > Once again, I appreciate the effort you’ve made to clarify your points. However, I feel that some of my concerns remain unresolved, and I’d like to summarize them clearly one last time:
> > > > > > >
> > > > > > > RL-CONGEST assumes node IDs. While it’s true that any unique assignment of IDs can work, this misses the point entirely. Canonical node IDs, which would be one way to assign such identifiers that are permutation-equivariant, are computationally infeasible to compute. Moreover, many GNNs do not rely on node IDs or random features that make nodes unique. However, you are acting like all of them would. You should at least admit that this does not hold for all GNNs and clearly communicate this.
> > > > > > >
> > > > > > > > In practical implementations (e.g., PyG), nodes also have been assigned IDs to manage their features.
> > > > > > >
> > > > > > > This is merely an implementation detail entirely hidden from the GNN itself. PyG ensures computations are conducted in a permutation-equivariant manner, independent of graph order. This is fundamentally different from models that explicitly use IDs as input features, which would violate these guarantees. I’m not sure why you are even mentioning this here, because it’s really not relevant to the problem we are discussing.
> > > > > > >
> > > > > > > > However, this setting does not conflict with equivariance or invariance since models can freely choose whether or not to use unique IDs as input features.
> > > > > > >
> > > > > > > While it is true that models can technically choose their inputs, GNNs that align with RL-CONGEST by incorporating node IDs diverge from widely used GNN practices. This raises significant concerns about how RL-CONGEST can meaningfully analyze GNNs, which deliberately avoid using node IDs to preserve their core properties. Again, **analyzing a GNN that is not able to uniquely identify nodes with RL-CONGEST is not sensible, as the additional IDs make RL-CONGEST more powerful by providing this capability.** Consequently, the complexity bounds you will get in the RL-CONGEST model will generally not hold for the GNN you want to analyze.
> > > > > > >
> > > > > > > Regarding the second concern:
> > > > > > > Your responses do not address my repeated concerns about its practical relevance or necessity for analyzing GNNs. Beyond reporting processing times, RL-CONGEST offers no clear benefit in understanding how GNNs utilize features or make predictions.
> > > > > > >
> > > > > > > This represents my final attempt to clarify this issue, as I sense there has been a persistent reluctance to directly address the core of my concerns. Until this issue is adequately addressed, I feel compelled to lower my score to a reject, as the paper’s assumptions currently appear misaligned with some widely-used GNN practices. If you still hold a differing perspective, I would encourage you to consider the points I’ve outlined carefully.

---

> ### Author Response · Authors · 2024-11-24
>
> Dear Reviewer YAM3,
>
> It seems that there are two key points where we may not yet have reached a consensus, and we would like to further clarify our perspective.
>
> ----------
>
> ### **On "node IDs" and "non-anonymity":**
>
> These terms refer to **distinct features** that allow nodes to be distinguishable (e.g., $[n] = \\{0, 1, \cdots, n - 1\\}$ would also suffice).
>
> ***We have updated our PDF, replacing "anonymous" with "identical-feature" and "non-anonymous" with "distinct-feature" or "unique-feature" to make these concepts clearer and more accessible to readers.*** The modifications are highlighted in magenta, and we invite you to review them.
>
> The **distinct-feature setting** is commonly applied in **almost all existing models**, as listed below:
> 1. LINKX [Lim et al., 2021]:
>
> LINKX uses:
> - $\mathbf{H}^{(\mathbf{A})} = \mathrm{MLP}_{\mathbf{A}}(\mathbf{A})$
> - $\mathbf{H}^{(\mathbf{X})} = \mathrm{MLP}_{\mathbf{X}}(\mathbf{X})$
> - $\mathbf{Y} = \mathrm{MLP}(\sigma(\mathbf{W}[\mathbf{H}^{(\mathbf{A})}; \mathbf{H}^{(\mathbf{X})}] + \mathbf{H}^{(\mathbf{A})} + \mathbf{H}^{(\mathbf{X})}))$
>
> The $\mathbf{H}^{(\mathbf{A})}$ term can be reformulated as $\mathrm{MLP}'(\sigma(\mathbf{A} \cdot \mathbf{I} \cdot \mathbf{W}))$, which uses the identity matrix (unique node features).
>
> 2. GCN, GAT, and other models which can be applied to real-world datasets:
>
> These models use real-world node features which are unique and distinguishable with high probability. Actually, **all models that are applicable to real-world datasets fall into this category**.
>
> 3. GNN expressiveness works (e.g., [Loukas, 2020; Sato et al., 2021]):
>
> These works use random features, which are unique with high probability. For example, assigning each node a feature randomly chosen from $[n^4]$ would result in distinct features with high probability.
>
> 4. GD-WL framework:
> In the GD-WL framework by Zhang et al., resistance distances $R(s, t)$ are used as features. Since $R(s, t) = 0$ iff $s = t$, each row of the resistance distance matrix is unique, creating distinguishable node features.
>
> Actually, according to our theory, they are all capable of solving the biconnectivity problem using the unique features.
>
> ***We have also updated our PDF to include the above discussions in Section 3.2, highlighted in magenta.***
>
> ----------
>
> ### **On whether unique features break equivariance or invariance:**
>
> Unique features do not break permutation equivariance or invariance. Instead, it is the **properties of the update functions and pooling layers** that determine whether a GNN model is equivariant or invariant. For example, in LINKX, when performing node classification, $\mathrm{MLP}(\mathbf{A})$ ensures permutation equivariance. To achieve permutation invariance for graph classification, we only need to add a permutation-invariant pooling layer after this step.
>
> Similarly, consider Dijkstra's single-source shortest path algorithm. Unique IDs are used solely to determine whether the shortest path to a node has been found. The resulting shortest path distance vector is always permutation equivariant. This demonstrates that it is not the presence of unique IDs but rather the design of the update function that determines whether a GNN model is permutation equivariant or invariant.
>
> ----------
>
> We hope these clarifications address your concerns and further illustrate the flexibility of our framework. Thank you again for your feedback.
>
> ----------
>
> **Reference:**
>
> [Lim et al., 2021]. Large Scale Learning on Non-Homophilous Graphs: New Benchmarks and Strong Simple Methods. NeurIPS 2021.
>
> [Sato et al., 2021] Ryoma Sato, Makoto Yamada, and Hisashi Kashima. Random Features Strengthen Graph Neural Networks. SDM 2021.

---

> > ### Comment · Reviewer_YAM3 · 2024-11-24
> >
> > Your justification for assuming unique IDs (or the distinct-feature setting, as you now call it) is becoming increasingly convoluted. While it is true that there are specific GNNs or datasets where unique IDs may be applicable, this assumption does not hold universally across a wide array of tasks. In particular, many synthetic tasks often do not satisfy this condition.
> >
> > Your claim that unique and distinguishable features exist for all models on real-world tasks is especially problematic:
> >
> > > These models use real-world node features which are unique and distinguishable with high probability. Actually, all models that are applicable to real-world datasets fall into this category.
> >
> > This assertion is unsupported in your response and your paper, there is not even a single citation provided to substantiate it. Is this really the case for widely used datasets? If so, can you provide evidence to demonstrate this?
> >
> > Moreover, my earlier argument regarding the impact of adding unique IDs on permutation equivariance/invariance was misunderstood again. I never claimed that unique features would generally break permutation equivariance/invariance. But adding IDs in an arbitrary order as node features (let’s say to a featureless dataset to make it align with the “distinct-feature setting”) inherently breaks this property because the output from the GNN will depend on the chosen order. Unless this order is constructed in a permutation-equivariant or invariant way (e.g., using computationally infeasible canonical IDs), there is no guarantee that permutation-equivariance or invariance will be preserved. Just assume that you are running GIN on graphs with node IDs, the output will depend on the chosen order. This serves as the justification for why most GNNs do not simply add unique identifiers as node IDs, just to provide context for this part of the discussion. However, this point is less critical compared to the key issue I highlighted earlier.
> >
> > Your previous reply continues to sidestep the core issue of why distinct features or node IDs can be assumed, and by now, I am beginning to question whether this is being done intentionally.

---

> > > ### Comment · Reviewer_YAM3 · 2024-11-24
> > >
> > > Based on the comments from other reviewers, it seems I am not alone in raising concerns about your argumentation regarding the assumption of unique features or node IDs. Despite several opportunities to address these issues, you have not provided sufficient support for your claims. Instead, you have introduced additional unsupported assertions to justify key aspects of your paper. Consequently, I have decided to lower my score to a reject.

---

> > > > ### Author Response · Authors · 2024-11-25
> > > >
> > > > Dear Reviewer YAM3,
> > > >
> > > > We are deeply grateful for your continued engagement in the discussion with us. We regret your decision to lower the score, but we respect it. We would like to make the following clarifications for you and the other reviewers:
> > > >
> > > > 1. In scenarios where unique features are not necessary (e.g., molecular property classification), the expressive power of MPGNNs and high-order GNNs has been extensively studied [Xu et al., 2019; Cai et al., 1989; Grohe, 1998; Grohe, 2017]. However, their capabilities remain limited to (anonymous) WL tests and are insufficient for tasks such as biconnectivity decision. Consequently, while analyzing GNNs without features (or with identical features) is important in applications such as molecular property classification, this was not the focus of our paper.
> > > > 2. Many researchers have explored adding features to enhance expressiveness. For example, in Zhang et al.'s work, distances were added as features to solve the biconnectivity decision problem. Therefore, our goal is to determine the types of features a GNN requires to solve specific graph problems, such as biconnectivity.
> > > > 3. Surprisingly, our theory demonstrates that any distinct features (e.g., the identity matrix $\mathbf{I}$) can achieve this goal, thereby generalizing Zhang et al.'s results. Moreover, our findings align with Loukas's experimental results.
> > > >
> > > > We deeply appreciate your effort and are not pressing for a score change. Our intention is solely to ensure that our ideas are clearly conveyed to you and the other reviewers, minimizing any potential misunderstandings.
> > > >
> > > > **Reference:**
> > > >
> > > > [Xu et al., 2019] Keyulu Xu*, Weihua Hu*, Jure Leskovec, Stefanie Jegelka. How Powerful are Graph Neural Networks? ICLR 2019.
> > > >
> > > > [Cai et al., 1989] Jin-yi Cai, Martin Furer, and Neil Immerman. An optimal lower bound on the number of variables for graph identification. FOCS 1989.
> > > >
> > > > [Grohe, 1998] Martin Grohe. Finite variable logics in descriptive complexity theory. Bull. Symb. Log., 4(4):345–398, 1998.
> > > >
> > > > [Grohe, 2017] Martin Grohe. Descriptive Complexity, Canonisation, and Definable Graph Structure Theory, volume 47 of Lecture Notes in Logic. Cambridge University Press, 2017.

---

### Official Review · Reviewer_ftna · 2024-11-02

**Soundness:** 3
**Presentation:** 2
**Contribution:** 3
**Rating:** 6
**Confidence:** 4

**Summary:**

This paper introduces a new computational model—the Resource Constrained CONGEST (RL-CONGEST) model—designed to address the inconsistencies and irrationalities in the current analysis of GNNs' expressivity. The RL-CONGEST model forms a framework for analyzing the expressivity of GNNs by introducing resource constraints and optional pre-processing and post-processing stages. Through this framework, it can reveal computational issues, such as the difficulty of hash function computation in the WL test and the role of virtual nodes in reducing network capacity, thereby providing theoretical support for understanding and improving the expressivity of GNNs.

**Strengths:**

1. This paper clearly identifies three key issues that are commonly overlooked in the current analysis of GNNs' expressivity, which represents a relatively novel perspective.

2. The RL-CONGEST model proposed in this paper provides a theoretical framework for the expressivity of GNNs.

3. The paper conducts an in-depth analysis of the computational complexity of the WL test, which is valuable for understanding the potential and limitations of GNNs and also demonstrates the paper's solid theoretical foundation.

**Weaknesses:**

1. Lack of Empirical Validation: The paper lacks empirical experiments to support the theoretical results.

2. Lack of Guidance on Model Design: The paper does not clearly propose how to use the RL-CONGEST model to enhance the expressive power of GNNs. Although a theoretical framework is presented, there are no specific implementation details or design principles provided.

**Questions:**

1.Can you provide some empirical experiments to verify the correctness of the analysis results of the RL-CONGEST model?

2.Is the RL-CONGEST model applicable to the analysis of all different types of GNNs and tasks on graphs?

3.Do the computational resource limitations mentioned in the article reflect the constraints in the real world? Are these limitations applicable to all types of GNNs?

4.Can you further provide design guidance on how to use this method to improve the model's expressive power?

5.Since the article mentions analyzing the expressive power of GNNs under resource constraints, is the RL-CONGEST model applicable to learning tasks on large graphs that are also resource-constrained?

---

> ### Author Response · Authors · 2024-11-15
> **Initial Response to Reviewer ftna**
>
> Dear Reviewer ftna,
>
> Thank you for your time and effort in reviewing our paper. We would like to address your concerns and questions as follows:
>
> **W1, W2, and Q1**:
>
> Thank you for these suggestions. Please note that our primary goal is to conduct a theoretical analysis to highlight issues in existing studies on the expressive power of GNNs and to propose a new analytical framework that avoids these issues. Our intention is not to design a specific GNN model with improved performance or expressiveness, nor to provide guidance for future work aimed at doing so.
>
> **Q2**:
>
> Yes, we believe the answer is affirmative. As discussed in Section 3.1 (Lines 201-209 in the revised PDF) of our paper, many GNNs conform to the "preprocessing-then-message-passing" framework. From a practical standpoint, mainstream GNN libraries, such as PyG (torch-geometric), implement GNNs with a "MessagePassing" base class, meaning that models built with these libraries naturally align with this framework. High-order GNNs, subgraph GNNs, and GNNs with additional features can also be implemented in these libraries by first constructing the $k$-WL graphs, subgraphs, or graphs with additional features, followed by message-passing operations, thereby fitting into the "preprocessing-then-message-passing" framework. As a result, we believe our analytical framework applies to the analysis of most GNNs.
>
> **Q3**:
>
> Yes, exactly. The resource limitation we consider reflects the constraints of real-world GNNs. Many GNNs use MLPs (or similar neural network models) as update functions, but these are far from Turing-complete, as required in condition (3) of Theorem 3 (Lines 308-310). For instance, we cannot expect an MLP of polynomial size to solve $\mathsf{NP}$-$\mathsf{complete}$ problems without error (unless $\mathsf{P} = \mathsf{NP}$). However, directly setting the resource limitation class $\mathsf{C}$ as a class specifically reflecting MLPs (e.g., $\mathsf{TC}^0$) would reduce flexibility, as future GNNs may adopt new architectures for update functions. Our framework remains adaptable by allowing adjustments to the class $\mathsf{C}$. For a hypothetical example, if we implemented the nodes' update functions with transformer-based LLM agents enhanced by Chain-of-Thought (CoT), which are claimed to solve problems within $\mathsf{P}$ **[Merrill et al., 2024; Li et al., 2024]**, we could set $\mathsf{C} = \mathsf{P}$ and derive new theoretical results based on this adjustment. We hope that our analysis framework can also inspire future work on graph agents, and have added it in red color in the revised PDF (Lines 384-387). This point is already discussed in Lines 381-389 (of the revised PDF), adjusting $\mathsf{C}$ in different ways may lead to varied outcomes. In this way, our RL-CONGEST framework serves as a "framework scheme" or "framework template".
>
> **Q4**:
>
> As noted in our response to W1, W2, and Q1, our focus is on identifying issues in the existing analysis of GNN expressiveness and introducing a new framework for this analysis, rather than providing a guideline for future work to enhance expressiveness. The idea is that once researchers propose a new GNN model, they can analyze its expressive power using our framework, rather than using ad-hoc methods, which may have limitations as discussed in Section 3.
>
> **Q5**:
>
> Yes, we believe this is correct. We leave the exploration of specific tasks to future studies, and we also outline other open questions for further research in Section 5.
>
> Thank you again for reviewing our paper. We hope this response clarifies our approach and addresses your questions. We look forward to any further discussions with you.
>
> **References**:
>
> **[Merrill et al., 2024]** William Merrill, and Ashish Sabharwal. The Expressive Power of Transformers with Chain of Thought. ICLR 2024.
>
> **[Li et al., 2024]** Zhiyuan Li, Hong Liu, Denny Zhou, and Tengyu Ma. Chain of Thought Empowers Transformers to Solve Inherently Serial Problems. ICLR 2024.

---

> ### Author Response · Authors · 2024-11-21
> **Follow-Up on Rebuttal Discussion**
>
> Dear Reviewer ftna,
>
> As we are now midway through the rebuttal phase, we want to kindly follow up to ensure that our responses have adequately addressed your concerns. Your feedback is highly valued, and we still looking forward to further discussion to clarify or expand on any points as needed. Please feel free to share any additional thoughts or questions you might have.
>
> Thank you once again for your time and effort in reviewing our paper.

---

### Meta-Review · Area_Chair_N1Rs · 2024-12-19

**Metareview:**

The paper assumes that GNNs can leverage unique node IDs or distinct features to enhance expressiveness, a practice fundamentally misaligned with the core principles of GNN design. Node IDs break permutation invariance/equivariance, which is critical for generalization across graph distributions. While the authors assert that their RL-CONGEST framework does not enforce the use of IDs, the reliance on them undermines the framework’s relevance to most real-world GNN applications. Multiple reviewers raised concerns about the framework's practical relevance and assumptions, particularly regarding the use of unique IDs and the inductive learning setting. The authors repeatedly failed to address these concerns directly. Overall, a recommendation of rejection is made.

**Additional Comments On Reviewer Discussion:**

1. Node IDs and Practical GNNs:

Concerns Raised: Reviewers (ftna, YAM3, b62D, DTJH) highlighted that using unique node IDs in the RL-CONGEST framework contradicts the design principles of permutation invariance/equivariance in GNNs, which are critical for generalization to unseen graphs. They also questioned how unique IDs align with real-world scenarios and datasets, where node features are often not unique.
Author Response: The authors argued that unique IDs are optional in their framework and do not break equivariance/invariance. They claimed that real-world datasets often have distinguishable features, allowing nodes to be uniquely identified. However, the authors failed to provide concrete evidence for this claim, and their examples (e.g., LINKX) were shown to be inapplicable to inductive settings.

2. Practical Relevance of RL-CONGEST:

Concerns Raised: Reviewers (YAM3, b62D, DTJH) questioned the practical applicability of RL-CONGEST, particularly in training GNNs for tasks like edge connectivity on unseen graphs with varying sizes and distributions. They asked for concrete examples of how the framework could be used to train MPNNs or compare existing GNN architectures.
Author Response: The authors provided an example of a distributed algorithm for edge-biconnectivity but admitted that their framework does not address the practical trainability of GNNs for such tasks. They claimed that training models falls outside the scope of their paper, which reviewers viewed as avoiding the core question of applicability.

3. Inductive Learning and Distinct Features:

Concerns Raised: Reviewers (b62D, YAM3) highlighted that the authors misunderstood inductive learning. They noted that using unique IDs or fixed-size matrices (e.g., LINKX) is incompatible with generalizing across graphs with varying sizes and distributions.
Author Response: The authors attempted to address this by suggesting padding techniques to handle varying graph sizes, comparing their approach to NLP. However, reviewers argued that this response failed to address inductive learning requirements for graphs and further demonstrated a misunderstanding of the core issue.

4. Framework’s Scope and Impact:

Concerns Raised: Reviewers (b62D, YAM3) expressed doubts about the framework’s ability to analyze expressiveness in real-world tasks. They questioned whether RL-CONGEST could quantify the impact of features like resistance distance or directly compare existing GNN architectures.
Author Response: The authors reiterated that RL-CONGEST focuses on algorithmic tasks and not downstream performance. They suggested their work encourages reevaluation of existing assumptions but provided no actionable insights for practitioners.

The first point weighs most in my decision, as it reflects the authors lack basic understanding of permutation invariance.

---

### Decision · Program_Chairs · 2025-01-22

Reject